# A Unification of Discrete, Gaussian, and Simplicial Diffusion

**Nuria Alina Chandra**\*  **Yucen Lily Li**\*  **Alan N. Amin**\*  **Alex Ali**
New York University    New York University    New York University    New York University

**Joshua Rollins**    **Sebastian W. Ober**    **Aniruddh Raghu**    **Andrew Gordon Wilson**
CUNY    BigHat Biosciences    BigHat Biosciences    New York University

## Abstract

To model discrete sequences such as DNA, proteins, and language using diffusion, practitioners must choose between three major methods: diffusion in discrete space, Gaussian diffusion in Euclidean space, or diffusion on the simplex. Despite their shared goal, these models have disparate algorithms, theoretical structures, and tradeoffs: discrete diffusion has the most natural domain, Gaussian diffusion has more mature algorithms, and diffusion on the simplex in principle combines the strengths of the other two but in practice suffers from a numerically unstable stochastic processes. Ideally we could see each of these models as instances of the same underlying framework, and enable practitioners to switch between models for downstream applications. However previous theories have only considered connections in special cases. Here we build a theory unifying all three methods of discrete diffusion as different parameterizations of the same underlying process: the Wright-Fisher population genetics model. In particular, we find simplicial and Gaussian diffusion as two large-population limits. Our theory formally connects the likelihoods and hyperparameters of these models and leverages decades of mathematical genetics literature to unlock stable simplicial diffusion. Finally, we relieve the practitioner of balancing model trade-offs by demonstrating it is possible to train a single model that can perform diffusion in any of these three domains at test time. Our experiments show that Wright-Fisher simplicial diffusion is more stable and outperforms previous simplicial diffusion models on conditional DNA generation. We also show that we can train models on multiple domains at once that are competitive with models trained on any individual domain.

## 1 Introduction

To generate high quality sequences conditioned on desired properties, practitioners build diffusion models of language, DNA, and proteins (Sahoo et al., 2024; Sarkar et al., 2024; Alamdari et al., 2023; Li et al., 2024). These models corrupt each letter in a sequence – the "forward" process – and train a model to reverse that corruption – the "backward" process. A model which has been trained to de-noise can be used for high-quality conditional generation (Wang et al., 2024b), for optimization (Gruver et al., 2023), and myriad other downstream tasks (Luo et al., 2022; Baron et al., 2025).

A practitioner has three main choices of forward process (Fig. 1b), each with their own strengths:

1. **Discrete:** occurs in the most natural domain (Campbell et al., 2022).

2. **Gaussian:** has more mature sampling and training procedures (Dieleman et al., 2022).

3. **Simplicial:** in theory inherits the continuous algorithms of Gaussian diffusion while in a natural space, but in practice suffers from numerical instability (Avdeyev et al., 2023).

Unfortunately, there is little theoretical infrastructure to compare these models, and thus practitioners have little tacit knowledge to rely on when selecting or designing a model. This gap in understanding is particularly evident in two basic comparison problems which have yet to be solved. First, despite

---

\*Equal contribution.

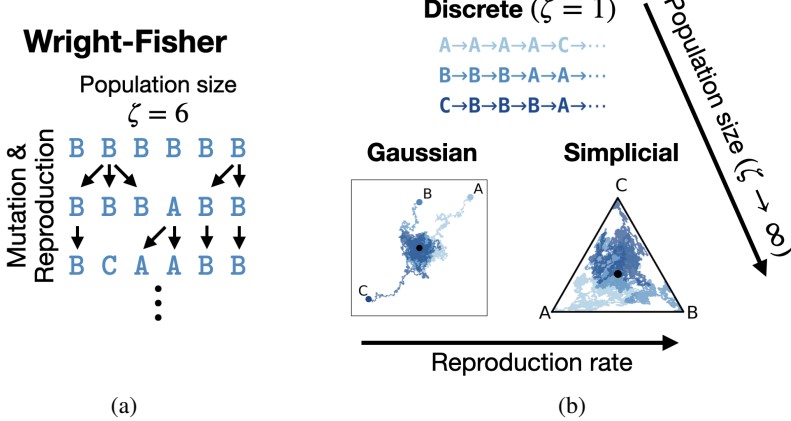

(a)                                                                (b)

Figure 1: **Discrete, Gaussian, and Simplicial diffusion for discrete data are unified by Wright-Fisher diffusion.** **(a)** Wright-Fisher diffusion with population size $\zeta = 6$, showing mutation and reproduction processes across generations. **(b)** The three diffusion methods emerge as different limits of Wright-Fisher: discrete diffusion corresponds to $\zeta = 1$, while Gaussian and simplicial diffusion arise as $\zeta \to \infty$ with zero and non-zero reproduction rates.

models from the three frameworks achieving similar likelihood values, there is a belief that the "continuous-space likelihood is not directly comparable with discrete-space likelihood" (Avdeyev et al., 2023). Second, forward processes in each of these models are specified by hyperparameters with vastly different interpretations. It is unclear how to qualitatively compare the assumptions embedded into each set of hyperparameters across models.

Here we address these theoretical and practical challenges by unifying these streams with a process from human population genetics – the Wright-Fisher (WF) model. Our contributions are as follows:

- We formally prove all three methods are instances of WF (Fig. 1). In particular discrete diffusion corresponds to the WF model with a population size of 1, and simplicial and Gaussian diffusion correspond to large population limits with and without reproduction.

- We use this connection to answer the two comparison questions above. Surprisingly, we show that likelihoods can only be compared in some cases, depending on a seemingly inconsequential parameterization choice introduced for only discrete diffusion models in Austin et al. (2021) which we call the **hollow parameterization**.

- We apply our theory to explain and solve the instability of simplicial diffusion by leveraging decades of mathematical genetics literature. We show that this stable simplicial diffusion is superior in conditional generation of DNA.

- We leverage our theory to show that a particular parameterization choice – the **sufficient-statistic parameterization** – allows one to train a single model that can perform diffusion on all three domains at test time[1]. We show in experiment that models trained this way are competitive with models trained on single domains. This removes the necessity for the practitioner to choose a particular model before training.

Our code is available at `https://github.com/yucenli/unify-diffusion`.

## 2 RELATED WORK

We discuss past unification theories and attempts at stable simplicial diffusion. In App. A we discuss related works in classical diffusion theory, and parameterizations of diffusion models.

**Theories unifying discrete and continuous diffusion**    Winkler et al. (2024) indirectly used a result from (Stone, 1963) to connect the special case of one-dimensional, unbiased discrete diffusion to

---

[1]Of independent interest, it also explains the root of the noted "time-invariance" of masking diffusion and extends this property to every diffusion model. We discuss this in App. D.

one-dimensional Gaussian diffusion. They use this observation to heuristically argue, or conjecture, the convergence of the backwards processes as well. Sahoo et al. (2025) suggested that by taking Gaussian diffusion and applying argmax, one recovers discrete diffusion. [2] They used this insight to answer the loss comparison problem by proving that the ELBO of discrete diffusion is always superior to that of continuous diffusion. Unfortunately, this is based on a mathematical error (details in App. B): by applying argmax to Gaussian diffusion one does not get a Markov process, a property which was crucial to their proof of the loss comparison question. In our approach, we build a mathematically rigorous foundation to compare these models.

**Stable simplicial diffusion models** Richemond et al. (2022) and Avdeyev et al. (2023) suggest diffusion on a simplex using two processes used in finance – the "Cox-Ingersoll-Ross process", and its normalization onto the simplex, the "Jacobi process" – and Benton et al. (2024) suggest "Wright–Fisher diffusion" in a toy experiment. However these models struggle from numerical instability. One solution to this instability is to essentially perform Gaussian diffusion (see App. A). Another is to build flow-matching models on the simplex (Stark et al., 2024; Tang et al., 2025; Davis et al., 2024; Eijkelboom et al., 2024). However these sacrifice the ability to straightforwardly calculate a likelihood and access to many diffusion algorithms, such as classifier guidance.

# 3 BACKGROUND AND MOTIVATION

First we describe diffusion models for discrete data and the challenges unifying the frameworks.

## 3.1 DIFFUSION MODELS FOR DISCRETE DATA

We consider modelling a distribution $p(x_0)$ over a discrete space of size $B$, and will extend to sequences of discrete objects below. Our model will begin with a distribution that is easy to sample from, $q(x_1)$, and then applies a stochastic process parametrized by $\theta$ from time 1 to 0. This produces a trajectory $q_\theta((x_t)_{t=0}^1)$ and we hope to pick $\theta$ so that $q_\theta(x_0) \sim p(x_0)$.

**Markov processes** To generate training data to fit $q_\theta((x_t)_{t=0}^1)$, we take samples $x_0 \sim p(x_0)$ and evolve it according to a Markov process to get a trajectory $p((x_t)_{t=0}^1)$. We can train $q_\theta$ on these trajectories by optimizing a negative ELBO

$$
\begin{aligned}
-\log q_\theta(x_0) &\leqslant -\mathbb{E}_{p((x_t)_{t=0}^1|x_0)} \log \frac{q_\theta((x_t)_{t=0}^1)}{p((x_t)_{t=0}^1|x_0)} \\
&= -\mathbb{E}_{p((x_t)_{t=0}^1|x_0)} \log \frac{q_\theta((x_t)_{t=0}^1|x_1)}{p((x_t)_{t=0}^1|x_0,x_1)} + \mathrm{KL}(p(x_1|x_0)|q(x_1)).
\end{aligned}
\tag{1}
$$

**The time dilation function** To make the second term of Eqn. 1 small we need $p(x_1|x_0) \approx q(x_1)$. Conveniently, applying a Markov process to $x_0$ usually leads to $p(x_t|x_0)$ converging to a stationary distribution $p(x_\infty)$ as $t \to \infty$, a good choice for $q(x_1)$. However our $t$ is on the interval $[0,1]$, so we compress $[0,\infty)$ into $[0,1]$: we pick an increasing "time dialation" function $\tau : [0,1] \to [0,\infty)$ and simulate $x_t$ so that it has had the Markov process applied to it for time time $\tau_t$. In particular, if $\tau_1$ is very large, $p(x_1|x_0) \approx p(x_\infty) = q(x_1)$. $\tau_t$ is a more convenient parametrization for our presentation than equivalent functions $\beta_t = \dot{\tau}_t, \alpha_t = \exp(-\tau_t)$ in other works (Shi et al., 2024). Picking $\tau_1$ very large, the second term of the ELBO can be made arbitrarily small, so we leave it out of the presentation below.

**Matching forward and backward flow** $q_\theta$ is usually parameterized to take $x_t, t$ and predict the $x_0$ that generated $x_t$, that is, approximate $p(x_0 \mid x_t, t)$; we represent this prediction $\tilde{x}_0 = q_\theta(x_0|x_t, t)$ as a vector of probabilities over the $B$ tokens $\sum_b \tilde{x}_{0,b} = 1$. Some rearrangement then allows one to rewrite the first term of Eqn. 1 as an expectation of a term $L$ that can be interpreted as the divergence between the "infinitesimal flow" forward $p$ and backward $q_\theta$ at $x_t$:

$$
E_{t \sim \mathrm{Unif}(0,1)} E_{p(x_t|x_0)} L(x_t, t, x_0, \tilde{x}_0).
$$

Thus getting a stochastic estimate of the ELBO has 3 steps: (1) Sample noisy $x_t$ by simulating the Markov process for time $\tau_t$, (2) Predict de-noised $x_0$ with $q_\theta(x_0 \mid x_t, t)$, and (3) Estimate the ELBO by computing the particular form of $L$.

---

[2] Interestingly, Stone (1963) also wrote discrete diffusion as the function of an underlying Gaussian diffusion. However the function from Stone (1963) was a path-dependent time-dilation rather than `argmax`.

**Moving to multiple dimensions**   To model sequences of discrete objects $x_0 = x_0^1 \cdots x_0^D$, we simply apply the forward process to each position $x_0^d$ independently. "Sample noisy $x_t$" remains the same, repeated for every $d$. The "infinitesimal flow" for each position is also independent: the "Estimate ELBO" step also remains the same, repeated for every $d$ and then summed across all $d$. Therefore, in the "Predict de-noised $x_0$" step we will predict $\tilde{x}_0^d = q_\theta(x_0^d|x_t, t)$ for each $d$.

### 3.2   Challenges comparing domains for discrete diffusion

**Comparing diffusion models**   A practitioner much choose a forward process which will determine how they train their diffusion model. For discrete diffusion, the forward process is mutation defined with a rate matrix $\mathcal{L}$; the form for $L$ was derived in Campbell et al. (2022). This gives Alg. 1, where $\vec{x}_0$ is the indicator vector for the token $x_0$, $\mathbb{D}(\lambda_1||\lambda_2) = \lambda_1 \log \frac{\lambda_1}{\lambda_2} - \lambda_1 + \lambda_2$ is the KL divergence between two Poisson distributions, $\hat{w}(\tilde{x}_0) := \sum_b \tilde{x}_{0b} \hat{w}(b)$, and $\dot{\tau}_t$ is the derivative of $\tau_t$. For Gaussian diffusion, the forward process is Brownian motion on embeddings $\mathrm{emb}(x_0) \in \mathbb{R}^r$; the form for $L$ was derived in Ho et al. (2020). This gives Alg. 2. Now, how can a practitioner compare how well each model fits its data, and how can they leverage their expert knowledge when designing their forward process? Unfortunately there is little infrastructure for answering these questions.

| **Algorithm 1** ELBO for discrete diffusion | **Algorithm 2** ELBO for Gaussian diffusion |
|---|---|
| 1: Sample $t \sim \mathrm{Unif}(0,1)$ | 1: Sample $t \sim \mathrm{Unif}(0,1)$ |
| 2: **Sample noisy $x_t$:** | 2: **Sample noisy $x_t$:** |
| 3: Sample $x_t \sim \mathrm{Categorical}(\vec{x}_0^T e^{\tau_t \mathcal{L}})$ | 3: Set $x_t = e^{-\tau_t}\mathrm{emb}(x_0) + \sqrt{1 - e^{-2\tau_t}}N(0, I)$ |
| 4: **Predict de-noised $x_0$:** | 4: **Predict de-noised $x_0$:** |
| 5: Predict $\tilde{x}_0 = q_\theta(x_0|x_t, t)$ | 5: Predict $\tilde{x}_0 = q_\theta(x_0|x_t, t)$ |
| 6: **Estimate ELBO:** | 6: **Estimate ELBO:** |
| 7: $\hat{w}(b) = (\vec{b}^T e^{\tau_t \mathcal{L}})(1/\vec{b}^T e^{\tau_t \mathcal{L}})^T$ | 7: $L = \frac{\dot{\tau}_t e^{-2\tau_t}}{(1-e^{-2\tau_t})^2}\|\mathrm{emb}(x_0) - \mathrm{emb}(\tilde{x}_0)\|^2$ |
| 8: $L = \sum_{b \neq x_t} \mathcal{L}_{b \to x_t} \dot{\tau}_t \mathbb{D}\left(\hat{w}(x_0)_{bx_t}||\hat{w}(\tilde{x}_0)_{bx_t}\right)$ | 8: |

*Likelihood comparison* We would like to compare the ELBOs $\mathbb{E}[L]$ of discrete and Gaussian diffusion, but the later are infinity due to a singularity as $t$ becomes small[3]. Practitioners must therefore choose a minimum $t_{\min}$[4]. Formally this is equivalent to estimating an ELBO for $\log p(x_{t_{\min}})$ instead of $\log p(x_0)$. However, $p(x_{t_{\min}})$ is a continuous density, fundamentally a different object than the probability of a discrete object $p(x_0)$. Paradoxically, the values of the ELBOs of the two models are often close suggesting they may nevertheless be formally comparable.

*Hyperparameter comparison* Discrete and Gaussian diffusion models are specified by hyperparameters $\mathcal{L}$ and $\mathrm{emb}$ with vastly different interpretations: a matrix whose entry $\mathcal{L}_{b_1 \to b_2}$ describes the rate at which $b_1$ mutates to $b_2$, versus an embedding function $\mathrm{emb}$ that takes the alphabet into Euclidean space $\mathbb{R}^r$ for some $r$ (we write $\mathrm{emb}(\tilde{x}_0)$ as shorthand for $\sum_b \tilde{x}_{0,b}\mathrm{emb}(b)$).

**Stability of simplex diffusion**   Below we'll also discuss simplicial diffusion which in principle combines the combines the strengths of discrete and Gaussian diffusion. In practice, it is numerically unstable and slow as **Sample noisy $x_t$** involves "sampling from Jacobi diffusion processes [which] is more expensive than commonly used SDEs", and **Estimate ELBO** involves a calculation which "at very small $t$ tends to become very large and cause numerical issues" (Avdeyev et al., 2023).

**Practical unification**   Currently, practitioners must commit to a $q_\theta(x_0 | x_t, t)$ trained on one these three modalities before training, restricting their access to downstream algorithms. Ideally they could avoid making this choice.

---

[3]To see this, note at initialization $\|\mathrm{emb}(x_0) - \mathrm{emb}(\tilde{x}_0)\|^2$ is roughly a constant, and for the classical choice $\tau_t = -\frac{1}{2}\log(1-t)$, the square error in Alg. 2 is weighted by $\frac{1}{2t^2}$, so the loss is $\gtrsim \int_0^1 t^{-2}dt = \infty$; a different choice of $\tau_t$ only acts as a change-of-variables, and therefore cannot make the loss finite.

[4]Some discrete diffusion models also have a singularity at $t \to 0^+$, requiring one to specify a $t_{\min}$ (Campbell et al., 2022; Lou et al., 2023). This is not the case for "schedule-conditioned" models, including masking, partially explaining its popularity (Amin et al., 2025; Shi et al., 2024).

# 4 UNIFYING DISCRETE AND GAUSSIAN DIFFUSION

To build the infrastructure for comparing domains for discrete diffusion, we unify discrete and Gaussian diffusion in a broader framework. Our results lead to better understanding of loss and hyperparameter comparisons. In the following section we extend our framework to simplicial diffusion.

## 4.1 UNIFICATION RESULT

Our idea is to represent each dimension of a sequence with $\zeta$ copies to get a *sequence of sequences*.

ex. for $\zeta = 4, x_0 = \text{A|C|C|T}$ is represented as $\text{AAAA|CCCC|CCCC|TTTT}$.

Then each letter in each sequence is evolved by the mutation matrix $\mathcal{L}$. When $\zeta = 1$ we get discrete diffusion and we show that as $\zeta \to \infty$ we get Gaussian diffusion. Below we discuss the case where $x_0$ is a single letter / token, which can naturally be extended to a multi-dimensional diffusion model.

$\vec{x}_t$ **on the simplex** We will ultimately arrive at a Gaussian limit in Euclidean space, but we first represent $x_t$ on the simplex. Above, $x_t$ was one of $B$ tokens; now it's one of $B^\zeta$ sequences of $B$ tokens $x_t = x_t^{(1)} \cdots x_t^{(\zeta)}$. It can be generated by sampling each $x_t^{(z)} \sim \text{Categorical}(\vec{x}_0^T e^{\tau_t \mathcal{L}})$. In App. E.1 we note that the loss and target $p(x_0 \mid x_t, t)$ *do not depend on the order* of $x_t$. Therefore we can represent $x_t$ as a normalized vector of counts $\vec{x}_{t,b} = \#\{b \text{ in } x_t\}/\zeta$. In App. E.1 we derive the ELBO, giving Alg. 3 – differences to discrete diffusion in Alg. 1 are in blue.

---

**Algorithm 3** ELBO for $\zeta$ discrete diffusion

---

1: Sample $t \sim \text{Unif}(0, 1)$
2: **Sample noisy $x_t$:**
3: Sample $\vec{x}_t \sim \text{Multinomial}(\zeta, \vec{x}_0^T e^{\tau_t \mathcal{L}})/\zeta$
4: **Predict de-noised $x_0$:**
5: Predict $\tilde{x}_0 = q_\theta(x_0 \mid \vec{x}_t, t)$
6: **Estimate ELBO:**
7: $\hat{w}(b) = (\vec{b}^T e^{\tau_t \mathcal{L}})(1/\vec{b}^T e^{\tau_t \mathcal{L}})^T$
8: $L = \sum_{b \neq b'} \mathcal{L}_{b \to b'} \dot{\tau}_t \zeta \vec{x}_{tb'} \mathbb{D}(\hat{w}(x_0)_{bb'} || \hat{w}(\tilde{x}_0)_{bb'})$

---

**Gaussian limit as $\zeta \to \infty$** The main idea of our proof below is that as $\zeta \to \infty$, trajectories converge quickly to $\vec{\pi}$, the stationary distribution of $\mathcal{L}$, and behave like Gaussians near $\vec{\pi}$ because of the central limit theorem (Fig. 2). As $\zeta \to \infty$ we will zoom further and further into the neighbourhood of $\pi$ where the diffusion occurs – we move from *diffusion on the simplex* to *diffusion in Euclidean space*. Our proof extends previous results in one-dimension (Stone, 1963), but uses more modern machinery; interestingly, we see that in the multi-dimensional case, the relevant Gaussian diffusion occurs in a subspace determined by the first eigenspace of $\mathcal{L}$. [5]

**Theorem 4.1.** *(Formal statement and proof in App. E.2) Call $0 > -\lambda_1 > -\lambda_2 > \ldots$ the eigenvalues of $\mathcal{L}$ and $P_1$ the projection onto the left eigenspace corresponding to $\lambda_1$. Without loss of generality, assume $\lambda_1 = 1$ [6]. For each $\zeta$ pick time dilation $\tau_t^\zeta = \frac{1}{2} \log(\zeta e^{-2\tau_t} - \zeta + 1)$ and rescale $\vec{x}_t^\zeta = \sqrt{\zeta - (\zeta-1)e^{2\tau_t}}(\vec{x}_t - \vec{\pi})/\sqrt{\vec{\pi}}$. Define the embedding into $\mathbb{R}^{\text{rank}(P_1)}$, $Q_1 = \mathbb{j}_1(\tilde{Q}_1 \tilde{Q}_1^T)^{-1/2} \tilde{Q}_1$ where $\tilde{Q}_1 = \text{diag}(\vec{\pi})^{-1/2} P_1 \text{diag}(\vec{\pi})^{1/2}$ and $\mathbb{j}_1$ is any isometry from $\text{Im}(\tilde{Q}_1) \to \mathbb{R}^{\text{rank}(P_1)}$.*

**When $\zeta = 1$ we get discrete diffusion**: $\tau_t^\zeta = \tau_t$ and $\vec{x}_t^\zeta$ is only linearly transformed $(\vec{x}_t - \vec{\pi})/\sqrt{\vec{\pi}}$.

**When $\zeta \to \infty$, we get Gaussian diffusion in the first eigenspace**:

- *Only the first eigenspace has signal: the component of $x_t^\zeta$ in $\text{Ker} Q_1$ becomes independent of $x_0$.*

- *The paths $(Q_1 \vec{x}_t^\zeta)_{t \in (0,1)}$ converge in distribution to paths from Gaussian diffusion with time dilation $\tau_t$ and embedding $\text{emb}(x_0) = Q_1(\vec{x}_0/\sqrt{\vec{\pi}})$.*

- *The ELBO in Alg. 3 converges to the ELBO in Alg. 2.*

---

[5]This is analogous to asymptotic methods that zoom into a point in a bounded space to get a limit in its unbounded tangent plane (ex. chapter 20 of van der Vaart (1998))

[6]This assumption is for convenience. Rescale $\mathcal{L}^{\text{new}} = \frac{1}{\lambda_1} \mathcal{L}$ and $\tau_t^{\text{new}} = \lambda_1 \tau_t$ to get the same diffusion.

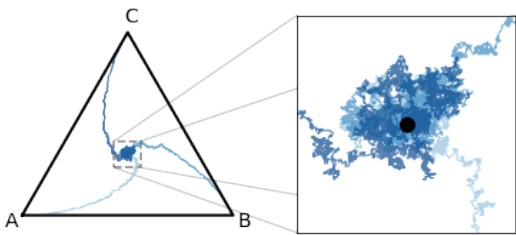

*Proof idea:* As $\zeta \to \infty$, by the law of large numbers, $\vec{x}_t$ approaches $\vec{x}_0^T e^{\tau_t \mathcal{L}}$ which itself goes to the stationary distribution of $\mathcal{L}$. We can therefore decompose

$$\vec{x}_t - \vec{\pi} = \underbrace{\vec{x}_0^T e^{\tau_t^\zeta \mathcal{L}} - \vec{\pi}}_{\text{signal}} + \underbrace{\vec{x}_t - \vec{x}_0^T e^{\tau_t^\zeta \mathcal{L}}}_{\text{noise}} .$$

Figure 2: **Discrete diffusion with a large population converges to Gaussian diffusion.** With $\zeta = 1000$, we show example trajectories $(\vec{x}_t)_t$ that converge to approximate Gaussians near $\vec{\pi}$.

The "noise" term is $\vec{x}_t - \mathbb{E}\vec{x}_t$. Since $x_t$ is an average of $\zeta$ samples, by the central limit theorem, it is approximately Gaussian with scale $\zeta^{-1/2}$ and independent of $x_0$. The "signal" term therefore is what allows us to predict $x_0$.

The only relevant behaviour is that of the slowest-decaying eigenspaces of $\mathcal{L}$: the top eigen-space represents the convergence to $\vec{\pi}$ and cancels with $-\vec{\pi}$, the next one is $P_1$ with eigenvalue $-1$, and all others vanish quickly. Therefore the signal is approximately $e^{-\tau_t^\zeta} P_1 \vec{x}_0$. This means

$$\vec{x}_t - \vec{\pi} \approx e^{-\tau_t^\zeta} P_1 \vec{x}_0 + \frac{1}{\sqrt{\zeta}} \mathcal{N}(0, \Sigma) \text{ for some } \Sigma.$$

Finally, choosing the right scaling and $\tau_t^\zeta$ gives us Gaussian diffusion. Most of the formal proof involves checking regularity conditions. $\square$

### 4.2 Application: Understanding comparisons of losses and hyperparameters

**Loss comparison** Thm. 4.1 suggests that there is virtually no difference to training a discrete diffusion model with $\zeta = 10^{100}$ and training Gaussian diffusion with Alg. 2 on a computer, suggesting their ELBOs are comparable. Yet the limiting Gaussian ELBO is infinite! Fig. 2 suggests why: paths from $\vec{x}_t$ have two phases, a nearly deterministic phase at low $t$ (Fig 2 left), and then a random phase (Fig 2 right). Diffusion models reversing these paths should therefore go through a random phase, until $p(x_0 \mid \vec{x}_t, t)$ becomes obvious, and then trace a deterministic path back to $x_0$. However, at initialization, $x_0$ is "never obvious" to the neural network $q_\theta(x_0 \mid \vec{x}_t, t)$, leading to mismatches to the deterministic paths in samples (Fig. 3 "Random"). As $\zeta$ gets larger, the paths get more deterministic, **causing the singularity in the limit**.

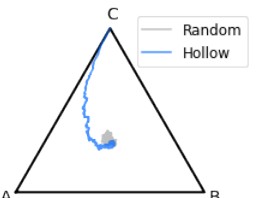

Figure 3: **The hollow parameterization leads to realistic reverse path samples.** $\zeta = 300$.

The practical solution is simple – weight the output of the neural network by the evidence for each $x_0$, $q_\theta(x_0 \mid x_t, t) \propto p(x_t \mid x_0, t) q_\theta(x_0)$ where $p(x_t \mid x_0, t)$ "automatically handles" deciding when $x_0$ is obvious (Fig. 3 "Hollow"). This was suggested in the appendix of Austin et al. (2021) as way to improve discrete diffusion models, but becomes important here as a way to build new Gaussian diffusion models with formally comparable likelihoods[7]. Amin et al. (2025) showed that in higher dimensions this becomes equivalent to using the **"hollow" predictor**[8] $q_\theta(x_0^d \mid x_t, t) \propto p(x_t^d \mid x_0^d, t) q_\theta(x_0^d \mid x_t^{-d}, t)$ where $x_t^{-d}$ is all positions except $d$. In App. E.3 we formally prove that the hollow parametrization removes the singularity of the ELBO.

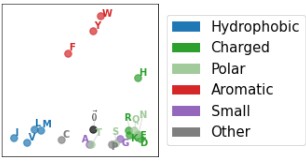

Figure 4: emb **of amino acids from BLOSUM** $\mathcal{L}$. $\text{emb}(x_0)$ from Thm. 4.1 for $\mathcal{L}$ from Amin et al. (2025).

**Hyperparameter comparison** Thm. 4.1 gives us a formula for emb determined by the slowest-decaying directions in $\mathcal{L}$. App. E.4 also shows that every emb can be induced from some $\mathcal{L}$. Remarkably, this connection accommodates embeddings in different dimensions $\mathbb{R}^r$: $r$ is determined by the dimension of the dominant eigenspace of $\mathcal{L}$. In Fig. 4 we show emb for the BLOSUM stochastic processes for amino acids, and see it clusters similar amino acids together. The practical implications are (1) one can sanity-check their designed $\mathcal{L}$ by plotting its induced embeddings, and (2) discrete diffusion offers a richer design space, as one can specify all the interacting eigenspaces of $\mathcal{L}$ rather than just the dominant one, emb.

[7]Note the hollow parametrization is specific to *discrete data* where there are only finitely many possible $x_0$.
[8]This does not require a change of architecture: the network can take in $x_t$ but must learn to disregard $x_t^d$.

## 5 UNIFYING SIMPLICIAL DIFFUSION

Now we add simplicial diffusion to our unification of discrete and Gaussian diffusion. Proving the equivalence of the forward process: we add reproduction to our population of $\zeta$ letters and simply refer to the well known result of Kimura (1955) from mathematical genetics. We also derive new results on the limit of the ELBO and explore our connection with theory of mathematical genetics; this will allow us to address the instabilities that plague simplicial diffusion models.

### 5.1 UNIFICATION RESULTS

**The Wright-Fisher model** We now allow our population of $\zeta$ to reproduce. The population is swapped with a new generation at rate $\zeta$ (that is, a new generation occurs at $\Delta\tau \sim \mathrm{Exp}(1)/\zeta$) and at each generation we create $\zeta$ "children" which pick a parent uniformly at random. Between generations, individuals also mutate according $\mathcal{L}$ (Fig. 1a). We now ask what happens when $\zeta \to \infty$.

**The limit of** $p((x_t)_t)$ Kimura (1955) was the first to derive the $\zeta \to \infty$ limit of the stochastic process. Unlike the mutation-only case which zooms in to $\vec{\pi}$, this limiting distribution has paths that travel throughout the simplex (Fig. 1b). This limit, often itself called "Wright-Fisher diffusion" is exactly the forward process in simplicial diffusion (Avdeyev et al., 2023). Details are in App. E.5.1. One biologically reasonable assumption past works make is a parent-independent mutation rate matrix, that is, $\mathcal{L} = \psi \times (\mathbb{1}\vec{\pi}^T - I)$ for stationary distribution $\vec{\pi}$ and mutation rate $\psi > 0$. This does not restrict the design space of simplicial diffusion, which is specified by an intensity parameter $\psi$ and stationary distribution $\vec{\pi}$, so we make the same assumption.

**The limit of the ELBO** We derive the limit of the discrete diffusion ELBO. Remarkably, we get an objective that matches "score functions" $\vec{s}$ like that heuristically derived in Avdeyev et al. (2023). It is also mathematically equivalent to the expression in Eqn 27 of Benton et al. (2024) with two differences: (1) we avoid taking the derivative of the neural network, and (2) their expression differs by an unknown constant from the ELBO, while ours is directly comparable with ELBOs from other models.

**Theorem 5.1.** *(Proof in App. E.5.2) As $\zeta \to \infty$, the discrete diffusion objective in Alg. 1 converges to the quantity in line 9 of Alg. 4.*

The main idea of the proof is an application of a Taylor expansion and Stirling's approximation; the main challenge is handling of behaviour at the boundaries of the simplex and regularity conditions.

### 5.2 APPLICATION: FAST AND STABLE SIMPLICIAL DIFFUSION

We have unified simplicial diffusion with discrete and Gaussian diffusion, in particular allowing likelihood comparison, which will be crucial in the following section. Our unification also immediately suggests a connection to the mathematical genetics literature. We now apply the solutions from that literature to improve simplicial diffusion models. Many of the formulas are standard but long – we save their statement and experimental validation to App. C.

---

**Algorithm 4** ELBO for simplicial diffusion. Our changes to Avdeyev et al. (2023) are coloured.

1: Sample $t \sim \mathrm{Unif}(0, 1)$
2: **Sample noisy $x_t$:**
3: Sample $m \sim A(\psi, \tau_t)$ with Alg. 5; if $\tau_t < 0.05$, use Alg. 6
4: Sample $\vec{x}_t \sim \mathrm{Dirichlet}(\psi\vec{\pi} + m\vec{x}_0)$.
5: **Predict de-noised $x_0$:**
6: Predict $\tilde{x}_0 = q_\theta(x_0 \mid x_t, t)$
7: **Estimate ELBO:**
8: Compute $\vec{s}(\vec{x}_t \mid x_0, t) = \nabla_{x_t} \log p(x_t|x_0, t)$ with Eqn. 2
9: $L = \frac{\dot{\tau}_t}{2}\|\vec{s}(\vec{x}_t \mid x_0, t) - \vec{s}(\tilde{x}_t \mid x_0, t)\|^2_{\mathrm{diag}(\vec{x}_t) - \vec{x}_t\vec{x}_t^T}$ (this is an ELBO); if $\tau_t < 0.05$, use Eqn. 3

---

**Sampling noisy $x_t$** Avdeyev et al. (2023) samples $x_t$ by costly and approximate simulation from a stochastic differential equation (SDE). Instead, the suggestively titled paper "Exact simulation of the Wright-Fisher diffusion" (Jenkins and Spanò, 2017) gives a fast exact formula for the marginals $x_t$. The algorithm samples $\vec{x}_t$ from a Dirichlet that is centred at the stationary mutation distribution $\vec{\pi}$ when $m = 0$ and becomes more concentrated around the signal $x_0$ when $m$ is larger. $m$ itself is

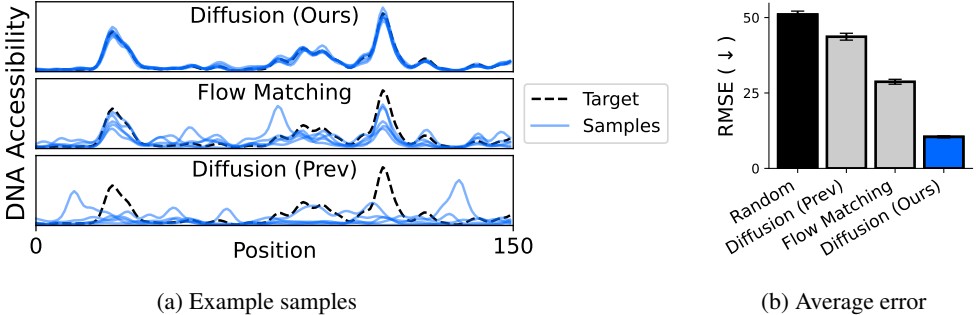

(a) Example samples

(b) Average error

Figure 5: **Improved simplicial diffusion performs accurate conditional DNA generation.** We generate DNA samples of length 500 conditioned on accessibility with a classifier. **(a)** For an example target, we plot predicted accessibility profiles at the centre 150 positions of 5 example samples from each model. We smooth profiles with a bandwidth of 2. **(b)** For 1000 targets and 10 samples from each model, we plot the error between the predicted and target profiles and its standard error.

an integer sampled from a distribution $A(\psi, \tau_t)$ that represents, going back in time $\tau_t$, how many ancestors the population descend from – it is small when $\tau_t$ is large, when everyone descended from a handful of individuals from far back in time. Benton et al. (2024) proposed the same procedure, but applied it in a toy setting.

**Computing the loss** For the loss, Avdeyev et al. (2023) derived a likelihood that involved calculating the derivative the predictor $q_\theta(x_0 \mid x_t, t)$ making it too expensive to train on. They instead suggest training a heuristically motivated loss matching the vector $\vec{s}$ to a ground truth. In Thm. 5.1 we recognize this loss as an ELBO and derive the appropriate scaling $\frac{\dot{\tau}_t}{2}$ and metric $\mathrm{diag}(\vec{x}_t) - \vec{x}_t \vec{x}_t^T$.

**Low $t$ behaviour** Both the simulation of $A(\psi, \tau_t)$ and the calculation of the gradients $\nabla_{x_t} \log p(x_t \mid x_0)$ involves an infinite series (Tavaré, 1984). Luckily the terms converge extremely fast. This is not true however at low $t$, which is the primary cause of the instability of simplicial diffusion. This instability is also well known in the genetics literature, with Griffiths (1984) emphatically stating that using the infinite series at low $t$ "produces nonsense from a computer."

The solution at low $t$ is to replace the series approximation, which gets worse with lower $t$, with a central limit approximation for $A(\psi, \tau_t)$ (Griffiths, 1984; Jenkins and Spanò, 2017) that improves with lower $t$; this is analogous to how reflected diffusion models were made stable despite their own infinite series expansion with the same problem Luo et al. (2022). We picked the $\tau_t < 0.05$ threshold as recommended by Jenkins and Spanò (2017). In App. C.3 we describe how to use this approximation to also stabilize the loss computation.

**State of the art DNA generation conditioned on a classifier** Simplicial diffusion models are state of the art tools for generating DNA conditioned on high-dimensional epigenetic properties (Avdeyev et al., 2023); however they have recently been surpassed by flow-matching models (Stark et al., 2024), which are more stable but sacrifice a closed-form ELBO and access to diffusion sampling algorithms. Given our stability improvements above, we expect to be able to generate higher quality sequences than previous methods. We fit the state of the art diffusion model (Avdeyev et al., 2023) and flow-matching model (Stark et al., 2024) to DNA data ($B = 4$) of length $D = 500$ and generate samples conditioned on achieving target "DNA accessibility profiles."

First we see our model leads to a much better fit of the data. The diffusion model from Avdeyev et al. (2023), was only able to achieve an average ELBO of 8 nats / position (12.7 before training), while a trivial model which predict uniform letters in each position achieves 1.39. In contrast, our model achieves an ELBO of 1.30. In Fig. 5 we also see our new model generates conditional samples with profiles that much better match the target. Experimental details are in App. F.

## 6 PRACTICAL UNIFIED DIFFUSION MODELS

Our results show that discrete, Gaussian and simplicial diffusion are three views of the same process. But which view should a practitioner choose for their particular downstream task? Unfortunately, there is limited theoretical infrastructure we can use to answer such a question.

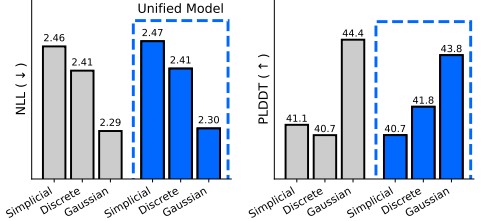 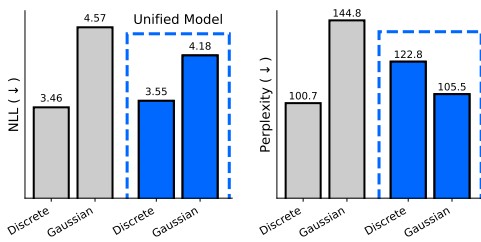

(a) Protein likelihood and sample quality      (b) Language likelihood and sample quality

Figure 7: **The sufficient statistic parametrization enables a single model to perform competitive discrete, Gaussian, and simplicial diffusion.** We compare individual models for each modality with a single unified model using the SSP. **(a)** We train on proteins and measure sample quality by predicted protein fold-ability (pLDDT). Each model was trained for the same amount of time. **(b)** We train on language and measure sample quality using the perplexity of a much larger language model. Each model was trained for 33 epochs.

Instead our theory provides a practical solution: leveraging our finding that these methods have comparable likelihoods, we show through a particular parameter choice (Fig. 6), one can train a single neural network that can perform diffusion on any domain at test time. In App. D we also show this parameterization will also allow us to make any diffusion model time-invariant, explaining and generalizing a celebrated property of masking diffusion.

## 6.1 THE SUFFICIENT STATISTIC PARAMETERIZATION (SSP)

The goal of a diffusion model is to predict $x_0^d$. To do so, one must integrate over the unseen $x_0^{-d}$ weighted by their likelihood of producing the data $x_t^{-d}$:

$$p(x_0^d \mid x_t^{-d}) = \int p(x_0^d \mid x_0^{-d})dp(x_0^{-d} \mid x_t^{-d}).$$

We can summarize this "evidence" in the normalized vector $\vec{\phi}(x_t^{d'}, t)_b \propto p(x_t^{d'} \mid t, x_0^{d'} = b)$ (Fig. 6).
Some algebra shows that $\vec{\phi}$'s are sufficient statistics, that is, they contain all relevant information about the diffusion process and $t$, leaving a regression task that invariant to both.

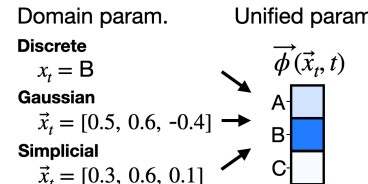

Figure 6: **The sufficient statistic parameterization represents $\vec{x}_t$ from all diffusion models in the same space.**

**Proposition 6.1.** *(Proof in App. E.7) There is a function $F^d$, **depending on $p(x_0)$ and not on the diffusion process or $t$**, such that*

$$p(x_0^d \mid x_t^{-d}, t) = F^d(\vec{\phi}(\vec{x}_t^1, t), \ldots, \vec{\phi}(\vec{x}_t^D, t)).$$

Therefore we can parametrize our neural network $q_\theta(x_0^d \mid x_t^{-d}, t) = F_\theta^d(\vec{\phi}(\vec{x}_t^1, t), \ldots, \vec{\phi}(\vec{x}_t^D, t))$ for a neural network $F_\theta^d$ that tries to learn the "universal" $F^d$.

## 6.2 APPLICATION: UNIFIED DIFFUSION MODELS

Practitioners must commit upfront to the domain their diffusion occurs. The SSP instead enables training a single neural network that can perform diffusion on any domain at test time: as long the target distribution $p(x_0)$ remains constant the optimum $F^d$ remains the same. Furthermore, we've shown above that the ELBOs of each modality are comparable, so we can train $F_\theta$ by alternating minimizing the ELBO of a different modality in each batch.

We train discrete, Gaussian, and simplicial diffusion models on proteins and compare to a single model trained using the SSP which alternates between discrete, Gaussian, and simplicial training steps. We trained our models to approach the performance of state-of-the-art protein diffusion model DPLM (Wang et al., 2024a) in likelihoods (2.36) and a "foldability" metric for samples (45.2) (Amin et al., 2025). In Fig. 7 we see that a single SSP model trained on proteins for 48 hours is competitive

in perplexity and sample quality with three single-domain models each trained for the same amount of time. We perform a similar experiment for discrete and Gaussian language models (simplicial diffusion models are challenging to scale to a large vocabulary size of $B \approx 3 \times 10^4$) and see similar results. We trained our language models with the same amount of training data as a state-of-the-art diffusion language model (Lou et al., 2023), matching its likelihood (SEDD uniform has an NLL of 3.70). Experimental details are in App. F, another downstream task is tested in App. G.1, and we also repeat these results on MNIST images in App. G.2.

## 7 CONCLUSION

Our theoretical and practical unification developed foundations that we used to improve simplicial diffusion and avoid the need to choose a specific model at train time. However, the theory suggests a number of other directions we have not yet explored.

Notable omissions from our presentation are reflected diffusion (Lou and Ermon, 2023), flow matching (Campbell et al., 2024), masking diffusion (Shi et al., 2024), and diffusion with insertions and deletions (Johnson et al., 2021). The later two can likely be easily accommodated with previous theories unifying masking and uniform diffusion on one hand (Amin et al., 2025), and substitution and insertion - deletion diffusion on the other (Johnson et al., 2021).

As well, our framework suggests new types of diffusion models "between" the three existing streams of diffusion which we only use as a lens for understanding existing models. Implementing these intermediate models may be of independent practical interest.

Finally, the SSP can be used to unify models beyond the three modalities. For instance it can be used to train models across hyperparameter settings, or optimize hyperparameters without retraining. In principle, the SSP can even be used to transfer a model to a modality it was not trained on.

## ACKNOWLEDGEMENTS

This work was supported in part by NSF CAREER IIS-2145492, NSF CDS&E- MSS 2134216, NSF HDR-2118310, BigHat Biosciences, and Capital One.

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

## A  EXTENDED RELATED WORK

We add more related work beyond those in Sec. 2.

**Classical theories unifying discrete and continuous stochastic processes**   There is a long history of deriving continuous limits of discrete processes, the "forward" processes of diffusion models. Groundbreaking work by Stone (1963) derived Gaussian diffusion as a limit to biased one-dimensional random walks. In one of the most celebrated results in mathematical genetics, Kimura (1955) also derived a continuous limit of the Wright-Fisher process with non-zero reproduction. These models were originally developed to describe the stochastic fluctuations of allele frequencies in the absence of selection, also called genetic drift. We (1) apply these results to understand and improve diffusion models, (2) also show convergence of the ELBO of diffusion models, and, to our knowledge, (3) derive a new result – the multi-dimensional Gaussian-diffusion limit of Wright-Fisher with zero reproductions – demonstrating previously un-characterized behaviour dependent on the eigenspace of the mutation operator. Results (2) and (3) are what allow us to compare likelihoods and hyperparameters.

**Parameterizations of discrete diffusion models**   In diffusion, one uses a neural network to "denoise" sequences; we call the choice of inputs and outputs of these neural networks the "**parameterization**". A number of works suggest superficially distinct, but ultimately equivalent parameterizations (Campbell et al., 2022; Lou et al., 2023). Austin et al. (2021) however suggested a distinct parametrization for discrete diffusion models by scaling the output of the neural network to "automatically" incorporate the information about the noised token about that particular location; we call this choice the "hollow" parametrization for reasons discussed below. Zheng et al. (2024), Ou et al. (2024), and Sahoo et al. (2024) suggested that masking diffusion enables a special choice of "time-invariant" parametrization; in App. D our theory shows on the contrary that every diffusion model can be made time-invariant.

**Gaussian diffusion which appears as simplicial diffusion**   Han et al. (2022); Mahabadi et al. (2023); Shabalin et al. (2025), and Floto et al. (2023) suggest a stable diffusion model on the simplex by applying softmax to Gaussian diffusion and using Itô's theorem. This parameterization is stable because forward and backward diffusion can occur as Gaussian diffusion in the logit-space. Lou and Ermon (2023) has a similar idea, swapping the softmax for an asymmetric transformation and Gaussian diffusion with reflected Gaussian diffusion. With these simplifications however, the process is exactly (reflected) Gaussian diffusion except the input to the neural network is transformed onto a simplex; in particular, it doesn't interact with the topology of the simplex. In other words, this implements simplicial diffusion in the parametrization of the neural network, but not in the sampling or loss computation.

**Another unification theory and a connection to evolution**   Li et al. (2025) looked at Gaussian diffusion with a generalized noising strategy; they noted a special case resembled masking diffusion. However the training procedure and ELBO of this special case are distinct from standard masking diffusion (Shi et al., 2024). Zhang et al. (2024) connect diffusion with the evolution process to suggest an optimization algorithm, but do not formally establish connections with biological evolution. In contrast, our work makes this connection explicit.

## B  MATHEMATICAL ERROR IN SAHOO ET AL. (2025)

In Theorem 3.1, Sahoo et al. (2025) shows that the ELBO of a discrete diffusion model is always tighter than that of a Gaussian diffusion model. In its proof, with $w_t$ from Gaussian diffusion, $z_t = \text{argmax}(w_t)$, and $x = z_0 = w_0$, they state "Since the transition $z_t \rightarrow z_s$ is Markov, we get: $q(z_s \mid w_t, z_t, x) = q(z_s \mid z_t, x)$". Putting aside the correctness of this statement, it is clear that the proof as stated requires the Markov property of $(z_t)_t$.

The way the Markov property is shown is as follows. They first define a discrete diffusion model, let's call this $(\tilde{z}_t)_t$, such that $\tilde{z}_0$ comes from the data distribution and $\tilde{z}$ evolves with respect to a uniform forward process with rate parameter $\beta(t)$ chosen such that the marginals match $p(z_t|z_0) = p(\tilde{z}_t|\tilde{z}_0)$. In Eqn. 29 they compute $\frac{d}{dt}p(z_t|z_0)$ and in Eqn. 32 they compute $\frac{d}{dt}p(\tilde{z}_t|\tilde{z}_0)$ for all starting points and show they are identical. After noting the equivalence of equations 29 and 32, they state "This pmf and the ODE are the unique signatures of a Uniform-state discrete diffusion process (Lou et al.,

2023; Schiff et al., 2025)." and from this conclude that the path distributions of $(\tilde{z}_t)_t$ and $(z_t)_t$ are equivalent, and in particular, that $(z_t)_t$ is Markov[9].

However, despite a similar result for Markov chains (two Markov processes with identical semi-groups are equivalent), $p(z_t|z_0) = p(\tilde{z}_t|\tilde{z}_0)$ and $\frac{d}{dt}p(z_t|z_0) = \frac{d}{dt}p(\tilde{z}_t|\tilde{z}_0)$ for all starting points is not enough to conclude the identity of the path distributions $p((z_t)_t|z_0) = p((\tilde{z}_t)_t|\tilde{z}_0)$. First note that $\frac{d}{dt}p(z_t|z_0) = \frac{d}{dt}p(\tilde{z}_t|\tilde{z}_0)$ is not an independent condition: it follows from $p(z_t|z_0) = p(\tilde{z}_t|\tilde{z}_0)$. Next consider this counter example:

- $\tilde{z}_0 = 1$ and $(\tilde{z}_t)_t$ evolves by switching sign with rate 1. Therefore $p(\tilde{z}_t = 0) = 1 - \frac{1}{2}e^{-2t}$.

- $z_0 = 1$ and $(z_t)_t$ has a 50% chance to stay at 0 forever and a 50% chance to swap sign at time $-\frac{1}{2}\log U$ for a $U \sim$ Uniform and never again. Therefore $p(\tilde{z}_t = 1) = \frac{1}{2}(1 + p(-\frac{1}{2}\log \text{Uniform} > t)) = 1 - \frac{1}{2}e^{-2t}$.

- When $z_0 = -1$ or $\tilde{z}_0 = -1$, then swap signs.

We have $p(z_t|z_0) = p(\tilde{z}_t|z_0)$ for all $z_0$ and therefore $\frac{d}{dt}p(z_t|z_0) = \frac{d}{dt}p(\tilde{z}_t|z_0)$ but clearly $p((z_t)_t) \neq p((\tilde{z}_t)_t)$.

Simple computer simulations indeed show that $p((\text{argmax}(w_t))_t)$ and $p((\tilde{z}_t)_t)$ are different. We show this in Fig. 8. Indeed a statistical test applied to these simulations shows $p((\text{argmax}(w_t))_t) \neq p((\tilde{z}_t)_t)$: a Mann-Whitney test shows that the paths of the argmax of Gaussian diffusion have more transitions that those of discrete diffusion with $p < 10^{-300}$.

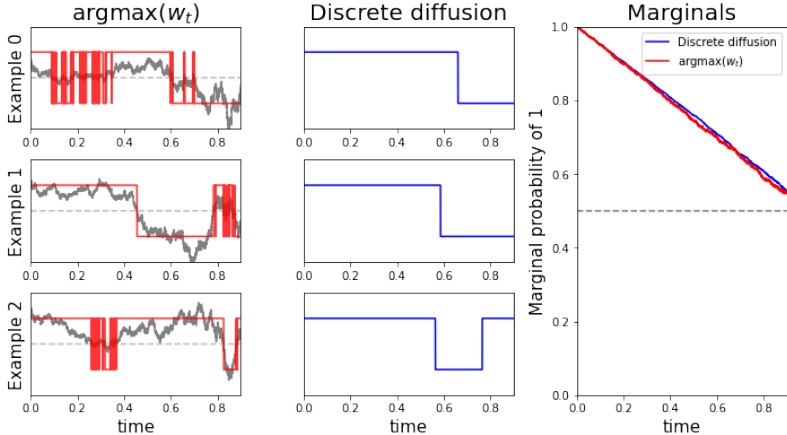

Figure 8: **The argmax of Gaussian diffusion appears different from discrete diffusion in simulation, despite having the same marginals.** We compare example paths of $p((\text{argmax}(w_t))_t)$ (left, red; we show Gaussian diffusion $w_t$ in grey), $p((\tilde{z}_t)_t)$ for uniform discrete diffusion (centre, blue), and their empirical marginals over 10'000 simulations (right); we simulate using a grid size of 0.0001. Note the two processes have the same marginals but their paths appear different; in particular, whenever $w_t$ is near 0, $(\text{argmax}(w_t))_t$ undergoes a very large number of transitions in a small time[10].

## C   WRIGHT-FISHER SAMPLING AND SCORE CALCULATIONS

Here we discuss the details of the methods in Sec. 5. Note, just like App. E.1, we can deal with $\vec{x}_t$ rather than the actual sequences $x_t$. In App. C.1 we discuss details about our algorithms, in particular how to sample form $A(\psi, \tau_t)$ and calculate the functions $\vec{s}(\vec{v} \mid x_0)$. In App. C.2 we discuss the computational complexity and stability of these algorithms in theory and in experiment. In App. C.3

---

[9]This interpretation of the text was confirmed in personal communication with the first author of Sahoo et al. (2025)

[10]Indeed, noting the self-similarity of Brownian motion, one can show that, conditioned on $w_t = 0$, with probability 1 $(z_t)_t$ makes infinitely many transitions in the interval $[t, t + \epsilon]$ for any $\epsilon > 0$. The probability of infinitely many transitions in a bounded interval for discrete diffusion however is 0.

we discuss how we sample and calculate the ELBO at low $t$. Finally, in App. C.4 we discuss our condiitonal sampling procedure.

We also note two more differences between our method and that of Avdeyev et al. (2023):

- Our neural network directly predicts $\vec{x}_0$ rather than the "score" $\vec{s}$.

- As described in App. E.5.1, we use the natural permutation-symmetric "multi-allelic" extension to the 1D SDE when $B > 2$, while they use a stick-breaking procedure.

- We use high-precision operations to calculate large alternating series accurately, as described in App. C.2.

## C.1 ALGORITHM DETAILS

**Sample noisy $x_t$** We've discussed the algorithm from Jenkins and Spanò (2017) in the main text. We now present their algorithm for sampling from $A(\psi, \tau_t)$.

---

**Algorithm 5** Exact sampling from ancestral process $A(\psi, \tau_t)$

---

1: Define coefficients: $c_{km}^{\psi} = \frac{(2k+\psi-1)(\psi+m)_{(k-1)}}{m!(k-m)!} e^{-k(k+\psi-1)\tau_t/2}$ for $k \geqslant m$
2: Sample $U \sim \text{Uniform}[0,1]$
3: Initialize $M \leftarrow 0$
4: Initialize an empty vector $\vec{k} = ()$
5: **while** True **do**
6:      Find $k_M \geqslant M$ such that $c_{(k_M+1)M}^{\psi} < c_{k_M M}^{\psi}$
7:      Make $k_M$ even: $k_M \leftarrow 2\lceil k_M/2 \rceil$
8:      Update lower bound: $S^- \leftarrow S^- + \sum_{k=M}^{k_M+1}(-1)^{k-M}c_{kM}^{\psi}$
9:      Update upper bound: $S^+ \leftarrow S^+ + \sum_{k=M}^{k_M}(-1)^{k-M}c_{kM}^{\psi}$
10:      Update $\vec{k} = (k_0, \ldots, k_{M-1}, k_M)$
11:      **while** $S^- < U < S^+$ **do**
12:          Update lower bound: $S^- \leftarrow S^- + \sum_{m=0}^{M}(c_{(k_m+2)m}^{\psi} - c_{(k_m+3)m}^{\psi})$
13:          Update upper bound: $S^+ \leftarrow S^- + \sum_{m=0}^{M}(-c_{(k_m+1)m}^{\psi} + c_{(k_m+2)m}^{\psi})$
14:          Update $\vec{k} = \vec{k} + (2, \ldots, 2)$
15:      **end while**
16:      **if** $S^- > U$ **then**
17:          **return** $m = M$
18:      **else if** $S^+ < U$ **then**
19:          $M \leftarrow M + 1$
20:      **end if**
21: **end while**

---

**Compute loss** We present a formula for $\vec{s}(\vec{v} \mid x_0, t) = \nabla \log p(\vec{x}_t \mid x_0, t)|_{\vec{v}}$ to enable computation of the loss. Avdeyev et al. (2023) computed these scores using a previously determined result with $B = 2$ then generalizing to higher dimensions with their stick-breaking procedure and a change of variables. We are instead able to derive it directly from first principles.

There are two infinite series which will be important,

$$G_\psi(\tau, x_0, \vec{x}_t) = 1 + \sum_{k=1}^{\infty}(-1)^k a_k^{\psi}(\tau, \pi_{x_0}, \vec{x}_{t,x_0})$$

$$F_\psi(\tau, x_0, \vec{x}_t) = 1 + \sum_{k=1}^{\infty}(-1)^k b_k^{\psi}(\tau, \pi_{x_0}, \vec{x}_{t,x_0})$$

where

$$a_k^{\psi}(\tau, \pi_{x_0}, \vec{x}_{t,x_0}) = e^{-\frac{k(k+\psi-1)\tau}{2}} \frac{(2k+\psi-1)(\psi)_{(k-1)}}{k!} {}_2F_1(-k, \psi+k-1; \psi\pi_{x_0}; \vec{x}_{t,x_0})$$

$$b_k^{\psi}(\tau, \pi_{x_0}, \vec{x}_{t,x_0}) = e^{-\frac{k(k+\psi+1)\tau}{2}} \frac{(\psi)_{(k)}}{k!} \frac{(2k+\psi+1)(\psi+k)}{(\psi+1)\psi} {}_2F_1(-k, \psi+k+1; \psi\pi_{x_0}+1; \vec{x}_{t,x_0})$$

where $_2F_1$ is the hypergeometric function. Although these look complicated, in practice, most terms in the numerators and denominator of $a$ and $b$ nearly cancel to 1, and, when $t$ is not too small, $e^{-k(k+\psi+1)\tau/2}$ decays extremely quickly.

Using the results in Tavaré (1984) we compute $\vec{s}(\vec{v} \mid x_0)$ in terms of these series. Since we're only interested in differences for calculating the ELBO, $\vec{s}(\vec{v} \mid x_0, t) - \vec{s}(\vec{v} \mid \tilde{x}_0, t)$ we ignore constants not depending on $x_0$.

**Proposition C.1.** *(Proof in App. E.6)*

$$p(\vec{x}_t \mid x_0, t) = \text{Dirichlet}(\pi\psi)(\vec{x}_t)G_\psi(\tau_t, x_0, \vec{v}).$$

*For $\vec{c}(\vec{v}) = \nabla \log \text{Dirichlet}(\pi\psi)(\vec{x}_t) = (\psi\vec{\pi} - \mathbb{1})/\vec{x}_t$ which does not depend on $x_0$,*

$$\vec{s}(\vec{v} \mid x_0, t) = \vec{c}(\vec{v}) + \vec{x}_0 w(x_0, \vec{v}) \tag{2}$$

*where*

$$w(x_0, \vec{v}) = \frac{e^{-\psi\tau_t/2}(\psi+1)}{\pi(x_0)} \frac{F_\psi(\tau_t, x_0, \vec{v})}{G_\psi(\tau_t, x_0, \vec{v})}.$$

With the hollow parameterization, calling $\vec{w}_b = w(b)$, we get

$$\vec{s}(\vec{v} \mid \tilde{x}_0, t)_b = \vec{c}(\vec{v})_b + \frac{e^{-\psi\tau_t/2}(\psi+1)}{\pi(x_0)} \frac{\tilde{x}_{0,b}F_\psi(\tau_t, b, \vec{v})}{\sum_{b'} \tilde{x}_{0,b}G_\psi(\tau_t, b', \vec{v})}.$$

## C.2 Computational complexity and stability

**Numerical stability** Sampling from $A(\psi, \tau_t)$ and calculating $G_\psi$ and $F_\psi$ involve alternating series of many terms which vary by many orders of magnitude, and cancel out leaving very small residuals – known as "catastrophic cancellation". To calculate these accurately, we may need higher precision than provided by `float64`; we perform any high precision calculations using the `mpmath` library Johansson and Others (2010). Avdeyev et al. (2023) did not use high precision in their calculations, potentially introducing errors and instability to their loss computation.

We perform all calculations at `float64` to take advantage of parallel GPU computations, estimate the error of each computation using a *condition number* and recompute just those terms with condition number above a threshold in `mpmath` on a CPU. In practice, we only need to perform calculations at high precision for small $t$, before we switch to the "low time regimen" $\tau_t < 0.05$ where we switch tot he Griffiths approximation.

The condition number of a series $a_1 + a_2 + \cdots + a_M$ is defined as $\eta = \sum_m |a_m|/|\sum_m a_m|$; one can estimate the error of their summation at finite precision by

$$\text{error} \approx \eta \times \text{precision}.$$

When there is catastrophic cancellation, the denominator in the definition of $\eta$ will be very large, representing that error might be high. To estimate $\eta$, we keep track of $\tilde{\sum}_m |a_m|$ ($\tilde{\sum}$ representing our finite-precision summation) and estimate $\eta \approx \tilde{\sum}_m |a_m|/|\tilde{\sum}_m a_m|$. If $\eta >$ desired error/float 64 precision $= 10^{-6} \times 2^{52}$, then we recompute at higher precision.

**Sampling** The complexity for sampling $\vec{x}_t$ involves (1) $O(m)$ for sampling $m$, which is $O(1/\tau_t)$ in expectation (see App. C.3), and (2) $O(B)$ time for sampling $\vec{x}_t$ from a Dirichlet. Crucially, the complex calculations involving an infinite series occur in (1) and are independent of the alphabet size $B$. Comparatively, sampling from the SDE requires $O(BT/\Delta t)$ compute. A higher $\Delta t$ will decrease compute but lead to lower-fidelity samples, especially at low $t$ where even small fluctuations in $x_t$ can lead to instability. We also parallelize the computations in Alg. 5 to benefit from GPU acceleration. The result is that, except when we must switch to high precision, our sampling procedure is much faster than that using an SDE (Fig. 9a), and much more stable at low $\tau$ (Fig. 9b).

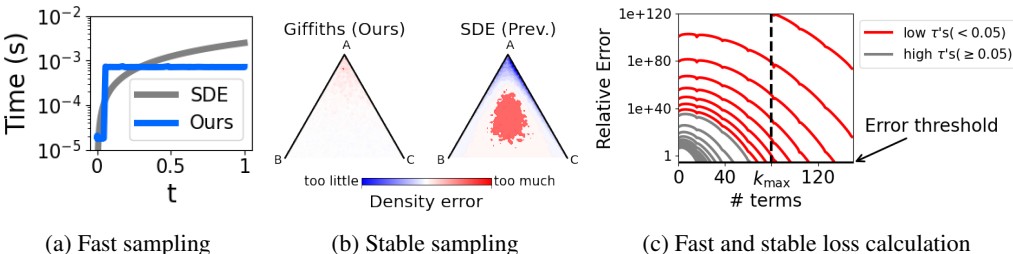

(a) Fast sampling        (b) Stable sampling        (c) Fast and stable loss calculation

Figure 9: **Leveraging mathematical genetics literature, we build fast and stable simplicial diffusion.** (a) We plot the time it takes to sample a sequence of $D = 500$ using an SDE, versus our exact sampling for various values of $t$ on an A100 80GB GPU. We threshold switching to the Griffiths approximation at $\tau_t = 0.1$. (b) For $\tau = 0.1$ and $B = 3$ we sample $3 \times 10^7$ points from the exact sampling method, Griffith's approximation, and using an SDE with 25 steps as used in Avdeyev et al. (2023). We then perform density estimates of these data and plot the error to the exact samples. We plot a $\times 6$ zoom into the vertex $A$. We see the SDE struggles to sample near the corner. We use $\psi = 3, \vec{\pi} = [0.25, 0.4, 0.35]$ (c) We plot the accuracy of our approximation of the infinite series $G_\psi(\tau, x_0, \vec{x}_t)$ including different numbers of terms for various values of $\tau \in [0, 0.2]$. We choose $\psi = B = 4$ and $\vec{\pi}$ uniform, and plot the relative error for two values of $\vec{x}_{t,x_0}$. We only use the series approximation for $\tau \geqslant 0.05$ (grey), which allows us to only use 80 terms. Meanwhile (Avdeyev et al., 2023) used 1000 terms to accommodate smaller $\tau$ (red). Our error threshold is $10^{-6}$.

**Loss computation** The complexity for computing the loss involves (1) $O(Bk_{\max})$ computations for the series $F_\psi$ and $G_\psi$, and (2) $O(B)$ computations for computing the loss given the vectors $\vec{s}$. Crucially, the complex calculations involving an infinite series occur in (1) and can be parallelized across $B$ allowing massive acceleration on GPU. $k_{\max}$ should become very large as $t$ becomes small, leading (Avdeyev et al., 2023) to choose $k_{\max} = 1000$. Instead we only use the series computation for $\tau \geqslant 0.05$, allowing us to use $k_{\max} = 80$ (Fig. 9c), and compute a $O(B)$ ELBO for $\tau < 0.05$ in App. C.3.

## C.3 LOW TIME REGIMEN

When $t$ is small, sampling from $A(\psi, \tau_t)$ or calculating $G_\psi, F_\psi$ become unstable and can require unbounded compute. Griffiths (1984) suggested a Gaussian approximation for $A(\psi, \tau_t)$ which we will also use for deriving stable approximations of $\vec{s}(\vec{v} \mid x_0, t)$ which require bounded compute.

**Sample noisy $x_t$** We copy the following from Jenkins and Spanò (2017). Note compute does not scale with $\tau_t$.

---

**Algorithm 6** Sampling from ancestral process $A(\psi, \tau_t)$ - Low $t$ approximation

---

1: Set $\beta \leftarrow \frac{1}{2}(\psi - 1)\tau_t$
2: **if** $\beta \neq 0$ **then**
3:      Set $\eta \leftarrow \beta/(e^\beta - 1)$
4:      Set $\mu \leftarrow \frac{2\eta}{\tau_t}$
5:      Set $\sigma^2 \leftarrow \frac{2\eta}{\tau_t} (\eta + \beta)^2 \left(1 + \frac{\eta}{\eta+\beta} - 2\eta\right) \beta^{-2}$
6: **else**
7:      Set $\mu \leftarrow \frac{2}{\tau_t}$
8:      Set $\sigma^2 \leftarrow \frac{2}{3\tau_t}$
9: **end if**
10: Sample $Z \sim \mathcal{N}(\mu, \sigma^2)$
11: **return** $m = \max(0, \lfloor Z + 0.5 \rfloor)$      ▷ Round to nearest non-negative integer

---

**Compute loss** The loss in this regimen, even with the Griffiths approximation, becomes intractable; instead we use the Griffiths approximation to simply bound the loss.

When $t$ is small, $x_0$ is almost always $b^* = \operatorname{argmax}_b \vec{x}_{t,b}$. We therefore set $\tilde{x}_0 = \delta_{b*}$. In practice, in our protein setting, we only see $b^* \neq \operatorname{argmax}_b \vec{x}_{t,b}$ with $\tau_t < 0.05$ at a rate of less than 1 in $7 \times 10^7$. Since $x_0 \neq b^*$ is so rare we only aim to find a loose bound. Calling $\vec{v} = \vec{x}_t$ we bound the loss by

$$L \leqslant \frac{\dot{\tau}_t}{2} \left( \|\vec{s}(\vec{v} \mid x_0, t) - \vec{c}(\vec{v})\|_{\operatorname{Diag}(\vec{v}) - \vec{v}\vec{v}^T} + \|\vec{s}(\vec{v} \mid b^*, t) - \vec{c}(\vec{v})\|_{\operatorname{Diag}(\vec{v}) - \vec{v}\vec{v}^T} \right)^2$$

$$= \frac{\dot{\tau}_t}{2} \left( w(x_0, \vec{v})\sqrt{\vec{v}_{x_0}} + w(b^*, \vec{v})\sqrt{\vec{v}_{b*}} \right)^2.$$

In the next proposition we give an alternate formula for $w(x_0, \vec{v})$ which will allow us to Griffith's approximation and a saddle point approximation to estimate $w(b^*, \vec{v})$. It will also allow us to bound $w(x_0, \vec{v})$. To our knowledge, this strategy is original.

**Proposition C.2.**

$$w(x_0, \vec{v}) = \vec{v}_{x_0}^{-1} \tilde{\mathbb{E}}_{\vec{v}_{x_0}, \vec{\pi}_{x_0}} m_t$$

where $\tilde{\mathbb{E}}_{\vec{v}_{x_0}, \vec{\pi}_{x_0}}$ is over the weighted, normalized distribution $p(A(\psi, \tau_t) = m_t) \frac{(\psi)_{(m_t)}}{(\psi \pi_{x_0})_{(m_t)}} \vec{v}_{x_0}^{m_t}$.

*Proof.* Inspection of first expression of the proof of Prop. C.1. $\qquad \square$

We can now bound $w(x_0, \vec{v}) \vee w(b^*, \vec{v}) \leqslant \vec{v}_{x_0}^{-1} \tilde{\mathbb{E}}_{1,p} m_t$ where $p = \vec{\pi}_{x_0} \wedge \vec{\pi}_{b*}$. Therefore, we get

$$L \leqslant 2\dot{\tau}_t \vec{v}_{x_0}^{-1} (\tilde{\mathbb{E}}_{1,p} m_t)^2. \tag{3}$$

Now we only need to calculate $\tilde{\mathbb{E}}_{1,p} m_t$; to do so we apply a saddle point approximation to Griffith's approximation to get Eqn. 4 below. Note compute does not scale with $\tau_t$.

**Saddle point approximation**     Let's take the Griffiths approx as $t$ becomes small, so $w_t \sim N(\mu, \sigma)$ where $\mu, \sigma$ are form Alg. 6. We can use a Stirling approximation to get

$$\frac{(\psi)_{(m_t)}}{(\psi p)_{(m_t)}} \propto (1 + O(1/m)) \left( 1 + \frac{(1-p)\psi}{\psi p + m_t - 1} \right)^{(\psi p + m_t - 1) + 1/2} (\psi + m_t - 1)^{(1-p)\psi}$$

$$= (1 + O(1/m))(\psi + m_t - 1)^{(1-p)\psi}.$$

We take a saddle point approximation of $\tilde{\mathbb{E}}_{1,p} m_t$, i.e. take its value as the maximizer of the approximate log likelihood

$$C - \frac{1}{2\sigma^2}(m_t - \mu)^2 + (1-p)\psi \log(\psi + m_t - 1) + O(1/m_t).$$

We therefore get the approximation

$$\tilde{\mathbb{E}}_{1,p} m_t \approx \left( (\mu - (\psi - 1)) + \sqrt{(\mu + (\psi - 1))^2 + 4(1-p)\psi\sigma^2} \right)/2. \tag{4}$$

Noting $m_t \sim \tau_t^{-1}$, this approximation has relative error roughly $O(\tau_t^2)$. And as $\tau_t \to 0$, $\mu \sim \tau_t^{-1}$ and $\sigma \sim \tau^{-1/2}$ so

$$\tilde{\mathbb{E}}_{1,p} m_t \approx \mu \approx 2/\tau_t.$$

## C.4    TIME REVERSAL SDE

Reversing the SDE Eqn. 5 using the result of Anderson (1982), we get

$$d\vec{z}_\tau = \left( \frac{\psi}{2}(\vec{\pi} - \vec{z}_\tau) - B(\mathbb{1}/B - \vec{z}_\tau) - \left( \operatorname{diag}(\vec{z}_\tau) - \vec{z}_\tau \vec{z}_\tau^T \right) \nabla \log p(\vec{x}_t \mid t) \right) d\tau$$

$$+ \operatorname{diag}\left( \sqrt{\vec{z}_\tau} \right) \left( I - \sqrt{\vec{z}_\tau}\sqrt{\vec{z}_\tau}^T \right) d\vec{W}_\tau$$

where $\vec{z}_{\tau_t} = \vec{x}_t$. $\vec{s}(\vec{x}_t \mid \tilde{x}_0, t)$ approximates $\mathbb{E}_{x_0|\vec{x}_t} \log p(\vec{x}_t \mid x_0, t) = \nabla \log p(\vec{x}_t \mid t)$, meaning we can substitute it into the place of $\nabla \log p(\vec{x}_t \mid t)$. We sample by discretizing this SDE and sampling backwards.

To perform classifier guidance conditioning on a variable $y$, we can add $\nabla \log p(y \mid x_t)$ to $\vec{s}(\vec{x}_t \mid \tilde{x}_0, t)$. In practice, we perform the classic one-step approximation $\nabla \log p(y \mid x_t) \approx \nabla \log \mathbb{E}_{x_0 \sim q_\theta(x_0|x_t, t)} p(y \mid x_0)$. If we have a classifier $f(x_0) = p(y \mid x_0)$ then we approximate $E_{x_0 \sim q_\theta(x_0|x_t, t)} p(y \mid x_0)$ using the "one-shot" prediction $f(\tilde{x}_0)$.

## D   TIME-INVARIANT DISCRETE DIFFUSION MODELS

Zheng et al. (2024), Ou et al. (2024), and Sahoo et al. (2024) noted that for masking diffusion, it is not necessary to pass $t$ to the neural network – it has "time-invariant" parametrization. Zheng et al. (2024) suggests this makes masking models a fundamentally different object than other diffusion models: "we reveal that both training and sampling of [masked models] are theoretically free from the time variable, arguably the key signature of diffusion models, and are instead equivalent to masked models." Our sufficient-statistic parameterization shows on the contrary that every diffusion model can be made time-invariant by a choice of parameterization, with masking as a special case.

Does this suggest that every diffusion may perform as well as masking diffusion after this choice of parameterization? Amin et al. (2025) suggests that masking performs well not because of its choice of parameterization, but because of "schedule conditioning".

**Time-invariance is a function of parameterization:** Masking is time-invariant due to a choice of parametrization. To see this, imagine applying a time-dependent rotation to each $x_t^d$; we are essentially performing the same diffusion but now must also pass $t$ to $q_\theta$ so it can "undo" the transformation. The $\vec{\phi}$ can be thought of as automatically transforming $x_t$ so $F^d$ is independent of time in any diffusion model.

**Masking uses SSP:** Indeed the SSP of masking diffusion, $\vec{\phi}(x_t^d, t) = \delta_{x_t}$ if $x_t \neq \text{mask}$ and $\vec{\phi}(x_t^d, t) = [\frac{1}{B}, \ldots, \frac{1}{B}]$ otherwise, is exactly the canonical parametrization. Thus the time-invariance of masking isn't special – rather masking's most convenient parametrization happens to be the SSP.

## E   THEORETICAL RESULTS

### E.1   MUTATION POPULATION DISCRETE DIFFUSION LOSS

In this appendix we derive Alg. 3 by showing it is equivalent to Alg. 1. Namely, we assume $D = 1$ and $x_t$ is a sequence of length $\zeta$ and show

- **Predict de-noised** $x_0$**:** the target of $q_\theta(x_0 \mid x_t, t)$, $p(x_0 \mid x_t, t)$, only depends on the vectorized $\vec{x}_t$.

- **Compute loss:** $L = \sum_{x' \neq x_t} \mathcal{L}_{x' \to x_t} \dot{\tau}_t \mathbb{D}\left( \frac{p(x'|x_0,t)}{p(x_t|x_0,t)} \middle\| \frac{p(x'|\tilde{x}_0,t)}{p(x_t|\tilde{x}_0,t)} \right)$ is equivalent to the form in Alg 3.

Given prediction and loss computation only depend on $\vec{x}_t$, we can also replace sampling $x_t$ with just sampling $\vec{x}_t \sim \text{Mult}(\zeta, \vec{x}_0^T e^{\tau_t \mathcal{L}})/\zeta$, giving Alg. 3.

**Predict de-noised** $x_0$    Simply note

$$p(x_0 \mid x_t, t) \propto p(x_0) p(x_t \mid x_0, t)$$

$$= p(x_0) \prod_{z=0}^{\zeta} (\vec{x}_0^T e^{\tau_t \mathcal{L}})_{x_t^{(z)}}$$

$$= p(x_0) \prod_{b=1}^{B} (\vec{x}_0^T e^{\tau_t \mathcal{L}})_b^{\zeta \vec{x}_{t,b}}.$$

**Compute loss** For sequences $x \neq x$ of length $\zeta$ which differ in exactly one position, say $x^{(z)} = b \neq b' = x'^{(z)}$, then $\mathcal{L}_{x \to x'} = \mathcal{L}_{b \to b'}$ and for every $x_0$

$$\frac{p(x' \mid x_0, t)}{p(x \mid x_0, t)} = \frac{\vec{x}_0 e^{\tau_t \mathcal{L}} \vec{b'}}{\vec{x}_0 e^{\tau_t \mathcal{L}} \vec{b}}.$$

If $x, x'$ differ in more than one position, then $\mathcal{L}_{x \to x'} = 0$. Call $x_t^{[z,b]}$ a sequence which has all the same letters as $x_t$ except has $b$ in position $z$. Then calling $\vec{p} = \vec{x}_0^T e^{\tau_t \mathcal{L}}$ and $\vec{q} = \tilde{x}_0^T e^{\tau_t \mathcal{L}}$,

$$
\begin{aligned}
L &= \sum_{x' \neq x_t} \mathcal{L}_{x' \to x_t} \dot{\tau}_t \mathbb{D} \left( \frac{p(x' \mid x_0, t)}{p(x_t \mid x_0, t)} \middle\| \frac{p(x' \mid \tilde{x}_0, t)}{p(x_t \mid \tilde{x}_0, t)} \right) \\
&= \sum_{z=0}^{\zeta} \sum_{b' \neq x_t^{(z)}} \mathcal{L}_{b' \to x_t^{(z)}} \dot{\tau}_t \mathbb{D} \left( \frac{\vec{p}_{b'}}{\vec{p}_{x_t^{(z)}}} \middle\| \frac{\vec{q}_{b'}}{\vec{q}_{x_t^{(z)}}} \right) \\
&= \sum_b \#\{z \mid x_t^{(z)} = b\} \sum_{b' \neq b} \mathcal{L}_{b' \to b} \dot{\tau}_t \mathbb{D} \left( \frac{\vec{p}_{b'}}{\vec{p}_b} \middle\| \frac{\vec{q}_{b'}}{\vec{q}_b} \right) \\
&= \sum_{b' \neq b} \mathcal{L}_{b' \to b} \dot{\tau}_t \zeta \vec{x}_{t,b} \mathbb{D} \left( \frac{\vec{p}_{b'}}{\vec{p}_b} \middle\| \frac{\vec{q}_{b'}}{\vec{q}_b} \right).
\end{aligned}
$$

## E.2 PROOF OF GAUSSIAN CONVERGENCE

Our formal statement of the theorem adds some mild positivity assumptions for $\tau$, $\pi$ and $P_1$ which are satisfied by any reasonable choice of $\tau$ and almost every choice of $\mathcal{L}$. It is also more specific about the limiting behaviour of $\vec{x}_t^\zeta$ in non-dominant eigenspaces: we also limit to Gaussian diffusion, but with meaningless embeddings sampled from random Gaussian vectors independent of $x_0$.

Let us interpret the embedding $Q_1$. In the case that $\mathcal{L}$ is doubly stochastic, or reversible, $\pi = [\frac{1}{B}, \ldots, \frac{1}{B}]$ and $\mathcal{L}$ is symmetric; in this case $Q_1 = j_1 P_1$ is just the orthogonal projection onto the dominant eigenspace. In the more general case that $\mathcal{L}$ satisfies detailed balance, $(\text{diag}(\pi)^{1/2} \mathcal{L} \text{diag}(\pi)^{-1/2})_{ij} = \sqrt{\frac{\pi_i}{\pi_j}} \mathcal{L}_{ij}$ is symmetric so $\tilde{Q}_1$ is the orthogonal projection onto the dominant eigenspace of the "symmetrized" generator.

In more general cases, we don't get a symmetrized operator or an orthogonal projection $\tilde{Q}_1$, so we must "correct" for this with the adjustment $(\tilde{Q}_1 \tilde{Q}_1^T)^{-1/2} \tilde{Q}_1$ which makes $Q_1^T Q_1$ an orthogonal projection.

**Theorem E.1.** *(Formal statement and proof of Thm. 4.1) Call $-\lambda_1 > -\lambda_2 > \ldots$ the negative eigenvalues of $\mathcal{L}$ and $P_1, P_2, \ldots$ the projections onto the corresponding left eigen-space. Without loss of generality, assume $\lambda_1 = 1$. Assume $\dot{\tau}_t$ is bounded on every compact interval of $(0,1)$, $\pi_b > 0$ and $P_1 \vec{b} \neq 0$ for all $b$ and $P_1 \vec{b} \neq P_1 \vec{b}'$ for any $b \neq b'$. For each $\zeta$ pick time dilation $\tau_t^\zeta = \frac{1}{2} \log \left( \zeta e^{2\tau_t} - \zeta + 1 \right)$ and rescale $\vec{x}_t^\zeta = \sqrt{\zeta - (\zeta - 1) e^{-2\tau_t}} (\vec{x}_t - \pi)/\sqrt{\pi}$. Define the embedding into $\mathbb{R}^{\text{rank}(P_i)}$, $Q_i = j_i (\tilde{Q}_i \tilde{Q}_i^T)^{-1/2} \tilde{Q}_i$ where $\tilde{Q}_i = \text{diag}(\pi)^{-1/2} P_i \text{diag}(\pi)^{1/2}$ and $j_i$ is any isometry from $\text{Im}(\tilde{Q}_i) \to \mathbb{R}^{\text{rank}(P_i)}$.*

*Fix an $x_0$.*

- *(Path convergence) Call $(\vec{z}_t)_{t=0}^1$ the paths with $\vec{z}_0 = Q_1(\vec{x}_0/\sqrt{\pi})$ evolving under the Ornstein-Uhlenbeck process*

$$
d\tilde{z}_\tau = -\tilde{z}_\tau d\tau + \sqrt{2} dW_\tau
$$

*for a Brownian motion $(W_\tau)_{\tau=0}^\infty$ and call $\vec{z}_t = \tilde{z}_{\tau_t}$. Then $(Q_1 \vec{x}_t^\zeta)_{t \in (0,1)}$ converges in distribution to $(\vec{z}_t)_{t \in (0,1)}$ in the sense of Lem. E.8.*

- *(Convergence of non-dominant directions) The component of $\vec{x}_t^\zeta$ in $\text{Ker} Q_1$ is $\sum_{i>1} \tilde{Q}_i \vec{x}_t^\zeta$. Each component $(Q_i \vec{x}_t^\zeta)_t$ also converges to a Gaussian diffusion independent of $\vec{x}_0$ with modified time-dilation and scaling: call $(\vec{z}_t)_{t=0}^1$ the paths with $\vec{z}_0 \sim \mathcal{N}(0, I)$ independent of $x_0$ evolving, forward and backward on $(-\infty, \infty)$, under the stationary Ornstein-Uhlenbeck process*

$$
d\tilde{z}_\tau = -\tilde{z}_\tau d\tau + \sqrt{2} dW_\tau
$$

*for a Brownian motion $(W_\tau)_{\tau=0}^\infty$ and call $\vec{z}_t = \tilde{z}_{\tau_t^{(i)}}$ where $\tau_t^{(i)} = \frac{\lambda_i}{2} \log(e^{2\tau_t} - 1)$. Then*

$$
((1 - e^{-2\tau_t})^{-1/2} Q_i \vec{x}_t^\zeta)_{t \in (0,1)} \rightsquigarrow (\vec{z}_t)_{t \in (0,1)}.
$$

- *Call the ELBO in Alg. 3*

$$L(\vec{x}_t^\zeta, t, \vec{x}_0, \tilde{x}_0) = \sum_{b_1 \neq b_2} \mathcal{L}_{b_2 \to b_1} \dot{\tau}_t^\zeta \zeta \vec{x}_{t,b_1}(\vec{x}_t^\zeta) \mathbb{D}\left(\hat{w}(x_0)_{b_2,b_1} || \hat{w}(\tilde{x}_0)_{b_2,b_1}\right)$$

*where $\vec{x}_{t,b_1}(\vec{v})$ is the inverse of the transform from $\vec{x}_{t,b_1}$ to $\vec{x}_{t,b_1}^\zeta$. Then, for all $\vec{v}, t, \vec{x}_0, \tilde{x}_0$*

$$L(\vec{v}, t, \vec{x}_0, \tilde{x}_0) \to \frac{\dot{\tau}_t e^{-2\tau_t}}{(1 - e^{-2\tau_t})^2} \left\| \mathrm{emb}(x_0) - \mathrm{emb}(\tilde{x}_0) \right\|^2,$$

*the ELBO in Alg. 2, which, in particular, is independent of the value of $\vec{v}$.*

*Proof.* We prove the convergence of paths using Lem. E.8 which makes use of standard techniques. We break the proof up into four sections: the first three verify the conditions of Lem. E.8 and the last shows the convergence of the ELBO.

**Part 1. Convergence of Marginals:** Note

$$\vec{z}_t \sim e^{-\tau_t} \vec{z}_0 + \sqrt{1 - e^{-2\tau_t}} \mathcal{N}(0, I).$$

We want to prove convergence to this quantity. Note, writing $\mathrm{Mult}$ for a multinomial distribution,

$$\vec{x}_t^\zeta \sim \frac{\sqrt{\zeta - (\zeta - 1)e^{-2\tau_t}}}{\zeta} \left( \mathrm{Mult}(\zeta, \vec{x}_0^T e^{\tau_t^\zeta \mathcal{L}}) - \zeta \vec{\pi} \right) / \sqrt{\vec{\pi}}$$

$$= (1 + o(1)) \sqrt{1 - e^{-2\tau_t}} \left( \zeta^{-1/2} (\mathrm{Mult}(\zeta, \vec{x}_0^T e^{\tau_t^\zeta \mathcal{L}}) - \vec{x}_0^T e^{\tau_t^\zeta \mathcal{L}}) + \zeta^{1/2} (\vec{x}_0^T e^{\tau_t^\zeta \mathcal{L}} - \vec{\pi}) \right) / \sqrt{\vec{\pi}}.$$

The second term is

$$\sqrt{1 - e^{-2\tau_t}} \zeta^{1/2} (\vec{x}_0^T e^{\tau_t^\zeta \mathcal{L}} - \vec{\pi}) = \sqrt{1 - e^{-2\tau_t}} \sum_i \zeta^{1/2} e^{-\lambda_i \tau_t^\zeta} P_i \vec{x}_0$$

$$= \sum_i \left( \frac{\zeta(1 - e^{-2\tau_t})}{(\zeta(e^{2\tau_t} - 1) + 1)^{\lambda_i}} \right)^{1/2} P_i \vec{x}_0$$

$$\to e^{-\tau_t} P_1 \vec{x}_0.$$

For the first term, we need a "uniform" central limit theorem as the underlying distribution changes with $\zeta$ because of $\vec{x}_0^T e^{\tau_t^\zeta \mathcal{L}}$. Lem. E.9 shows that $\zeta^{-1/2}(\mathrm{Mult}(\zeta, \vec{x}_0^T e^{\tau_t^\zeta \mathcal{L}}) - \vec{x}_0^T e^{\tau_t^\zeta \mathcal{L}})$ approaches $\mathcal{N}(0, \mathrm{diag}(\vec{p}_t) - \vec{p}_t \vec{p}_t^T)$ for $\vec{p}_t = \vec{x}_0^T e^{\tau_t^\zeta \mathcal{L}}$, which itself approaches $\vec{\pi}$ as $\tau_t^\zeta \to \infty$. Therefore the first term, divided by $\sqrt{\vec{\pi}}$ approaches

$$\sqrt{1 - e^{-2\tau_t}} \mathcal{N}\left(0, I - \sqrt{\vec{\pi}} \sqrt{\vec{\pi}}^T\right).$$

Note $\tilde{Q}_i \sqrt{\vec{\pi}} = \sqrt{\vec{\pi}}^{-1} P_i \vec{\pi} = 0$ for each $i$ and, for $i > 1$, $\tilde{Q}_i(P_1 \vec{x}_0 / \sqrt{\vec{\pi}}) = \sqrt{\vec{\pi}}^{-1} P_i P_1 \vec{x}_0 = 0$. Therefore, as desired,

$$Q_1 x_t^\zeta \rightsquigarrow \sqrt{1 - e^{-2\tau_t}} \mathcal{N}(0, I) + e^{-\tau_t} \mathrm{emb}(x_0),$$

and for $i > 1$,

$$(1 - e^{-2\tau_t})^{-1/2} Q_i x_t^\zeta \rightsquigarrow \mathcal{N}(0, I).$$

**Part 2. Local uniform convergence of conditionals:** Note

$$\vec{z}_t | \vec{z}_s \sim e^{-(\tau_t - \tau_s)} \vec{z}_s + \sqrt{1 - e^{-2(\tau_t - \tau_s)}} \mathcal{N}(0, I).$$

We want to prove convergence to this quantity. Note

$$\vec{x}_t | \vec{x}_s \sim \sum_b \mathrm{Mult}(\zeta \vec{x}_{s,b}, \vec{b}^T e^{(\tau_t^\zeta - \tau_s^\zeta)\mathcal{L}}) / \zeta$$

where $\vec{x}_t = \sqrt{\vec{\pi}} \circ \vec{x}_t^\zeta / \sqrt{\zeta - (\zeta - 1)e^{-2\tau_t}} + \pi$ are the "unscaled" versions of the vector and $\vec{x}_s$ is similar. It will be convenient below to extend this definition to $\vec{x}_s^\zeta$ for which $\zeta \vec{x}_{s,b}$ are not integers, but which still satisfy $\sum_b \sqrt{\pi_b} \vec{x}_{t,b}^\zeta = 0$. To do so, we just round $\zeta \vec{x}_{s,b}$ down to $\lfloor \zeta \vec{x}_{s,b} \rfloor$.

Fix $\vec{v}$. We now show $\vec{x}_t^\zeta | \vec{x}_s^\zeta = \vec{v} \rightsquigarrow \vec{z}_t | \vec{z}_s = \vec{v}$; a very similar argument also shows $\vec{x}_t^\zeta \rightsquigarrow \vec{z}_t$. Call $\vec{x}^\zeta$ a variable distributed as $\vec{x}_t^\zeta | \vec{x}_s^\zeta = \vec{v}$, so, calling

$$w_t^\zeta = \frac{\sqrt{\zeta - (\zeta - 1)e^{-2\tau_t}}}{\zeta},$$

$$N_{s,b}^\zeta = \sqrt{\pi_b}\vec{v}_b/w_s^\zeta + \zeta\pi_b,$$

$$C_{t,b}^\zeta \sim \text{Mult}\left(\left\lfloor N_{s,b}^\zeta \right\rfloor, \vec{b}^T e^{(\tau_t^\zeta - \tau_s^\zeta)\mathcal{L}}\right) \text{ independent across } b,$$

then

$$\vec{x}_t^\zeta \sim w_t^\zeta \left(\sum_b C_{t,b}^\zeta - \zeta\pi\right)/\sqrt{\pi}$$

$$= w_t^\zeta \left(\sum_b \left[(C_{t,b}^\zeta - N_{s,b}^\zeta \vec{b}^T e^{(\tau_t^\zeta - \tau_s^\zeta)\mathcal{L}}) + N_{s,b}^\zeta (\vec{b}^T e^{(\tau_t^\zeta - \tau_s^\zeta)\mathcal{L}} - \pi)\right]\right)/\sqrt{\pi}$$

noting $\sum_b p_{t,b}^\zeta = \zeta$. This is exactly the "noise, signal" breakdown we had in the proof sketch.

For the signal (second term), first note

$$\sum_b \pi_b(\vec{b}^T e^{(\tau_t^\zeta - \tau_s^\zeta)\mathcal{L}} - \vec{\pi}) = \vec{\pi}^T e^{(\tau_t^\zeta - \tau_s^\zeta)\mathcal{L}} - \vec{\pi} = 0,$$

so, ignoring the $\pi$ term in $N_{s,b}^\zeta$ the second term is

$$\frac{w_t}{w_s}\left(\sum_b \sqrt{\pi_b}\vec{v}_b(\vec{b}^T e^{(\tau_t^\zeta - \tau_s^\zeta)\mathcal{L}} - \vec{\pi})\right)/\sqrt{\pi} = \frac{w_t}{w_s}\left(\left(\sqrt{\pi} \circ \vec{v}\right)^T e^{(\tau_t^\zeta - \tau_s^\zeta)\mathcal{L}}\right)/\sqrt{\pi}$$

$$= (1 + o(1))\sum_i \left(\frac{1 - e^{-2\tau_s}}{1 - e^{-2\tau_t}}\right)^{(\lambda_i - 1)/2} e^{-\lambda_i(\tau_t - \tau_s)}\tilde{Q}_i\vec{v}.$$

For the first term, we again apply Lem. E.9, noting $N_{s,b}^\zeta = (1 + o(1))\zeta\pi_b$ to get

$$\sum_b w_t(C_{t,b}^\zeta - N_{s,b}^\zeta \vec{b}^T e^{(\tau_t^\zeta - \tau_s^\zeta)\mathcal{L}})/\sqrt{\vec{\pi}}$$

$$\rightsquigarrow \sqrt{1 - e^{-2\tau_t}}\sum_b \sqrt{\pi_b}\mathcal{N}\left(0, \text{diag}(\vec{b}^T e^{(\tau_t^\zeta - \tau_s^\zeta)\mathcal{L}}) - e^{(\tau_t^\zeta - \tau_s^\zeta)\mathcal{L}^T}\vec{b}\vec{b}^T e^{(\tau_t^\zeta - \tau_s^\zeta)\mathcal{L}}\right)/\sqrt{\vec{\pi}}$$

$$= \sqrt{1 - e^{-2\tau_t}}\mathcal{N}\left(0, \text{diag}(\vec{\pi}^T e^{(\tau_t^\zeta - \tau_s^\zeta)\mathcal{L}}) - e^{(\tau_t^\zeta - \tau_s^\zeta)\mathcal{L}^T}\text{diag}(\vec{\pi})e^{(\tau_t^\zeta - \tau_s^\zeta)\mathcal{L}}\right)/\sqrt{\vec{\pi}}$$

$$= \sqrt{1 - e^{-2\tau_t}}\mathcal{N}\left(0, \text{diag}(\vec{\pi}) - (\sum_i e^{-\lambda_i(\tau_t^\zeta - \tau_s^\zeta)}P_i)\text{diag}(\vec{\pi})(\sum_i e^{-\lambda_i(\tau_t^\zeta - \tau_s^\zeta)}P_i^T)\right)/\sqrt{\vec{\pi}}$$

$$= \sqrt{1 - e^{-2\tau_t}}\mathcal{N}\left(0, I - (\sum_i e^{-\lambda_i(\tau_t^\zeta - \tau_s^\zeta)}\tilde{Q}_i)(\sum_i e^{-\lambda_i(\tau_t^\zeta - \tau_s^\zeta)}\tilde{Q}_i^T)\right).$$

Therefore, as desired,

$$Q_1\vec{x}_t^\zeta | \vec{x}_s^\zeta = \vec{v} \sim e^{-(\tau_t - \tau_s)}Q_1\vec{v} + \sqrt{(1 - e^{-2\tau_t})\left(1 - \frac{1 - e^{-2\tau_s}}{1 - e^{-2\tau_t}}e^{-2(\tau_t - \tau_s)}\right)}\mathcal{N}(0, I)$$

$$= \vec{v} \sim e^{-(\tau_t - \tau_s)}Q_1\vec{v} + \sqrt{1 - e^{-2(\tau_t - \tau_s)}}\mathcal{N}(0, I)$$

and similarly

$$(1 - e^{-2\tau_t})^{-1/2} Q_i \vec{x}_t^\zeta \mid \vec{x}_s^\zeta = \vec{v} \sim \left( \frac{1 - e^{-2\tau_s}}{1 - e^{-2\tau_t}} \right)^{\lambda_i/2} e^{-\lambda_i(\tau_t - \tau_s)} ((1 - e^{-2\tau_2})^{-1/2} Q_i \vec{v})$$
$$+ \sqrt{1 - \left( \frac{1 - e^{-2\tau_s}}{1 - e^{-2\tau_t}} \right)^{\lambda_i} e^{-2\lambda_i(\tau_t - \tau_s)}} \mathcal{N}(0, I)$$
$$= e^{-(\tau_t^{(i)} - \tau_s^{(i)})} ((1 - e^{-2\tau_2})^{-1/2} Q_i \vec{v})$$
$$+ \sqrt{1 - e^{-2(\tau_t^{(i)} - \tau_s^{(i)})}} \mathcal{N}(0, I)$$

Finally, convergence is clearly uniform for nearby $\vec{v}$ using the uniformity of Lem. E.9.

**Part 3. Tightness:** Pick $s < t \in (0, 1)$.

$$\mathbb{E} \|\vec{x}_t^\zeta - \vec{x}_s^\zeta\|^2 = \mathbb{E} \|\mathbb{E}[\vec{x}_t^\zeta | \vec{x}_s^\zeta] - \vec{x}_s^\zeta\|^2 + \mathbb{E} \|\vec{x}_t^\zeta - \mathbb{E}[\vec{x}_t^\zeta | \vec{x}_s^\zeta]\|^2$$

The first term has, for each $x_0$,

$$\mathbb{E} \|\mathbb{E}[\vec{x}_t^\zeta | \vec{x}_s^\zeta] - \vec{x}_s^\zeta\|^2 = \mathbb{E} \|w_t (\vec{x}_s e^{(\tau_t^\zeta - \tau_s^\zeta)\mathcal{L}} - \vec{\pi}) / \sqrt{\pi} - \vec{x}_s^\zeta\|^2$$
$$= \frac{1}{\min_b \pi_b} \mathbb{E} \|w_t (\vec{x}_s e^{(\tau_t^\zeta - \tau_s^\zeta)\mathcal{L}} - \vec{x}_s) - (w_s - w_t)(\vec{x}_s - \vec{\pi})\|^2$$
$$\leqslant \frac{1}{\min_b \pi_b} \mathbb{E} \left( |w_t| \|\vec{x}_s e^{(\tau_t^\zeta - \tau_s^\zeta)\mathcal{L}} - \vec{x}_s\| + |w_s - w_t| \|\vec{x}_s - \vec{\pi}\| \right)^2$$
$$= \frac{1}{\min_b \pi_b} \mathbb{E} \left( |w_t| \|(\vec{x}_s - \vec{\pi})^T (I - e^{(\tau_t^\zeta - \tau_s^\zeta)\mathcal{L}})\| + |w_s - w_t| \|\vec{x}_s - \vec{\pi}\| \right)^2$$
$$\leqslant \frac{1}{\min_b \pi_b} \mathbb{E} \left( |w_t| (1 - e^{-(\tau_t^\zeta - \tau_s^\zeta)\lambda_B}) \|\vec{x}_s - \vec{\pi}\| + |w_s - w_t| \|\vec{x}_s - \vec{\pi}\| \right)^2$$
$$= \frac{1}{\min_b \pi_b} \left( |w_t| (1 - e^{-(\tau_t^\zeta - \tau_s^\zeta)\lambda_B}) + |w_s - w_t| \right)^2 \mathbb{E} \|\vec{x}_s - \vec{\pi}\|^2$$
$$\leqslant \frac{\zeta}{\min_b \pi_b} \left( (1 - e^{-(\tau_t^\zeta - \tau_s^\zeta)\lambda_B}) + |1 - \frac{w_s}{w_t}| \right)^2$$
$$\times \left( \mathbb{E} \text{TrCov}(\text{Mult}(\zeta, \vec{x}_0^T e^{\tau_s^\zeta \mathcal{L}} / \zeta) + \|\vec{x}_0^T e^{\tau_s^\zeta \mathcal{L}} - \vec{\pi}\|^2 \right)$$
$$\leqslant \frac{1}{\min_b \pi_b} \left( (1 - e^{-(\tau_t^\zeta - \tau_s^\zeta)\lambda_B}) + |1 - \frac{w_s}{w_t}| \right)^2 (1 + \zeta e^{-2\tau_s^\zeta})$$
$$\leqslant \frac{1}{\min_b \pi_b} \left( (1 - e^{-(\tau_t^\zeta - \tau_s^\zeta)\lambda_B}) + |1 - \frac{w_s}{w_t}| \right)^2 \left( 1 + \frac{1}{e^{2\tau_s} - 1} \right)$$

Now,

$$1 - e^{-(\tau_t^\zeta - \tau_s^\zeta)\lambda_B} = 1 - e^{-2\lambda_B(\tau_t - \tau_s)} \left( \frac{1 - e^{-2\tau_s}(1 - \zeta^{-1})}{1 - e^{-2\tau_t}(1 - \zeta^{-1})} \right)^{\lambda_B/2}$$
$$\leqslant 1 - e^{-2\lambda_B(\tau_t - \tau_s)}$$
$$+ 1 - \left( \frac{1 - e^{-2\tau_s}(1 - \zeta^{-1})}{1 - e^{-2\tau_t}(1 - \zeta^{-1})} \right)^{\lambda_B/2}.$$

When $|\tau_s - \tau_t| < 1/4\lambda_B$

$$1 - e^{-2\lambda_B(\tau_t - \tau_s)} \leqslant 4\lambda_B(\tau_t - \tau_s) \leqslant 4\lambda_B |t - s| \sup_{u \in [s,t]} \dot{\tau}_u.$$

Next note that if $\alpha \geqslant 1$, $x \mapsto 1 - x^\alpha$ has decreasing derivative, from $0$ to $-\alpha$ on the interval $x \in [0, 1]$, so, it is dominated on this interval by $\alpha(1 - x)$. If $\zeta > 1$,

$$
\begin{aligned}
1 - \left(\frac{1 - e^{-2\tau_s}(1 - \zeta^{-1})}{1 - e^{-2\tau_t}(1 - \zeta^{-1})}\right)^{\lambda_B/2} &\leqslant 1 - \left(\frac{1 - e^{-2\tau_s}(1 - \zeta^{-1})}{1 - e^{-2\tau_t}(1 - \zeta^{-1})}\right)^{1 \vee (\lambda_B/2)} \\
&\leqslant (1 \vee (\lambda_B/2))\left(1 - \left(\frac{1 - e^{-2\tau_s}(1 - \zeta^{-1})}{1 - e^{-2\tau_t}(1 - \zeta^{-1})}\right)\right) \\
&\leqslant (1 \vee (\lambda_B/2))\left(\frac{(e^{-2\tau_s} - e^{-2\tau_t})(1 - \zeta^{-1})}{1 - e^{-2\tau_t}}\right) \\
&\leqslant \frac{1 \vee (\lambda_B/2)e^{-2\tau_s}}{1 - e^{-2\tau_t}}\left(1 - e^{-2(\tau_t - \tau_s)}\right) \\
&\leqslant \frac{4 \vee (2\lambda_B)e^{-2\tau_s}}{1 - e^{-2\tau_t}}|t - s| \sup_{u \in [s,t]} \dot{\tau}_u
\end{aligned}
$$

Finally

$$
1 - \frac{w_s}{w_t} = 1 - \left(\frac{1 - e^{-2\tau_s}(1 - \zeta^{-1})}{1 - e^{-2\tau_t}(1 - \zeta^{-1})}\right)^{1/2}.
$$

which is similar to above.

The second term has

$$
\begin{aligned}
\mathbb{E}\|\vec{x}_t^\zeta - \mathbb{E}[\vec{x}_t^\zeta | \vec{x}_s^\zeta]\|^2 &\leqslant \frac{2\zeta}{\min_b \pi_b} \sum_b \mathbb{E}\mathrm{TrCov}(\mathrm{Mult}(\zeta \vec{x}_{s,b}, \vec{b}^T e^{(\tau_t^\zeta - \tau_s^\zeta)\mathcal{L}})/\zeta \mid \vec{x}_t^\zeta) \\
&= \frac{2}{\min_b \pi_b} \sum_b \mathbb{E}\vec{x}_{s,b}^\zeta \sum_{b'} (\vec{b}^T e^{(\tau_t^\zeta - \tau_s^\zeta)\mathcal{L}}\vec{b}')(1 - \vec{b}^T e^{(\tau_t^\zeta - \tau_s^\zeta)\mathcal{L}}\vec{b}') \\
&\leqslant \frac{2}{\min_b \pi_b}\left(\sum_{b \neq b'} \vec{b}^T e^{(\tau_t^\zeta - \tau_s^\zeta)\mathcal{L}}\vec{b}' + \sum_b (1 - \vec{b}^T e^{(\tau_t^\zeta - \tau_s^\zeta)\mathcal{L}}\vec{b})\right) \\
&= \frac{4}{\min_b \pi_b} \sum_b (1 - \vec{b}^T e^{(\tau_t^\zeta - \tau_s^\zeta)\mathcal{L}}\vec{b}) \\
&\leqslant \frac{4B}{\min_b \pi_b}(1 - e^{-(\tau_t^\zeta - \tau_s^\zeta)\lambda_B})
\end{aligned}
$$

which is bounded similar to the first term.

**Part 4. Convergence of the ELBO:** Define $p = \vec{x}_0^T e^{\tau_t^\zeta \mathcal{L}}$. We've shown above that

$$
p = \vec{\pi} + \sqrt{\frac{1}{\zeta(e^{2\tau_t} - 1)}} P_1 \vec{x}_0 + o(\zeta^{-1/2})
$$

so

$$
\frac{p_{b_2}}{p_{b_1}} = \frac{\pi_{b_2}}{\pi_{b_1}} + \frac{1}{\pi_{b_1}}\sqrt{\frac{1}{\zeta(e^{2\tau_t} - 1)}}\left(\vec{b}_2 - \frac{\pi_{b_2}}{\pi_{b_1}}\vec{b}_1\right)^T P_1 \vec{x}_0 + o(\zeta^{-1/2})
$$

and similar for $q$. Using a second-order Taylor expansion on $\mathbb{D}$, we get

$$
\mathbb{D}(\hat{w}(x_0)_{b_2,b_1} \| \hat{w}(\tilde{x}_0)_{b_2,b_1}) = \frac{1}{2}\frac{\pi_{b_1}}{\pi_{b_2}}\frac{1}{\pi_{b_1}^2 \zeta(e^{2\tau_t} - 1)}\left(\left(\vec{b}_2 - \frac{\pi_{b_2}}{\pi_{b_1}}\vec{b}_1\right)^T P_1(\vec{x}_0 - \tilde{x}_0)\right)^2 + o(\zeta^{-1}).
$$

Next note $\dot{\tau}_t^\zeta = \dot{\tau}_t \frac{e^{2\tau_t}}{e^{2\tau_t} - 1} + o(1)$. Finally note

$$
\vec{x}_t(\vec{v}) = \sqrt{\pi} \circ \vec{v}/\sqrt{\zeta - (\zeta - 1)e^{-2\tau_t}} + \pi = \pi + o(1).
$$

Putting this together, we get

$$L(\vec{v}, t, \vec{x}_0, \tilde{x}_0)$$

$$= \sum_{b_1 \neq b_2} \mathcal{L}_{b_2 \to b_1} \dot{\tau}_t^{\zeta} \zeta \vec{x}_{t, b_1}(\vec{v}) \mathbb{D}\left(\frac{p_{b_2}}{p_{b_1}} \middle\| \frac{q_{b_2}}{q_{b_1}}\right)$$

$$= \dot{\tau}_t \sum_{b_1 \neq b_2} \mathcal{L}_{b_2 \to b_1} \frac{e^{2\tau_t}}{e^{2\tau_t} - 1} \pi_{b_1} \frac{1}{2\pi_{b_2} \pi_{b_1}} \frac{1}{(e^{2\tau_t} - 1)} \left(\left(\vec{b}_2 - \frac{\pi_{b_2}}{\pi_{b_1}} \vec{b}_1\right)^T P_1(\vec{x}_0 - \tilde{x}_0)\right)^2 + o(1)$$

$$= \frac{\dot{\tau}_t e^{2\tau_t}}{2(e^{2\tau_t} - 1)^2} \sum_{b_1 \neq b_2} \mathcal{L}_{b_2 \to b_1} \left(\left(\vec{b}_2 - \sqrt{\frac{\pi_{b_2}}{\pi_{b_1}}} \vec{b}_1\right)^T \tilde{Q}_1 \left((\vec{x}_0 - \tilde{x}_0)/\sqrt{\vec{\pi}}\right)\right)^2 + o(1)$$

$$= \frac{\dot{\tau}_t e^{-2\tau_t}}{(1 - e^{-2\tau_t})^2} \left\| \tilde{Q}_1 \left((\vec{x}_0 - \tilde{x}_0)/\sqrt{\vec{\pi}}\right)\right\|_{\Sigma}^2 + o(1)$$

where

$$\Sigma = \frac{1}{2} \sum_{b_1 \neq b_2} \mathcal{L}_{b_2 \to b_1} \left(\vec{b}_2 - \sqrt{\frac{\pi_{b_2}}{\pi_{b_1}}} \vec{b}_1\right) \left(\vec{b}_2 - \sqrt{\frac{\pi_{b_2}}{\pi_{b_1}}} \vec{b}_1\right)^T.$$

To solve $\Sigma$, we note

$$\sum_{b_1 \neq b_2} \mathcal{L}_{b_2 \to b_1} \vec{b}_2 \vec{b}_2^T = \sum_{b_2} \vec{b}_2 \vec{b}_2^T \sum_{b_1 \neq b_2} \mathcal{L}_{b_2 \to b_1} = -\sum_{b_2} \vec{b}_2 \vec{b}_2^T \mathcal{L}_{b_2, b_2}$$

$$\sum_{b_1 \neq b_2} \frac{\pi_{b_2}}{\pi_{b_1}} \mathcal{L}_{b_2 \to b_1} \vec{b}_2 \vec{b}_2^T = \sum_{b_1} \vec{b}_1 \vec{b}_1^T \sum_{b_2 \neq b_1} \frac{\pi_{b_2}}{\pi_{b_1}} \mathcal{L}_{b_2 \to b_1} = -\sum_{b_1} \vec{b}_1 \vec{b}_1^T \mathcal{L}_{b_1, b_1}$$

$$\sum_{b_1 \neq b_2} \sqrt{\frac{\pi_{b_2}}{\pi_{b_1}}} \mathcal{L}_{b_2 \to b_1} \vec{b}_2 \vec{b}_1^T = \mathrm{diag}(\sqrt{\vec{\pi}}) \left(\mathcal{L} - \mathrm{diag}\mathcal{L}\right) \mathrm{diag}(1/\sqrt{\vec{\pi}})$$

$$\sum_{b_1 \neq b_2} \sqrt{\frac{\pi_{b_2}}{\pi_{b_1}}} \mathcal{L}_{b_2 \to b_1} \vec{b}_1 \vec{b}_2^T = (\mathrm{diag}(\sqrt{\vec{\pi}}) \left(\mathcal{L} - \mathrm{diag}\mathcal{L}\right) \mathrm{diag}(1/\sqrt{\vec{\pi}}))^T.$$

So,

$$\Sigma = -\frac{1}{2} \mathrm{diag}(\sqrt{\vec{\pi}}) \mathcal{L} \mathrm{diag}(1/\sqrt{\vec{\pi}}) - \frac{1}{2} (\mathrm{diag}(\sqrt{\vec{\pi}}) \mathcal{L} \mathrm{diag}(1/\sqrt{\vec{\pi}}))^T.$$

In particular, since $\tilde{Q}_1^T \mathrm{diag}(\sqrt{\vec{\pi}}) \mathcal{L} \mathrm{diag}(1/\sqrt{\vec{\pi}}) = -\tilde{Q}_1^T$,

$$\tilde{Q}_1^T \Sigma \tilde{Q}_1 = \tilde{Q}_1^T \tilde{Q}_1 = Q_1^T Q_1.$$

This gives us

$$L(\vec{v}, t, \vec{x}_0, \tilde{x}_0) \to \frac{\dot{\tau}_t e^{-2\tau_t}}{(1 - e^{-2\tau_t})^2} \|\mathrm{emb}(x_0) - \mathrm{emb}(\tilde{x}_0)\|^2.$$

$\square$

### E.3 Hollow parameterization solves Gaussian ELBO singularity

Here we show that the hollow parametrization introduced in Sec. 4.2 resolves the singularity of the Gaussian ELBO in Alg. 2 at $t \to 0^+$. Before going into the proof, let us give some intuition. Assume, $x_0^d$ were distributed uniformly and independently. Then

$$p(x_0^d \mid x_t, t) \propto p(x_t^d \mid x_0^d, t) p(x_0^d \mid x_t^{-d}, t),$$

where $x_t^{-d}$ includes all positions but $d$. However

$$p(x_0^d \mid x_t^{-d}, t) = \int p(x_0^d \mid x_0^{-d}) dp(x_0^{-d} \mid x_t^{-d}, t) = \text{Uniform}.$$

Therefore, we get $p(x_0^d \mid x_t, t) \propto p(x_t^d \mid x_0^d, t)$. At initialization, we can say our neural network $q_\theta(x_0^d \mid x_t^{-d}, t) \approx \text{Uniform}$, so,

$$q_\theta(x_0^d \mid x_t, t) \approx p(x_0^d \mid x_t, t).$$

Therefore, **the hollow parametrization initializes the diffusion model near a uniform, site-wise independent model.** The proof below involves a lot of algebra, but the basic intuition for why we should not see singularities is that by initializing at a *valid* diffusion model, we get comparable ELBOs.

Again we assume $D = 1$ for simplicity as results are straightforward to generalize to higher $D$.

**Proposition E.2.** *Assume* emb *is injective and $\tau_t$ is increasing and differentiable. Define*

$$L = \frac{\dot{\tau}_t e^{-2\tau_t}}{(1 - e^{-2\tau_t})^2} \|\mathrm{emb}(x_0) - \mathrm{emb}(\tilde{x}_0)\|^2,$$

*and the normalized vectors $\vec{\phi}(x_t, t) \propto p(x_t \mid x_0, t)$. For $\tilde{x}_0$ build using the hollow predictor $\tilde{x}_0 = \vec{\phi}(x_t, t) \circ \vec{q}/\vec{\phi}(x_t, t)^T \circ \vec{q}$ for a vector $t$ bounded away from $0$ and $\infty$,*

$$0 < c = \min_b \vec{q}_b \leqslant \max_b \vec{q}_b < C < \infty,$$

*we have*

$$\mathbb{E}_{t, x_0, x_t} L < \infty.$$

*Proof.* Note first
$$\|\mathrm{emb}(x_0) - \mathrm{emb}(\tilde{x}_0)\|^2 \leqslant \|\mathrm{emb}\|^2 \|\vec{x}_0 - \tilde{x}_0\|$$
and, simplifying $\vec{\phi} = \vec{\phi}(x_t, t)$,
$$\mathbb{E}_{x_0 \mid x_t} \|\vec{x}_0 - \tilde{x}_0\| = \|\vec{\phi} \circ \vec{p}/\vec{\phi}^T \vec{p} - \vec{\phi} \circ \vec{q}/\vec{\phi}^T \vec{q}\|$$
for $\vec{p}_b = p(x_0)$.

Call $b = \mathrm{argmax}_{b'} \vec{\phi}_{b'}$, so

$$\|\vec{\phi} \circ \vec{p}/\vec{\phi}^T \vec{p} - \vec{\phi} \circ \vec{q}/\vec{\phi}^T \vec{q}\| \leqslant \left( \frac{\vec{\phi}_b \vec{p}_b}{\vec{\phi}^T \vec{p}} - \frac{\vec{\phi}_b \vec{q}_b}{\vec{\phi}^T \vec{q}} \right)^2 + (1 - \vec{\phi}_b)^2 \sum_{b' \neq b} \left( \frac{\vec{p}_{b'}}{\vec{\phi}^T \vec{p}} - \frac{\vec{q}_{b'}}{\vec{\phi}^T \vec{q}} \right)^2$$

$$= \left( \frac{1}{1 + \sum_{b' \neq b} \frac{\vec{\phi}_{b'} \vec{p}_{b'}}{\vec{\phi}_b \vec{p}_b}} - \frac{1}{1 + \sum_{b' \neq b} \frac{\vec{\phi}_{b'} \vec{q}_{b'}}{\vec{\phi}_b \vec{q}_b}} \right)^2$$

$$+ \left( \frac{C}{c} \right)^2 B(1 - \phi_b)^2$$

$$\leqslant \left( 1 - \frac{1}{1 + \frac{CB(1 - \phi_b)}{c}} \right)^2$$

$$\leqslant \left( \frac{CB}{c} \right)^2 (1 - \phi_b)^2 + \left( \frac{C}{c} \right)^2 B(1 - \phi_b)^2.$$

We've therefore bounded $\mathbb{E}_{t, x_0, x_t} L$ above by some constant times $\mathbb{E}_{t, x_t} \frac{\dot{\tau}_t e^{-2\tau_t}}{(1 - e^{-2\tau_t})^2} (1 - \max_b \vec{\phi}_b)^2$.

Note without the hollow parameterization, we wouldn't have the $(1 - \max_b \vec{\phi}_b)^2$ term; we now show this becomes small very fast as $t \to 0$ (because $x_0$ becomes "obvious" from $x_t$), cancelling out the singularity.

Next note, calling $b = \mathrm{argmin}_{b'} \|\mathrm{emb}(b') - \vec{x}_t\|$,

$$(1 - \max_b \vec{\phi}_b)^2 = \left( 1 - \frac{1}{1 + \sum_{b' \neq b} \exp(-\frac{1}{2(1 - e^{-2\tau_t})^2} (\|\mathrm{emb}(b') - \vec{x}_t\|^2 - \|\mathrm{emb}(b) - \vec{x}_t\|^2))} \right)^2$$

$$\leqslant \sum_{b' \neq b} \exp \left( -\frac{1}{2(1 - e^{-2\tau_t})} (\|\mathrm{emb}(b') - \vec{x}_t\|^2 - \|\mathrm{emb}(b) - \vec{x}_t\|^2) \right),$$

which is only large if $\vec{x}_t$ is roughly equidistant to two potential $x_0$. Call $\epsilon = \min_{b \neq b'} \|\text{emb}(b) - \text{emb}(b')\|/4$, so, if $\min_{b'} \|\text{emb}(b') - \vec{x}_t\| < \epsilon$ then, by the triangle inequality

$$
\begin{aligned}
\|\text{emb}(b') - \vec{x}_t\|^2 - \|\text{emb}(b) - \vec{x}_t\|^2 &\geqslant (\|\text{emb}(b) - \text{emb}(b')\| - \|\text{emb}(b) - \vec{x}_t\|)^2 \\
&\quad - \|\text{emb}(b) - \vec{x}_t\|^2 \\
&= \|\text{emb}(b) - \text{emb}(b')\| \\
&\quad - 2\|\text{emb}(b) - \text{emb}(b')\|\|\text{emb}(b) - \vec{x}_t\| \\
&\geqslant 16\epsilon^2 - 8\epsilon^2 = 8\epsilon^2.
\end{aligned}
$$

Therefore, $\mathbb{E}_{t,x_t} \frac{\dot{\tau}_t e^{-2\tau_t}}{(1-e^{-2\tau_t})^2}(1 - \max_b \vec{\phi}_b)^2$ is bounded by

$$
B\mathbb{E}_t \frac{\dot{\tau}_t e^{-2\tau_t}}{(1-e^{-2\tau_t})^2}\left(\exp\left(-\frac{4\epsilon^2}{(1-e^{-2\tau_t})}\right) + p(\min_{b'}\|\text{emb}(b') - \vec{x}_t\| \geqslant \epsilon)\right).
$$

To deal with the first term, perform a change of variables $u = (1 - e^{-2\tau_t})^{-1}$, giving

$$
\mathbb{E}_t \frac{\dot{\tau}_t e^{-2\tau_t}}{(1-e^{-2\tau_t})^2}\exp\left(-\frac{4\epsilon^2}{(1-e^{-2\tau_t})}\right) = \frac{1}{2}\int_0^\infty du \exp(-4\epsilon^2 u) < \infty.
$$

For the second term, note

$$
\begin{aligned}
p(\min_{b'}\|\text{emb}(b') - \vec{x}_t\| \geqslant \epsilon) &\leqslant \sum_b p(x_0 = b)p(\|\mathcal{N}(0, (1-e^{-2\tau_t})I_{r\times r})\| > \epsilon) \\
&= p(\chi_r^2/\epsilon^2 > 1/(1-e^{-2\tau_t}))
\end{aligned}
$$

where $\chi_r^2$ is a chi-squared distribution with $r$ degrees of freedom. Finally, by the same change of variables $u$ as above, we get

$$
\mathbb{E}_t \frac{\dot{\tau}_t e^{-2\tau_t}}{(1-e^{-2\tau_t})^2}p(\min_{b'}\|\text{emb}(b') - \vec{x}_t\| \geqslant \epsilon) = \frac{1}{2}\int_0^\infty du\, p(\chi_r^2/\epsilon^2 > u) = \frac{1}{2}\mathbb{E}\chi_r^2/\epsilon^2 < \infty.
$$

$\square$

### E.4 Every embedding can be induced from some infinitesimal generator

Define an injective embedding $\text{emb} : \{1, \ldots, B\} \to \mathbb{R}^r$ for some $r$. For an infinitesimal generator $\mathcal{L}$ with a unique stationary distribution $\vec{\pi}$, define $Q_1 = \mathsf{j}_1(\tilde{Q}_1 \tilde{Q}_1^T)^{-1/2}\tilde{Q}_1$, $\mathsf{j}_1$ is some isometry, $\tilde{Q}_1 = \text{diag}(\vec{\pi})^{-1/2}P_1\text{diag}(\vec{\pi})^{1/2}$ where $P_1$ is the projection onto the first left eigenspace. Is there a choice of $\mathcal{L}$ such that $Q_1(\vec{x}_0/\sqrt{\vec{\pi}_{x_0}}) = \text{emb}(x_0)$ for every $x_0$?

If we restrict to $\mathcal{L} \in \mathbb{R}^{B \times B}$ then the answer is no. To see this, call $W \in \mathbb{R}^{B \times r}$ the matrix with $W\vec{b} = \text{emb}(b)$. Then, defining $D = \text{diag}(\vec{\pi})^{-1/2}$, we need $W^T W = DQ_1Q_1^T D = DPD$ for some orthogonal projection $P$ or rank $r$. The space $\{W^T W \mid W \in \mathbb{R}^{B \times r}\}$ generates all rank-$r$ positive-semi-definite matrices, an algebraic variety of dimension $B \times r$. Meanwhile, $P$ has $r(B-r)$ degrees of freedom and $D$ has $B - 1$, so $DPD$ generates an algebraic variety of dimension at most $B \times r - r^2 + B - 1$, which is less than $B \times r$ when $r$ is large.

If however we allow $r + 1$ "dummy" tokens, to let $\mathcal{L} \in \mathbb{R}^{(B+r)\times(B+r)}$, then the next proposition shows that the answer is yes. This demonstrates an important distinction between the design space of Gaussian and discrete diffusions: dummy variables which never appear in the data have no effect on the training of Gaussian diffusion, but can serve as transient states in discrete diffusion.

**Proposition E.3.** *There is some infinitesimal generator $\mathcal{L} \in \mathbb{R}^{(B+r+1)\times(B+r+1)}$ such that $Q_1(\vec{x}_0/\sqrt{\vec{\pi}}) = \text{emb}(x_0)$ for every $x_0 \in \{1, \ldots, B\}$. There are infinitely many such generators.*

*Proof.* Call $W \in \mathbb{R}^{B \times r}$ the matrix with $W\vec{b} = \text{emb}(b)$. Call $\Lambda = W^T W$ and without loss of generality, assume its first $r$ rows are linearly independent. We split the proof into two parts: first we show that $Q_1^T Q_1$ can equal $DPD$ for any orthogonal projection matrix $P$ with $P\sqrt{\vec{\pi}} = 0$ and $D = \text{diag}(\vec{\pi})^{-1/2}$ for any distribution $\pi$; then we show that $\Lambda$ can be written as the top $B \times B$ submatrix of $DPD$ for some choice opf $D$ and $P$. This will show that there is a $Q_1$ such that $Q_1(\cdot/\sqrt{\vec{\pi}}.)$ is equivalent to $\text{emb}$ up to isometry.

**Part 1**  Pick an orthogonal projection $P$ and a distribution $\pi$ such that $P\sqrt{\vec{\pi}} = 0$ Call $\tilde{P} = \operatorname{diag}(\vec{\pi})^{-1/2} P \operatorname{diag}(\vec{\pi})^{1/2}$ and

$$\mathcal{L}_\mu = -\left(I - \mathbb{1}\vec{\pi}^T\right) + \mu\tilde{P}.$$

Clearly, for every $\mu$, $\mathcal{L}_\mu \mathbb{1} = 0$ and $\vec{\pi}^T \mathcal{L}_\mu = 0$. Also, $\mathcal{L}_1 = -I + \mathbb{1}\vec{\pi}^T$, so for $\mu$ in a neighbourhood of 1, $\mathcal{L}_\mu$ has positive entries off the diagonal – therefore it's an infinitesimal generator – and $\mathcal{L}_\mu$ has a kernel of dimension $1$ – so $\pi$ is the unique stationary distribution of $\mathcal{L}_\mu$.

When $\mu$ is slightly greater than 0, the first eigenspace of $\mathcal{L}_\mu$ is that of $\tilde{P}$; in particular, when $\tilde{P}$ is a projection, $P_1 = \tilde{P}^T$ so $\tilde{Q}_1 = P$. Note $Q_1^T Q_1 = \tilde{Q}_1^T(\tilde{Q}_1\tilde{Q}_1^T)^{-1}\tilde{Q}_1$ which is the projection onto the orthogonal complement of $\operatorname{Ker}\tilde{Q}_1 = \operatorname{Ker}P$; therefore it is equal to $P$.

$P$ and $\vec{\pi}$ are the same for any small value of $\mu$, justifying the "infinitely many" proposal in the statement.

**Part 2**  First we need to ensure the rare case that $\mathbb{1}$ is orthogonal to the top eigenspace of $\Lambda$ does not occur. To ensure this, simple add another embedding $\operatorname{emb}(B+1) = \sum_b \vec{w}_b\operatorname{emb}(b)$ for some $\vec{w}$ to get a new matrix $\Lambda$ adding this extra token:

$$\tilde{\Lambda} := \begin{bmatrix} \Lambda & \Lambda\vec{w} \\ (\Lambda\vec{w})^T & \vec{w}^T\Lambda\vec{w} \end{bmatrix}$$

Pick a $\vec{v}$ so $\vec{v}^T\Lambda\vec{v} \neq 0$ and $\vec{w} = \eta\vec{v}$. As $\eta \to \infty$, $\tilde{\Lambda}/\eta^2 \to \vec{v}^T\Lambda\vec{v}(\vec{e}\vec{e}^T)$ where $e$ is the indicator vector for position $B+1$. Therefore for some $\eta$, the top eigenvector approaches $\vec{e}$ and is not orthogonal to $\mathbb{1}$. Below we simply assume that $\mathbb{1}$ is not orthogonal to the top eigenspace of $\Lambda$.

Decompose $\Lambda = \eta V \operatorname{diag}(\vec{\lambda}/\eta)V^T$ for a matrix $V \in \mathbb{R}^{B \times r}$ with orthonormal columns, a vector $\lambda$ of eigenvalues, and a scalar $\eta > \max_i \lambda_i$ to be chosen later. For an orthonormal matrix $U \in \mathbb{R}^{r \times r}$ to be chosen later, define

$$\tilde{V} = \begin{bmatrix} V\operatorname{diag}(\vec{\lambda}/\eta)^{1/2} \\ U(I - \operatorname{diag}(\vec{\lambda}/\eta))^{1/2} \end{bmatrix}$$

so $\tilde{V}$ has orthonormal columns. Define the orthogonal projection $P = \tilde{V}\tilde{V}^T$, so in particular, the upper $B \times B$ submatrix of $P$ is $\Lambda/\eta$.

Finally we'll pick $\eta$ and $U$ to get a positive normalized vector $\vec{\pi}$ such that $\vec{\pi}_b = 1/\eta$ for all $b \in \{1, \dots, B\}$ and $\tilde{V}\sqrt{\vec{\pi}} = 0$, completing the proof. Breaking $\vec{\pi}$ into its first $B$ components and other $r$ components, $[\mathbb{1}/\eta, \vec{\pi}_2]$, we can write the equation $\tilde{V}^T\sqrt{\vec{\pi}} = 0$ as

$$\vec{\pi}_2 = -\eta^{-3/2}U^{-1}\operatorname{diag}\left(\frac{\vec{\lambda}}{I - \vec{\lambda}/\eta}\right)^{1/2}V^T\mathbb{1}.$$

We can always choose $U$ to rotate to get $\vec{\pi}_2 = \mathbb{1}\eta'$ where

$$\eta' = \eta^{-3/2}\left\|\operatorname{diag}\left(\frac{\vec{\lambda}}{I - \vec{\lambda}/\eta}\right)^{1/2}V^T\mathbb{1}\right\|/\sqrt{r}.$$

Finally we need to solve for $\eta$ in

$$1 = B/\eta + \eta^{-3}\left\|\operatorname{diag}\left(\frac{\vec{\lambda}}{I - \vec{\lambda}/\eta}\right)^{1/2}V^T\mathbb{1}\right\|^2.$$

This is possible by the intermediate value theorem as the right hand side goes to 0 as $\eta \to \infty$ and goes to $\infty$ as $\eta \to \max_i \lambda_i$ from above (as we've assumed $V_{:,i}^T\mathbb{1} \neq 0$ for $i$ where $\lambda_i$ is the maximum eigenvalue). $\qquad\square$

### E.5 PROOF OF WRIGHT-FISHER CONVERGENCE

Define $\Delta^B \subset \mathbb{R}^B$ be the simplex, i.e. the set of non-negative vectors with components summing to 1. Let $(\vec{x}_t^\zeta)_{t=0}^1$ be a stochastic process on $(\frac{1}{\zeta}\mathbb{Z}^B) \cap \Delta^B$ with $\vec{x}_0^\zeta = \vec{x}_0$ evolving with respect to $\mathcal{L}^{\mathrm{mut}} + \zeta\mathcal{L}^{\mathrm{wf}}$ where

$$\mathcal{L}^{\mathrm{wf}}_{\vec{x}^\zeta \to \vec{x}'^\zeta} = \frac{\zeta!}{\prod_b \zeta\vec{x}_b'^\zeta!} \prod_b (\vec{x}_b^\zeta)^{\zeta\vec{x}_b'^\zeta} = \mathrm{Mult}(\zeta, \vec{x}^\zeta)(\zeta\vec{x}'^\zeta),$$

and, if $\vec{x}^\zeta, \vec{x}'^\zeta$ differ by one count $b \to b'$,

$$\mathcal{L}^{\mathrm{mut}}_{\vec{x}^\zeta \to \vec{x}'^\zeta} = (\psi(\mathbb{1}\vec{\pi}^T - I))_{b,b'} = \psi\vec{\pi}_{b'}$$

otherwise it's 0. Let $(\vec{z}_t)_t$ be a continuous Wright-Fisher process, that is, $\vec{z}_t = \vec{x}_0$ and

$$d\vec{z}_t = \frac{\psi}{2}(\vec{\pi} - \vec{z}_t)dt + \mathrm{diag}\left(\sqrt{\vec{z}_t}\right)\left(I - \sqrt{\vec{z}_t}\sqrt{\vec{z}_t}^T\right)d\vec{W}_t \tag{5}$$

where $(W_t)_t$ is a Brownian motion.

#### E.5.1 CONVERGENCE OF THE FORWARD PROCESS

We have convergence of the forward processes from previous literature.

**Theorem E.4.** *(Thm 1.1 Ethier and Kurtz (1986, Chapter 10)) Assume $\mathcal{L} = \psi \times (\mathbb{1}\vec{\pi}^T - I)$. In the topology of convergence of compact sets, $(\vec{x}_t^\zeta)_{t\in[0,1)} \rightsquigarrow (\vec{z}_{\tau_t})_{t\in[0,1)}$.*

Note when $B = 2$, $(\vec{z}_t)_t$ is distributed as the Jacobi process described in Avdeyev et al. (2023).

When $B > 2$ Avdeyev et al. (2023) considers $B - 1$ parallel Wright-Fisher processes with $B = 2$; they then use a stick-breaking procedure to get an SDE on the simplex. This SDE is distinct to ours in Eqn. 5 and is not symmetric to the order of letters – it requires us to specify a first letter, second letter, and so on, which behave differently in paths $(x_t)_t$ – except for at stationary. We instead directly consider the multi-allelic Wright-Fisher from Ethier and Kurtz (1986, Chapter 10) which is invariant to permutations of letters in the alphabet and simplifies our derivations.

#### E.5.2 CONVERGENCE OF THE ELBO

Call $\vec{s}(\vec{v} \mid x_0) = \nabla \log p(z_t|x_0,t)|_{z_t=\vec{v}}$, and $\vec{s}(\vec{v} \mid \tilde{x}_0, t) = \sum_b \tilde{x}_{0,b}\vec{s}(\vec{v} \mid x_0 = b, t)$.

**Theorem E.5.** *(Proof of Thm 5.1) Call the ELBO in Alg. 1*

$$L^\zeta(\vec{x}^\zeta, t, x_0, \tilde{x}_0) = \sum_{\vec{x}_t'^\zeta \neq \vec{x}_t^\zeta} (\zeta\mathcal{L}^{\mathrm{wf}}_{\vec{x}_t'^\zeta \to \vec{x}_t^\zeta} + \mathcal{L}^{\mathrm{mut}}_{\vec{x}_t'^\zeta \to \vec{x}_t^\zeta})\dot{\tau}_t \mathbb{D}\left(\frac{p(\vec{x}_t'^\zeta \mid x_0, t)}{p(\vec{x}_t^\zeta \mid x_0, t)} \middle\| \sum_b \tilde{x}_{0b}\frac{p(\vec{x}_t'^\zeta \mid x_0 = b, t)}{p(\vec{x}_t^\zeta \mid x_0 = b, t)}\right).$$

*For $\vec{v}$ for which $\zeta\vec{v}$ are not integers, define $L(\vec{v}, t, x_0, \tilde{x}_0) = L^\zeta(\vec{x}^\zeta, t, x_0, \tilde{x}_0)$ for a $\vec{x}^\zeta$ nearest to $\vec{v}$. Then, for all $\vec{v}$ in the interior of $\Delta^B$, $t \in (0,1)$, $\tilde{x}_0 \in \Delta^B$, and $x_0$,*

$$L^\zeta(\vec{v}, t, x_0, \tilde{x}_0) \to \frac{\dot{\tau}_t}{2}\|\vec{s}(\vec{v} \mid x_0, t) - \vec{s}(\vec{v} \mid \tilde{x}_0, t)\|^2_{\mathrm{diag}\vec{v} - \vec{v}\vec{v}^T}$$

*Proof.* **Overview of proof:** For notational convenience, define

$$\mathbb{D}(\vec{x}_t'^\zeta) = \mathbb{D}\left(\frac{p(\vec{x}_t'^\zeta \mid x_0, t)}{p(\vec{x}_t^\zeta \mid x_0, t)} \middle\| \sum_b \tilde{x}_{0b}\frac{p(\vec{x}_t'^\zeta \mid x_0 = b, t)}{p(\vec{x}_t^\zeta \mid x_0 = b, t)}\right).$$

Much of the proof consists of checking uniform convergence and regularity conditions. The main idea however is that when $\zeta$ is very large, the transition rates $\zeta\mathcal{L}^{\mathrm{wf}}_{\vec{x}_t'^\zeta \to \vec{x}_t^\zeta} + \mathcal{L}^{\mathrm{mut}}_{\vec{x}_t'^\zeta \to \vec{x}_t^\zeta}$ are only large for $\vec{x}_t'^\zeta$ very close to $\vec{v}$. For those terms, we can perform a second order Taylor expansion

$$\mathbb{D}(\vec{x}_t'^\zeta) \approx \frac{1}{2}\|\vec{s}(\vec{v} \mid x_0, t) - \vec{s}(\vec{v} \mid \tilde{x}_0, t)\|^2_{(\vec{x}_t'^\zeta - \vec{v})(\vec{x}_t'^\zeta - \vec{v})^T}$$

so

$$L^\zeta(\vec{v}, t, x_0, \tilde{x}_0) \approx \frac{\dot{\tau}_t}{2}\|\vec{s}(\vec{v} \mid x_0, t) - \vec{s}(\vec{v} \mid \tilde{x}_0, t)\|^2_\Sigma$$

where $\Sigma = (\sum_{\vec{x}_t'^\varsigma \neq \vec{x}_t^\varsigma} \varsigma \mathcal{L}^{\text{wf}}_{\vec{x}_t'^\varsigma \to \vec{x}_t^\varsigma} + \mathcal{L}^{\text{mut}}_{\vec{x}_t'^\varsigma \to \vec{x}_t^\varsigma})(\vec{x}_t'^\varsigma - \vec{v})(\vec{x}_t'^\varsigma - \vec{v})^T$. Finally, we show $\sum_{\vec{x}_t'^\varsigma \neq \vec{x}_t^\varsigma} \mathcal{L}^{\text{mut}}_{\vec{x}_t'^\varsigma \to \vec{x}_t^\varsigma}(\vec{x}_t'^\varsigma - \vec{v})(\vec{x}_t'^\varsigma - \vec{v})^T \to 0$, and through a central limit theorem, $\sum_{\vec{x}_t'^\varsigma \neq \vec{x}_t^\varsigma} \varsigma \mathcal{L}^{\text{wf}}_{\vec{x}_t'^\varsigma \to \vec{x}_t^\varsigma}(\vec{x}_t'^\varsigma - \vec{v})(\vec{x}_t'^\varsigma - \vec{v})^T \to \operatorname{diag}\vec{v} - \vec{v}\vec{v}^T$.

Crucial to our proof is Lem. E.10 which states that for each $\vec{x}_t^\varsigma$ in the interior of the simplex,

$$p(\vec{x}_t^\varsigma \mid x_0, t) = \mathbb{E}_{m \sim A^{(\varsigma)}(\psi, \tau_t)} \mathbb{E}_{\vec{p} \sim \operatorname{Dir}(\psi\vec{\pi} + m\vec{x}_0)} \operatorname{Mult}(\varsigma, \vec{p})(\vec{x}_t^\varsigma)$$

for a distribution $A^{(\varsigma)}(\psi, \tau_t)$ such that $A^{(\varsigma)}(\psi, \tau_t)(m) \to A(\psi, \tau_t)(m)$ quickly for each $m \leqslant \varsigma$ as $\varsigma \to \infty$.

**Part 1: Eliminating the boundary** For a small $\epsilon > 0$, call $\Delta_\epsilon^B$ the points in $\Delta^B$ that have an entry less than $\epsilon$; in particular, define $\epsilon < (4B)^{-2/\min_b \vec{v}_b}$. We first show that the contribution form the epsilon-boundary vanishes, i.e.

$$E(\varsigma) = \sum_{\vec{x}_t'^\varsigma \notin \Delta_\epsilon^B} (\varsigma \mathcal{L}^{\text{wf}}_{\vec{x}_t'^\varsigma \to \vec{x}_t^\varsigma} + \mathcal{L}^{\text{mut}}_{\vec{x}_t'^\varsigma \to \vec{x}_t^\varsigma})\dot{\tau}_t \mathbb{D}(\vec{x}_t'^\varsigma) \to 0.$$

First note for large enough $\varsigma$, $\mathcal{L}^{\text{mut}}_{\vec{x}_t'^\varsigma \to \vec{x}_t^\varsigma} = 0$ for all $\vec{x}_t'^\varsigma \notin \Delta_\epsilon^B$ and

$$\mathcal{L}^{\text{wf}}_{\vec{x}_t'^\varsigma \to \vec{x}_t^\varsigma} \leqslant \binom{\varsigma}{\varsigma\vec{x}_t^\varsigma}(\min_b \vec{x}_t'^\varsigma)^{\min_b \varsigma\vec{x}_{t,b}^\varsigma(\vec{v})} \leqslant C(\epsilon^{2\min_b \vec{v}_b})^\varsigma < C(4B)^{-\varsigma}$$

for some $C > 0$. Also note for any $\vec{x}_t^\varsigma$,

$$1 \geqslant p(\vec{x}_t^\varsigma \mid x_0, t) \geqslant p(A(\psi, \tau_t) = 0)\mathbb{E}_{\vec{p} \sim \operatorname{Dir}(\psi\vec{\pi})} \operatorname{Mult}(\varsigma, \vec{p})(\vec{x}_t^\varsigma)$$

Taking the leading term of the divergence $\mathbb{D}(\vec{x}_t'^\varsigma)$,

$$E(\varsigma) \lesssim \sum_{\vec{x}_t'^\varsigma \notin \Delta_\epsilon^B} (4B)^{-\varsigma} \frac{-\log \mathbb{E}_{\vec{p} \sim \operatorname{Dir}(\psi\vec{\pi})} \operatorname{Mult}(\varsigma, \vec{p})(\vec{x}_t'^\varsigma)}{\mathbb{E}_{\vec{p} \sim \operatorname{Dir}(\psi\vec{\pi})} \operatorname{Mult}(\varsigma, \vec{p})(\vec{x}_t'^\varsigma)}.$$

Now $\operatorname{Mult}(\varsigma, \vec{p})(\vec{x}_t'^\varsigma) \geqslant (\min_b \vec{p}_b)^\varsigma$ so the denominator is $\geqslant P_{\vec{p} \sim \operatorname{Dir}(\psi\vec{\pi})}(\min_b \vec{p} \geqslant 1/2B)(2B)^{-\varsigma}$. Therefore

$$E(\varsigma) \lesssim (4B)^{-\varsigma} \sum_{\vec{x}_t'^\varsigma \notin \Delta_\epsilon^B} \frac{\varsigma \log(2B)}{(2B)^{-\varsigma}}$$
$$\lesssim 2^{-\varsigma}\varsigma \times \varsigma^{B-1}$$
$$\to 0$$

since there are $O(\varsigma^{B-1})$ elements with $\vec{x}_t'^\varsigma \notin \Delta_\epsilon^B$.

**Part 2: Uniform convergence of the likelihood** Next we show $\frac{p(\vec{x}_t^\varsigma(\vec{v})|x_0,t)}{p(\vec{z}_t = \vec{v}|x_0,t)} = 1 + O(\varsigma^{-1})$ uniformly in $\Delta_\epsilon^B$. While something like this is implied by the convergence of the process from previous work, the fast uniform convergence will be important for our results below.

We will do so by showing the same property for each of the quotients

$$\frac{\mathbb{E}_{m \sim A^{(\varsigma)}(\psi, \tau_t)} \mathbb{E}_{\vec{p} \sim \operatorname{Dir}(\psi\vec{\pi} + m\vec{x}_0)} \operatorname{Mult}(\varsigma, \vec{p})(\vec{x}_t^\varsigma(\vec{v}))}{\mathbb{E}_{m \sim A^{(\varsigma)}(\psi, \tau_t)} \operatorname{Dir}(\psi\vec{\pi} + m\vec{x}_0)(\vec{x}_t^\varsigma(\vec{v}))}, \frac{\mathbb{E}_{m \sim A^{(\varsigma)}(\psi, \tau_t)} \operatorname{Dir}(\psi\vec{\pi} + m\vec{x}_0)(\vec{x}_t^\varsigma(\vec{v}))}{\mathbb{E}_{m \sim A(\psi, \tau_t)} \operatorname{Dir}(\psi\vec{\pi} + m\vec{x}_0)(\vec{v})}.$$

The first quotient converges by the concentration of a Bayesian posterior (Miller, 2019). In particular, by the uniform Stirling approximation (Robbins, 1955) uniformly for $\vec{x}_t^\varsigma \in \Delta_{\epsilon/2}^B$,

$$\operatorname{Mult}(\varsigma, \vec{p})(\vec{x}_t^\varsigma) = (1 + O(\varsigma^{-1}))\left(\prod_b \vec{x}_b^\varsigma\right)^{-1/2}(2\pi\varsigma)^{-(B-1)/2}e^{-\varsigma\operatorname{KL}(\vec{x}_t^\varsigma \|\vec{p})}.$$

We'd like to write this as approximately a normal density with mean $\vec{x}^\zeta$ and variance restricted to vectors summing to $0$ $\{\vec{w} \in \mathbb{R}^B \mid \vec{w}^T \mathbb{1} = 0\}$. We can do so with a Taylor expansion; for $\vec{p}$ near $\vec{x}^\zeta$,

$$\mathrm{KL}(\vec{x}_t^\zeta || \vec{p}) = \frac{1}{2}\|\vec{x}_t^\zeta - \vec{p}\|_{\mathrm{diag}(\vec{x}_t^\zeta)^{-1}}^2 - O(\|\vec{x}_t^\zeta - \vec{p}\|^3).$$

We can also write $\|\vec{x}^\zeta - \vec{p}\|_{\mathrm{diag}(\vec{x}_t^\zeta)^{-1}}^2 = \|\vec{x}^\zeta - \vec{p}\|_{\Lambda^\dagger}^2$ where $\Lambda = \mathrm{diag}\,\vec{x}^\zeta - \vec{x}^\zeta \vec{x}^{\zeta T}$ has kernel orthogonal to vectors summing to $0$ and

$$\Lambda^\dagger = \mathrm{diag}(\vec{x}_t^\zeta)^{-1} - \frac{1}{B}\vec{x}_t^{\zeta-1}\mathbb{1}^T - \frac{1}{B}\mathbb{1}\vec{x}_t^{\zeta-1,T} + \frac{\sum_b \vec{x}_{t,b}^{\zeta-1}}{B^2}\mathbb{1}\mathbb{1}^T.$$

Note also that the pseudo-determinant of $\Lambda$ is $\prod_b \vec{x}_b^\zeta$ so we can write

$$\mathrm{Mult}(\zeta, \vec{p})(\vec{x}_t^\zeta) = (1 + O(\zeta^{-1}))(1 - O(\zeta\|\vec{x}^\zeta - \vec{p}\|^3))\mathcal{N}(\vec{x}_t^\zeta, \zeta^{-1}\Lambda).$$

This allows us to write

$$\frac{\mathbb{E}_{P(\vec{p})}\mathrm{Mult}(\zeta, \vec{p})(\vec{x}_t^\zeta)}{P(\vec{x}_t^\zeta)} = (1 + O(\zeta^{-1}))\frac{\mathbb{E}_{\vec{w}\sim\mathcal{N}(0,\Lambda)}P(\vec{x}_t^\zeta + \zeta^{-1/2}\vec{w})(1 - O(\|\vec{w}\|^3/\zeta^{1/2}))}{P(\vec{x}_t^\zeta)}.$$

For a small $\delta < \epsilon/4$ call $\phi$ a $C^\infty$ function with support in the $\delta$-ball, and which is $1$ in the $\delta/2$-ball. We break the numerator up into

$$\mathbb{E}_{\vec{w}\sim\mathcal{N}(0,\Lambda)}\phi(\zeta^{-1/2}\vec{w})P(\vec{x}_t^\zeta + \zeta^{-1/2}\vec{w})(1 - O(\|\vec{w}\|^3/\zeta^{1/2}))$$
$$+ \mathbb{E}_{\vec{w}\sim\mathcal{N}(0,\Lambda)}(1 - \phi(\zeta^{-1/2}\vec{w}))P(\vec{x}_t^\zeta + \zeta^{-1/2}\vec{w})(1 - O(\|\vec{w}\|^3/\zeta^{1/2})).$$

The second term is less than

$$\mathbb{E}_{P(\vec{p})}(1 - \phi(\vec{p} - \vec{x}_t^\zeta))\mathcal{N}(0,\Lambda)(\sqrt{\zeta}(\vec{x}_t^\zeta - \vec{p})) \leq \mathbb{E}_{P(\vec{p})}\mathbb{1}(\|\vec{x}_t^\zeta - \vec{p}\| > \delta/2)\mathcal{N}(0,\Lambda)(\sqrt{\zeta}(\vec{x}_t^\zeta - \vec{p}))$$
$$\lesssim e^{-\zeta C\delta^2}$$

for some $C$. For the first term, we can define $\tilde{P}(\vec{p}) = P(\vec{p})\phi(\vec{p} - \vec{x}_t^\zeta)$ which is a compactly supported $C^\infty$ function. Therefore

$$\mathbb{E}_{\vec{w}\sim\mathcal{N}(0,\Lambda)}\tilde{P}(\vec{x}_t^\zeta + \zeta^{-1/2}\vec{w})(1 - O(\|\vec{w}\|^3/\zeta^{1/2}))$$
$$= \tilde{P}(\vec{x}_t^\zeta) + \nabla\tilde{P}(\vec{x}_t^\zeta)^T\mathbb{E}_{\vec{w}\sim\mathcal{N}(0,\Lambda)}(\zeta^{-1/2}\vec{w} + O(\zeta\|w\|^2))(1 - O(\|\vec{w}\|^3/\zeta^{1/2}))$$
$$= P(\vec{x}_t^\zeta) + O(\zeta^{-1}).$$

For the second quotient, note the denominator is bounded below for $\vec{v} \in \Delta_\epsilon^B$. By Lem. E.10

$$\sup_{\vec{v}\in\Delta_\epsilon^B} |\mathbb{E}_{m\sim A^{(\varsigma)}(\psi,\tau_t)}\mathrm{Dir}(\psi\vec{\pi} + m\vec{x}_0)(\vec{x}_t^\zeta(\vec{v})) - \mathbb{E}_{m\sim A(\psi,\tau_t)}\mathrm{Dir}(\psi\vec{\pi} + m\vec{x}_0)(\vec{x}_t^\zeta(\vec{v}))|$$
$$\lesssim \zeta^{-1}\sum_m e^{-cm^2}\sup_{\vec{v}\in\Delta_{\epsilon/2}^B}\mathrm{Dir}(\psi\vec{\pi} + m\vec{x}_0)(\vec{v}).$$

Since $\sup_{\vec{v}\in\Delta_{\epsilon/2}^B}\mathrm{Dir}(\psi\vec{\pi} + m\vec{x}_0)(\vec{v}) \leq (m + \psi)^\psi(1 - \epsilon/2)^{m-1}$ is eventually decreasing in $m$, the whole quotient is $O(\zeta^{-1})$. Next note the derivative of $\mathbb{E}_{m\sim A(\psi,\tau_t)}\mathrm{Dir}(\psi\vec{\pi} + m\vec{x}_0)(\cdot)$ is bounded on the compact set $\Delta_\epsilon^B$ so

$$\sup_{\vec{v}\in\Delta_\epsilon^B} |\mathbb{E}_{m\sim A(\psi,\tau_t)}\mathrm{Dir}(\psi\vec{\pi} + m\vec{x}_0)(\vec{x}_t^\zeta(\vec{v})) - \mathbb{E}_{m\sim A(\psi,\tau_t)}\mathrm{Dir}(\psi\vec{\pi} + m\vec{x}_0)(\vec{v})|$$
$$= O(\|\vec{x}_t^\zeta(\vec{v}) - \vec{v}\|) = O(\zeta^{-1}).$$

**Part 3: Taylor expansion of the divergence**  Given the calculation above, for $\zeta$ large enough and any $\vec{x}_t'^\zeta = \vec{x}_t^\zeta(\vec{v}) + O(\zeta^{-1/2})$, we can approximate

$$\frac{p(\vec{x}_t'^\zeta \mid x_0, t)}{p(\vec{x}_t^\zeta(\vec{v}) \mid x_0, t)} = \exp\left(\log p\left(\vec{z}_t = \vec{x}_t'^\zeta \mid x_0, t\right) - \log p\left(\vec{z}_t = \vec{x}_t^\zeta(\vec{v}) \mid x_0, t\right)\right) + O(\zeta^{-1})$$
$$= 1 + \vec{s}(\vec{v} \mid x_0, t)^T(\vec{x}_t'^\zeta - \vec{v}) + O(\zeta^{-1}).$$

A second order Taylor expansion then gives

$$\mathbb{D}(\vec{x}_t'^\zeta) = \frac{1}{2}\left((\vec{s}(\vec{v} \mid x_0, t) - \vec{s}(\vec{v} \mid \tilde{x}_0, t))^T (\vec{x}_t'^\zeta - \vec{v})\right)^2 + o(\zeta^{-1})$$

$$= \frac{1}{2}\|\vec{s}(\vec{v} \mid x_0, t) - \vec{s}(\vec{v} \mid \tilde{x}_0, t)\|^2_{(\vec{x}_t'^\zeta - \vec{v})(\vec{x}_t'^\zeta - \vec{v})^T} + o(\zeta^{-1}).$$

Given the calculation above, we note that since $\mathcal{L}^{\text{mut}}_{\vec{x}_t'^\zeta \to \vec{x}_t^\zeta(\vec{v})}$ is only non-zero for $\zeta$ values of $\vec{x}_t'^\zeta$ each with $\vec{x}_t'^\zeta = \vec{x}_t^\zeta(\vec{v}) + O(\zeta^{-1})$,

$$\sum_{\vec{x}_t'^\zeta} \mathcal{L}^{\text{mut}}_{\vec{x}_t'^\zeta \to \vec{x}_t^\zeta} \mathbb{D}(\vec{x}_t'^\zeta) = O(\zeta \times \zeta^{-2}) = o(1).$$

This gives

$$L^\zeta(\vec{v}, t, x_0, \tilde{x}_0) = \frac{\dot{\tau}_t}{2}\|\vec{s}(\vec{v} \mid x_0, t) - \vec{s}(\vec{v} \mid \tilde{x}_0, t)\|^2_\Sigma + o(1)$$

where

$$\Sigma = \sum_{\vec{x}_t'^\zeta \in \Delta_\epsilon^B} \zeta \mathcal{L}^{\text{wf}}_{\vec{x}_t'^\zeta \to \vec{x}_t^\zeta}(\vec{x}_t'^\zeta - \vec{v})(\vec{x}_t'^\zeta - \vec{v})^T.$$

The proof is therefore finished if we show $\Sigma \to \text{diag}\,\vec{v} - \vec{v}\vec{v}^T$.

**Part 4: Convergence of $\Sigma$**   Note, by the uniform Stirling approximation (Robbins, 1955) uniformly for $\vec{x}'^\zeta \in \Delta_\epsilon^B \backslash \{\vec{x}_t^\zeta\}$, the infinitesimal generator approximates a Normal distribution near $\vec{v}$,

$$\mathcal{L}^{\text{wf}}_{\vec{x}'^\zeta \to \vec{x}^\zeta(\vec{v})} = (1 + o(1))\left(\prod_b \vec{v}_b\right)^{-1/2} (2\pi\zeta)^{-(B-1)/2} e^{-\zeta\text{KL}(\vec{v}\|\vec{x}'^\zeta)}$$

$$= (1 + o(1) + O(\zeta\|\vec{v} - \vec{x}'^\zeta\|^3))\mathcal{N}\left(\vec{v}, \zeta^{-1}(\text{diag}(\vec{v}) - \vec{v}\vec{v}^T)\right)(\vec{x}'^\zeta)$$

Noting that, by Pinsker's inequality, $\text{KL}(\vec{v}\|\vec{x}'^\zeta) \geqslant 2\|\vec{v} - \vec{x}'^\zeta\|^2_1 \geqslant \frac{2}{B}\|\vec{v} - \vec{x}'^\zeta\|^2$, for some very small $\delta > 0$

$$\|\Sigma - (\text{diag}\,\vec{v} - \vec{v}\vec{v}^T)\| \lesssim \sum_{\vec{x}_t'^\zeta \in \Delta_\epsilon^B, \|\vec{x}_t'^\zeta - \vec{v}\| > \zeta^{-1/3-\delta}} \zeta^{-(B-1)/2+1} e^{-\zeta\text{KL}(\vec{v}\|\vec{x}'^\zeta)}$$

$$\lesssim \zeta^{-(B-1)/2+B} e^{-\frac{2}{B}\zeta\zeta^{-2/3-2\delta}}$$

$$= o(1)$$

$\square$

## E.6   Wright-Fisher loss calculations

See the discussion above Prop. C.1 for definitions.

**Proposition E.6.** *(Proof of Prop. C.1)*

$$p(\vec{x}_t \mid x_0, t) = \text{Dirichlet}(\pi\psi)(\vec{x}_t)G_\psi(\tau_t, x_0, \vec{x}_t).$$

*For $\vec{c}(\vec{x}_t) = \nabla \log \text{Dirichlet}(\pi\psi)(\vec{x}_t)$ which does not depend on $x_0$,*

$$\vec{s} = \vec{s}(\vec{x}_t \mid x_0, t) = \vec{c}(\vec{x}_t) + \vec{x}_0 w(x_0)$$

*where*

$$w(x_0) = \frac{e^{-\psi\tau_t/2}(\psi + 1)}{\pi(x_0)}\frac{F_\psi(\tau_t, x_0, \vec{x}_t)}{G_\psi(\tau_t, x_0, \vec{x}_t)}.$$

*Proof.* For $m_t \sim A(\psi, \tau_t)$,

$$p(\vec{x}_t \mid x_0, t) = \mathbb{E}_{m_t}\text{Dirichlet}(\psi\pi + m_t x_0)(\vec{x}_t)$$

$$= \prod_{b \neq x_0} \vec{x}_{t,b}^{\psi\pi_b - 1} \mathbb{E}_{m_t} \frac{\Gamma(\psi + m_t)}{\Gamma(\psi\pi_{x_0} + m_t)\prod_{b \neq x_0}\Gamma(\psi\pi_b)}\vec{x}_{t,x_0}^{\psi\pi_{x_0} + m_t - 1}$$

$$= \frac{\Gamma(\psi)}{\prod_{b \in \mathcal{B}}\Gamma(\psi\pi_b)}\prod_{b \in \mathcal{B}}\vec{x}_{t,b}^{\psi\pi_b - 1}\mathbb{E}_{m_t}\frac{\Gamma(\psi\pi(x_0))\Gamma(\psi + m_t)}{\Gamma(\psi)\Gamma(\psi\pi_{x_0} + m_t)}\vec{x}_{t,x_0}^{m_t}$$

$$= \text{Dirichlet}(\psi\pi)(\vec{x}_t)\mathbb{E}_{m_t}\frac{(\psi)_{(m_t)}}{(\psi\pi(x_0))_{(m_t)}}\vec{x}_{t,x_0}^{m_t}.$$

From Eqn. 5.2 of Tavaré (1984), we have

$$p(m_t = j) = \sum_{k=j}^{\infty} e^{-k(k+\psi-1)\tau_t/2}(-1)^k(-1)^j \frac{(2k+\psi-1)(j+\psi)_{(k-1)}}{j!(k-j)!}.$$

He wrote, in Eqn. A5,

$$\sum_{j=1}^{\infty} x^j p(m_t = j)$$

$$= \sum_{k=1}^{\infty} e^{-k(k+\psi-1)\tau_t/2}(-1)^k(2k+\psi-1) \sum_{j=1}^{k} \frac{x^j}{j!} \frac{(j+\psi)_{(k-1)}}{(k-j)!(-1)^j}$$

$$= \sum_{k=1}^{\infty} e^{-k(k+\psi-1)\tau_t/2}(-1)^k(2k+\psi-1) \sum_{j=1}^{k} \frac{x^j}{j!} \frac{(\psi)_{(j+k-1)}(-k)_{(j)}}{k!\psi_{(j)}}$$

$$= \sum_{k=1}^{\infty} e^{-k(k+\psi-1)\tau_t/2} \frac{(-1)^k(2k+\psi-1)(\psi)_{(k-1)}}{k!} \sum_{j=1}^{k} \frac{x^j}{j!} \frac{(\psi+k-1)_{(j)}(-k)_{(j)}}{\psi_{(j)}}.$$

The last sum is then written as $_2F_1(-k, \psi+k-1; \psi; x) - 1$ for the hyper-geometric function $_2F_1$. A very simple extension gives us

$$\sum_{j=1}^{\infty} \frac{(\psi)_{(j)}}{(\psi\pi_{x_0})_{(j)}} x^j p(m_t = j) = \sum_{k=1}^{\infty} e^{-k(k+\psi-1)t/2} \frac{(-1)^k(2k+\psi-1)(\psi)_{(k-1)}}{k!}$$

$$\times \left( _2F_1(-k, \psi+k-1; \psi\pi_{x_0}; x) - 1 \right).$$

Including the $j = 0$ term, by Eqn 5.3 of Tavaré (1984), cancels out the $-1$ in the brackets above, so our expectation

$$E_{m_t} \frac{(\psi)_{(m_t)}}{(\psi\pi_{x_0})_{(m_t)}} \vec{x}_{t,x_0}^{m_t} = 1 + \sum_{k=1}^{\infty} e^{-k(k+\psi-1)\tau_t/2} \frac{(-1)^k(2k+\psi-1)(\psi)_{(k-1)}}{k!}$$

$$\times {}_2F_1(-k, \psi+k-1; \psi\pi_{x_0}; \vec{x}_{t,x_0})$$

$$= G_{\psi}(t, x_0, \vec{x}_t).$$

Finally, using identities of the hypergeometric function,

$$\nabla_{\vec{x}_{t,x_0}} G_{\psi}(t, x_0, \vec{x}_t) = \sum_{k=1}^{\infty} e^{-k(k+\psi-1)\tau_t/2} \frac{(-1)^k(2k+\psi-1)(\psi)_{(k-1)}}{k!} \frac{-k(\psi+k-1)}{\psi\pi_{x_0}}$$

$$\times {}_2F_1(-k+1, \psi+k; \psi\pi_{x_0}+1; \vec{x}_{t,x_0})$$

$$= \frac{1}{\psi\pi_{x_0}} \sum_{k=1}^{\infty} e^{-k(k+\psi-1)\tau_t/2} \frac{(-1)^{k-1}(2k+\psi-1)(\psi+k-1)(\psi)_{(k-1)}}{(k-1)!}$$

$$\times {}_2F_1(-k+1, \psi+k; \psi\pi_{x_0}+1; \vec{x}_{t,x_0})$$

$$= \frac{1}{\psi\pi_{x_0}} \sum_{k=0}^{\infty} e^{-(k+1)(k+\psi)\tau_t/2} \frac{(-1)^k(2k+\psi+1)(\psi+k)(\psi)_{(k)}}{k!}$$

$$\times {}_2F_1(-k, \psi+k+1; \psi\pi_{x_0}+1; \vec{x}_{t,x_0})$$

$$= \frac{e^{-\psi t/2}(\psi+1)}{\pi_{x_0}} \sum_{k=0}^{\infty} e^{-k(k+\psi+1)\tau_t/2} \frac{(-1)^k(\psi)_{(k)}}{k!} \frac{(2k+\psi+1)(\psi+k)}{(\psi+1)\psi}$$

$$\times {}_2F_1(-k, \psi+k+1; \psi\pi_{x_0}+1; \vec{x}_{t,x_0})$$

$$=: \frac{e^{-\psi t/2}(\psi+1)}{\pi_{x_0}} F_{\psi}(t, x_0, \vec{x}_t).$$

$\square$

## E.7 PROOF OF SUFFICIENT STATISTICS

**Proposition E.7.** *(Proof of Prop. 6.1) There is a function $F^d$,* **depending on $p(x_0)$ and not on the** **diffusion process or** *$t$, such that*

$$p(x_0^d \mid x_t^{-d}, t) = F^d(\vec{\phi}(\vec{x}_t^1, t), \dots, \vec{\phi}(\vec{x}_t^D, t)).$$

*Proof.*

$$p(x_0^d \mid x_t^{-d}) = \int p(x_0^d \mid x_0^{-d}) dp(x_0^{-d} \mid x_t^{-d})$$

$$= \frac{1}{p(x_t^{-d})} \int p(x_0^d \mid x_0^{-d}) p(x_t^{-d} \mid x_0^{-d}) dp(x_0^{-d})$$

$$= \frac{1}{p(x_t^{-d})} \int p(x_0^d \mid x_0^{-d}) \prod_{d' \neq d} p(x_t^{d'} \mid x_0^{d'}) dp(x_0^{-d})$$

$$= \frac{\prod_{d' \neq d} \sum_b p(x_t^{d'} \mid x_0^{d'} = b)}{p(x_t^{-d})} \int p(x_0^d \mid x_0^{-d}) \prod_{d' \neq d} \frac{p(x_t^{d'} \mid x_0^{d'})}{\sum_b p(x_t^{d'} \mid x_0^{d'} = b)} dp(x_0^{-d})$$

$$= E_{p(x_0^{-d})} \left( p(x_0^d \mid x_0^{-d}) \prod_{d' \neq d} \vec{\phi}(x_t^{d'})_{x_0^{d'}} \right) \Big/ E_{p(x_0^{-d})} \left( \prod_{d' \neq d} \vec{\phi}(x_t^{d'})_{x_0^{d'}} \right),$$

$\square$

## E.8 LEMMAS

Our first lemma establishes conditions for convergence of paths using standard techniques inspired by arguments used throughout Ethier and Kurtz (1986) or Bass (2011) for example.

**Lemma E.8.** *Say $(\vec{x}_t^\zeta)_{t \in (0,1)}$ are Markov processes on $\mathbb{R}^r$ for $\zeta = 1, 2, \dots$ and $(\vec{z}_t)_{t \in (0,1)}$ is another Markov process on $\mathbb{R}^r$. Say the following conditions are satisfied*

1. *(Convergence of marginals) $\vec{x}_t^\zeta \rightsquigarrow \vec{z}_t$ for each $t$.*

2. *(Local uniform convergence of conditionals) Conditional distributions exist such that for each $\vec{v} \in \mathbb{R}^r$, $s < t$, and bounded compactly supported measurable function $f$, there is an $\epsilon > 0$, such that*

$$\sup_{\|\vec{w} - \vec{v}\| < \epsilon} |\mathbb{E}_{\vec{x}_t^\zeta \mid \vec{x}_s^\zeta = \vec{w}} f - \mathbb{E}_{\vec{z}_t \mid \vec{z}_s = \vec{w}} f| \to 0.$$

3. *(Tightness) For every $[a, b] \subset (0, 1)$, there are $\beta, \theta, M > 0$ such that for all $s, t \in [a, b]$, $\sup_{\zeta > M} \mathbb{E}\|\vec{x}_s^\zeta - \vec{x}_t^\zeta\|^\beta < C(s - t)^\theta$.*

*Then, with the topology of convergence on compact sets[11], the paths converge in distribution*

$$(\vec{x}_t^\zeta)_{t \in (0,1)} \rightsquigarrow (\vec{z}_t)_{t \in (0,1)}.$$

*Proof.* Pick a compact set $[a, b] \subset (0, 1)$. We show $(\vec{x}_t^\zeta)_{t \in [a,b]} \rightsquigarrow (\vec{z}_t)_{t \in [a,b]}$. Say $(\vec{x}_t^{\zeta_m})_{t \in [a,b]}$ is a subsequence which doesn't enter a neighbourhood of $(\vec{z}_t)_{t \in [a,b]}$; we'll now show a contradiction. By Prokhorov's theorem, since it's tight by Assumption 3 and Thm. 8.8 of Ethier and Kurtz (1986, Chapter 3), it has a subsequence which converges to a process $(\vec{y}_t)_{t \in [a,b]}$. As we'll show below, for every set $a \leqslant t_1 < t_2 < \cdots < t_m \leqslant b$, $(\vec{y}_t)_{t \in \{t_i\}_{i=1}^m} = (\vec{z}_t)_{t \in \{t_i\}_{i=1}^m}$. This must mean $(\vec{y}_t)_t = (\vec{z}_t)_t$ by the Kolmogorov extension theorem, a contradiction.

What remains is to show, for $a \leqslant t_1 < t_2 < \cdots < t_m \leqslant b$, $(\vec{x}_t^\zeta)_{t \in \{t_i\}_{i=1}^m} \rightsquigarrow (\vec{z}_t)_{t \in \{t_i\}_{i=1}^m}$. It is sufficient to prove that for any $t_1 < \cdots < t_m$ and compactly supported continuous function on $\mathbb{R}^r$, $h$,

$$Eh(\vec{x}_1^\zeta, \dots, \vec{x}_m^\zeta) \to Eh(\vec{z}_1, \dots, \vec{z}_m). \tag{6}$$

---

[11]This is a standard topology for these results. See for example Thm 1.1 of Ethier and Kurtz (1986, Chapter 10).

By the Stone-Weierstrass theorem, each such $h$ can be arbitrarily well approximated by product of $m$ univariate functions, so it is sufficient to consider $h(\vec{z}_1, \ldots, \vec{z}_m) = \prod_{i=1}^{m} h_i(\vec{z}_i)$. Finally, by the Markov property,

$$\mathbb{E}h(\vec{x}_1^\zeta, \ldots, \vec{x}_m^\zeta) = \mathbb{E}_{\vec{x}_1^\zeta | \vec{x}_0^\zeta} h_1(\vec{x}_1^\zeta) \mathbb{E}_{\vec{x}_2^\zeta | \vec{x}_1^\zeta} h_2(\vec{x}_2^\zeta) \cdots \mathbb{E}_{\vec{x}_m^\zeta | \vec{x}_{m-1}^\zeta} h_m(\vec{x}_m^\zeta).$$

We can call $\tilde{h}_{m-1}^\zeta(\vec{x}_{m-1}^\zeta) = h_m(\vec{x}_{m-1}^\zeta) E_{\vec{x}_m^\zeta | \vec{x}_{m-1}^\zeta} h_m(\vec{x}_m^\zeta)$. By Assumption 2 $\tilde{h}_{m-1}^\zeta(\vec{x}_{m-1}^\zeta)$ converges uniformly to $h_m(\vec{x}_{m-1}^\zeta) E_{\vec{z}_m | \vec{z}_{m-1} = \vec{x}_{m-1}^\zeta} h_m(\vec{z}_m)$, a bounded function with compact support. Therefore, to prove Eqn. 6 it is sufficient to show the result replacing $h$ with $h_1 \times h_2 \times \cdots \times h_{m-2} \times \tilde{h}_{m-1}$. By induction, we reach $h = \tilde{h}_1$ for which we get Eqn. 6 by Assumption 1. $\qquad\square$

Our next Lemma is a non-asymptotic bound on the convergence of multinomials to Normal distributions. It states that as long as $\zeta \to \infty$ and the probabilities don't get too low, we can bound the expectation of a function by $O(\zeta^{-1/2})$.

**Lemma E.9.** *Let $Y_\zeta \sim Mult(\zeta, \vec{p})$ for probability vector $\vec{p} \in \mathbb{R}^B$ with $\min_i p_i \geqslant c > 0$. Call $Z_\zeta = \zeta^{-1/2}(Y_\zeta - \zeta p)$. For any bounded measurable function $f$,*

$$|\mathbb{E}f(Z_\zeta) - \mathbb{E}f(Z)| = o_{c,B,f}(1)$$

*where $Z \sim \mathcal{N}(0, \operatorname{diag}(\vec{p}) - \vec{p}\vec{p}^T)$ and the rate of decay $o_{c,B,f}(1)$ only depends on $c$, $B$, and $f$.*

*Proof.* For every $\epsilon$, pick a compactly supported $C^\infty$ function $g_\epsilon$ such that $\|g_\epsilon - f\|_\infty < \epsilon/2$, so

$$|\mathbb{E}f(Z_\zeta) - \mathbb{E}f(Z)| = \epsilon + |\mathbb{E}g_\epsilon(Z_\zeta) - \mathbb{E}g_\epsilon(Z)| = \epsilon + o_{c,B,g_\epsilon}(1)$$

by Thm 1.3 of Gotze (1991). $\qquad\square$

Our final lemma characterizes the distribution of the finite population Wright-Fisher process as described in Sec. 5 and App. C.

**Lemma E.10.** *For each $\vec{x}_t^\zeta$ in the interior of the simplex,*

$$p(\vec{x}_t^\zeta \mid x_0, t) = \mathbb{E}_{m \sim A^{(\zeta)}(\psi, \tau_t)} \mathbb{E}_{\vec{p} \sim \operatorname{Dir}(\psi\vec{\pi} + m\vec{x}_0)} \operatorname{Mult}(\zeta, \vec{p})(\vec{x}_t^\zeta)$$

*for a distribution over the natural numbers $A^{(\zeta)}(\psi, \tau_t)$ supported on $\{1, \ldots, \zeta\}$ such that $|A^{(\zeta)}(\psi, \tau_t)(m) - A(\psi, \tau_t)(m)| = C\zeta^{-1}\exp(-C'm^2)$ for constants $C, C'$ only depending on $\psi, \tau_t$, each $m$.*

*Proof.* This is standard in the population genetics literature. Define $A^{(\zeta)}(\psi, \tau_t)(m)$ the probability that $m$ alleles survive backwards in the coalescent of population $\zeta$ up to time $\tau_t$. Conditioned on observing $m$ individuals with allele $x_0$, Hoppe (1984) showed that sampling more individuals from the population is equivalent to sampling from a Pólya urn with allele probabilities $\psi\vec{\pi} + m\vec{x}_0$, giving the Dirichlet multinomials.

Tavaré (1984) shows $A(\psi, \tau_t)(m) = \lim_{\zeta \to \infty} A^{(\zeta)}(\psi, \tau_t)(m)$ and for $m > 0$

$$A^{(\zeta)}(\psi, \tau_t)(m) = \sum_{k=m}^{\zeta} e^{-k(k+\psi-1)\tau_t/2}(-1)^{k-m} \frac{(2k+\psi-1)(m+\psi)_{(k-1)}}{m!(k-m)!} \frac{(\zeta-k+1)_{(k)}}{(\zeta+\psi)_{(k)}}.$$

Note

$$\frac{(m+\psi)_{(k-1)}}{(k-m)!} \leqslant \frac{(m+\psi)_{(k-1)}}{(k-1)!} \frac{(k-1)!}{(k-m)!} \leqslant k^{m+\psi}(m+\psi)^c k^m$$

for some $c > 0$ and

$$\left| \frac{(\zeta-k+1)_{(k)}}{(\zeta+\psi)_{(k)}} - 1 \right| \lesssim k^2/\zeta.$$

Therefore

$$|A^{(\zeta)}(\psi, \tau_t)(m) - A(\psi, \tau_t)(m)| \lesssim \sum_{k=m}^{\infty} e^{-k(k+\psi-1)\tau_t/2}(2k + \psi - 1)k^{2m+\psi}\frac{m^c}{m!}\left(\frac{k^2}{\zeta} \wedge 1\right)$$

$$\lesssim \zeta^{-1}\frac{m^c}{m!}\sum_{k=m}^{\infty} e^{-k(k+\psi-1)\tau_t/2}k^{2m+\psi+3}$$

$$\leqslant \zeta^{-1}\frac{m^c}{m!}e^{-m(m+\psi-1)\tau_t/2}\sum_{j=0}^{\infty} e^{-j(m+\psi-1)\tau_t/2}(j + m)^{2m+\psi+3}$$

and

$$\sum_{j=0}^{\infty} e^{-j(m+\psi-1)\tau_t/2}(j + m)^{2m+\psi+3} \leqslant \sum_{j=0}^{m}(j + m)^{2m+\psi+3}$$

$$+ \sum_{j=m}^{\infty} e^{-j(m+\psi-1)\tau_t/2}(j + m)^{2m+\psi+3}$$

$$\leqslant m(2m)^{2m+\psi+3} + \sum_{j=m}^{\infty} e^{-j(m+\psi-1)\tau_t/2}(2j)^{2m+\psi+3}$$

$$\leqslant (2m)^{2m+\psi+4} + 2^{2m+\psi+3}\sum_{j=0}^{\infty} e^{-j(m+\psi-1)\tau_t/2}j^{2m+\psi+3}$$

$$= (2m)^{2m+\psi+4} + 2^{2m+\psi+3}\sum_{j=1}^{\infty} e^{-(j-\frac{4}{\tau_t}\log j)(m+\psi-1)\tau_t/2}j^{5-\psi}$$

$$\leqslant (2m)^{2m+\psi+4} + 2^{2m+\psi+3}\sum_{j=1}^{\infty} e^{-(j-\frac{4}{\tau_t}\log j)(\psi-1)\tau_t/2}j^{5-\psi}$$

$$\lesssim (2m)^{2m+\psi+4} + 2^{2m+\psi+3}.$$

$$\square$$

## F   EXPERIMENTAL DETAILS

### F.1   DNA

We describe the experiments in Sec. 5.2.

**Training and data**   For all DNA models, we use the same base CNN model and optimizer hyper-parameters used to train DDSM (Avdeyev et al., 2023) and Dirichlet flow matching (Stark et al., 2024) with code from `https://github.com/jzhoulab/ddsm` used with compliance with their licence and `https://github.com/HannesStark/dirichlet-flow-matching` used with an MIT licence. We train our Wright-Fisher simplicial model on the FlyBrain enhancer data from `https://zenodo.org/records/10184648`.

We trained our model on an A100 80GB GPU over 11 h for 700 epochs like Avdeyev et al. (2023). We trained a DDSM model using the code in `https://github.com/jzhoulab/ddsm` and used a pre-trained flow-matching model from `https://github.com/HannesStark/dirichlet-flow-matching`.

**Computational comparison**   All three models we tested need to pass their noisy $\vec{x}_t$ through a neural network. We chose the same neural network for our diffusion model as used in (Avdeyev et al., 2023) and (Stark et al., 2024). For a reasonably sized model, like the ESM model for protein experiments, the neural network computations took 75% of our compute time on average, meaning the overhead from sampling and loss computations cannot be more than 25%.

However the DNA architecture was very small, at only 3 M parameters. For the DNA setting then we precomputed and cached $\vec{x}_t$, $F_\psi$ and $G_\psi$ so that a majority of training time would come from the neural network. Indeed our model took 3 hours on an A100 to train for 200 epochs, comparable to 7 hours on an A6000 for 200 epochs in Stark et al. (2024).

**DNA accessibility (ATAC) predictor** To get accurate predictions of a position-resolution epigenetic marker for DNA-accessibility (a property one often wants to design), we use the CNN bpAITAC model from Chandra et al. (2025) to predict chromatin accessibility traces with code from `https://github.com/nuriachandra/bpAITAC`. The model is trained on embryonic drosophila chromatin accessibility from the Calderon et al. (2022) developmental fly dataset 16-20 hour subset, with ATAC-seq reads combined across cell types, using held-out chromosome chr2L for validation. bpAITAC was trained on a single NVIDIA TITAN RTX GPU (24GB) with early stopping based on validation loss.

bpAITAC produces two outputs: 1) total counts (a measure of regional accessibility), and 2) probability distribution of the counts. The base-pair resolution counts prediction is easily computed by multiplying the two outputs. The resulting per-base counts are modelled by a Poisson distribution. That is, `bpAITAC` takes a one-hot-encoded sequence $x_0$ of length $D = 500$ and predicts a positive 250-dimensional vector that represents the predicted "accessibility-profile" in the centre 250 positions of the sequence. For a target profile of 250 numbers, $\vec{y} \in \mathbb{N}^{250}$, we compute the probability by using the `bpAITAC` predictions as means of independent Poisson distributions

$$p(\vec{y}|x_0) = \prod_{d=1}^{D} \text{Poisson}\,(\vec{v}_d)\,(\vec{y}_d)$$

where $\vec{v} = $ `bpAITAC`$(x_0)$ is the output of the predictor. Since `bpAITAC` is a neural network which accepts one-hot-encoded $x_0 t$, we may also pass $x_0$ which have each position $x_{0,d}$ lying on the simplex.

**Evaluation** We use the `ode_likelihood` function to evaluate the likelihood of the trained diffusion model in the code of Avdeyev et al. (2023).

We collected 100 trace predictions from Calderon et al. (2022) validation chromosome chr2L to use as targets, picking the 100 peaks with the highest combined signal. For each target and model we sampled 10 conditional samples using 1000 function evaluations. We sampled from our simplicial diffusion model using the procedure described in App. C.4. To sample from the flow matching model in Stark et al. (2024), we modified the `get_cls_score` function in their code to return the one-step predictor that we used in our App C.4. Finally, we write custom code based on reversing an SDE to sample from the simplicial diffusion model in Avdeyev et al. (2023). To do so, we note they perform diffusion in a space with each position $\vec{v}_{t,d} \in [0,1]^{B-1}$. We compute their prediction $\tilde{x}_0(\vec{v}_t)$ by transforming the output of their neural network and then compute a prediction of $\nabla_{\vec{v}_t} \log p(y|\tilde{x}_0(\vec{v}_t))$ with a one-step estimator as in in our App C.4, and add it to their score for $\vec{v}$ every step. We add this modification into their function `Euler_Maruyama_Sampler`.

To calculate $\tilde{x}_0(\vec{v}_t)$ we note they build a neural network to predict $\vec{s} = \nabla_{\vec{v}_t} \log p(\vec{v}_t)$ which equals

$$\sum_b \tilde{x}_{0,b} \nabla_{\vec{v}_t} \log p(\vec{v}_t|x_0 = b)$$

for some implicit prediction $\tilde{x}_{0,b}$ which we must solve for. Now note, by the choice of the reverse stick-breaking procedure of Avdeyev et al. (2023), $\hat{U}_{b,b'} := (\nabla_{\vec{v}_t} \log p(\vec{v}_t|x_0 = b))_{b'} = \nabla_{v_{t,b'}} \log p(v_{t,b'}|v_{0,b'} = \delta_{b,b'})$ for $b' \leqslant b$ and $(\nabla_{\vec{v}_t} \log p(\vec{v}_t|x_0 = b))_{b'} = \nabla_{\vec{v}_{t,b'}} \log \text{Beta}(1, B - b')(\vec{v}_{t,b'})$ otherwise. So, $\vec{s} = U\tilde{x}_0 = U_{:,:-1}\tilde{x}_{0,:-1} + U_{:,-1}(1 - \tilde{x}_{0,:-1}^T \mathbb{1}) = (U_{:,:-1} - U_{:,-1}\mathbb{1}^T)\tilde{x}_{0,:-1} + U_{:,-1}$. Therefore we can solve for $\tilde{x}_{0,:-1}$ by solving this linear system.

## F.2 Protein

We describe the protein experiments in Sec. 6.2.

**Training and data** For all protein models, we started from pre-trained ESM2 150M weights (Lin et al., 2023) under an MIT license as in MDLM (Wang et al., 2024a). We trained with a learning rate of $10^{-5}$ for an A100 80GB GPU over 48 h for 3 million sequences, substantially less than the training budget of Wang et al. (2024a). We trained on UniRef50 (Suzek et al., 2007) data from `https://zenodo.org/records/6564798`.

**Evaluation** From each model we sampled 1000 sequences of length 200. We used a uniform grid of 100 points and integrated backwards, and we applied 4 corrector steps per predictor step as described in Campbell et al. (2022). Then we predicted pLDDTs of sequences with Omegafold Wu et al. (2022) under the Apache-2.0 License, with 1 cycle for each sequence.

### F.3 LANGUAGE

We describe the language experiments in Sec. 6.2.

**Training and data** We used the same architecture and training settings as Lou et al. (2023), using their code at `https://github.com/louaaron/Score-Entropy-Discrete-Diffusion` under an MIT license. We trained our model on 4 A100 80GB GPUs for between 40 and 50 hours total on 33 billion tokens taken from the `lm1b` dataset. We used a learning rate of $3 \times 10^{-4}$ and an EMA of 0.9999. Our diffusion transformer had an embedding dimension of 768 with 12 layers and 12 attention heads. The Gaussian models used pre-trained BERT embeddings scaled by a factor of 8.

For our individual discrete and Gaussian language models, each device used a physical batch size of 64 and took 2 gradient accumulation steps for an overall batch size of 512. For our unified model, we accumulated over Gaussian and discrete batches to get an overall batch size of 1024.

**Evaluation** We sample using 1000 iterations. Following Lou et al. (2023), we evaluate the sample quality of our models through the generative perplexity of their unconditional samples according to GPT2-large.

## G SUPPLEMENTARY EXPERIMENTS

### G.1 ANTIBODY OPTIMIZATION DOWNSTREAM TASK

We test our unified models from Sec. 6.2 on a different downstream task. Hie et al. (2023) suggested that generative protein models can be used to suggest mutations that improve the stability of antibody sequences. To test our diffusion model's ability to successfully improve antibody properties, we perturb a parental VHH sequence by noising a UniRef50-trained diffusion model by $t$ then denoising with 128 steps.[12] To emulate a realistic wet lab setting, we investigate sampling 50 unique single- and double-point mutants of the seed VHH by rejection sampling. We repeat this process 100 times, selecting the top resulting sequence from each repeat "experiment" according to a proprietary thermostability oracle. The amount of noising for each of the individual Gaussian, simplicial, and discrete diffusion models was determined by a hyperparameter sweep over $t \in [0.01, 0.02, 0.05, 0.1, 0.2, 0.5]$, where the chosen hyperparameter gave the most unique sequences with fewer than or equal to 5 mutations to the parental sequence. This hyperparameter was then shared with each sub-model of the unified model.

The thermostability oracle we used is an ensemble of 10 CARP/ByteNet regressors (Yang et al., 2024), pretrained on approximately 537,000 sequences from phage display, processed using Next Generation Sequencing (NGS), and 9556 $T_m$ datapoints obtained from NanoDSF. The resulting ensemble achieved a test cross-validated Spearman correlation of 0.72.

In Fig. 10 we see that unification does not substantially harm performance on this downstream task.

### G.2 FITTING IMAGE DATA: MNIST

We perform the analysis of Fig. 7 for image data and find a similar result. We evaluate our unified discrete diffusion framework on the MNIST dataset, consisting of 28x28 grayscale images. We discrete the pixel intensities to $N = 8$ levels using uniform quantization, preserving the continuous structure of the token identities while reducing the computational cost. We compare the performance of our single unified model (SSP) to the performance of three individually-trained diffusion models: discrete, simplicial, and Gaussian.

All models use a U-Net backbone with an embedding size of 128, 4 downsampling/upsampling blocks, and ReLU activations. Models are trained with the Adam optimizer with learning rate 0.001 and batch size 128 for 20 epochs. For Gaussian diffusion, we map each class index $x \in \{0, \cdots, C\}$ to a 2D continuous embedding with a circular parameterization $\mathrm{emb}(x) = (\cos(\theta), \sin(\theta))$ where $\theta = \frac{x}{C-1}\pi$. This embeddings encodes the similarity of different pixel values and ensures that the resulting diffusion process closely resembles continuous diffusion. We found models with a 1-D parameterization $\mathrm{emb}(x) = 2 \times (\frac{x}{C-1}) - 1$ performed much worse.

We evaluate the model performance using validation likelihood, as shown in Figure 11. First, as in Fig. 7 we find that the likelihoods between the unified model are competitive with the individually

---

[12]Note that this follows established methods for ML-based antibody diversification as in Raghu et al. (2025).

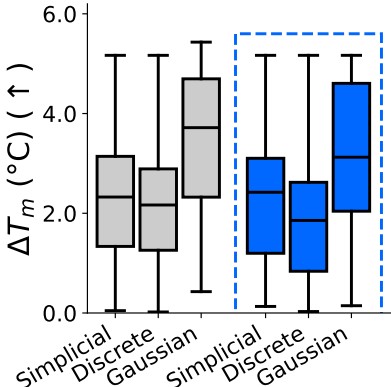

Figure 10: **The sufficient statistic parametrization enables a single model to perform competitive discrete, Gaussian, and simplicial optimization of antibodies.** Using our protein models from Fig. 7, we "denoise" antibody sequences and plot the predicted improvement in melting temperature in libraries of size 100.

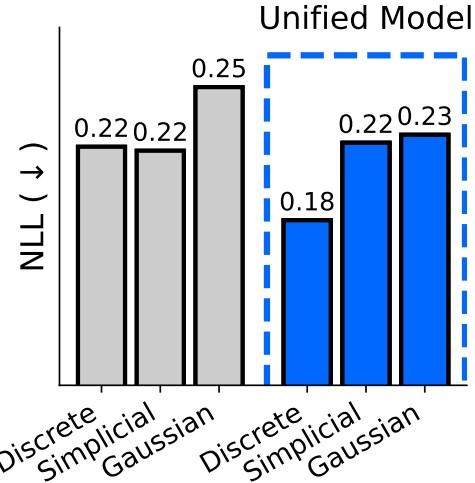

Figure 11: **The SSP enables a single model to fit image data across 3 modalities.** We perform the analysis of Fig. 7 for image data and find a similar result on MNIST.

trained models. In fact, we are even able to achieve slightly better performance for discrete and Gaussian diffusion, perhaps because the parameterization is easier to learn from, or because of a benefit from learning on diverse data.

As well, while we might expect Gaussian diffusion to achieve the best data fit due to the continuous nature of the data, we see the opposite: among our individual models, Gaussian surprisingly achieves the worst likelihood. This demonstrates the importance of considering multiple types of diffusion paradigms depending on the downstream tasks, thereby motivating our approach of training a single unified model.

We also generate 64 unconditional samples per model using 1,000 steps of ancestral sampling. Through our visualizations in Figure 12, we see that the unified model does not lead to a noticeable drop in sample quality compared to individual models.

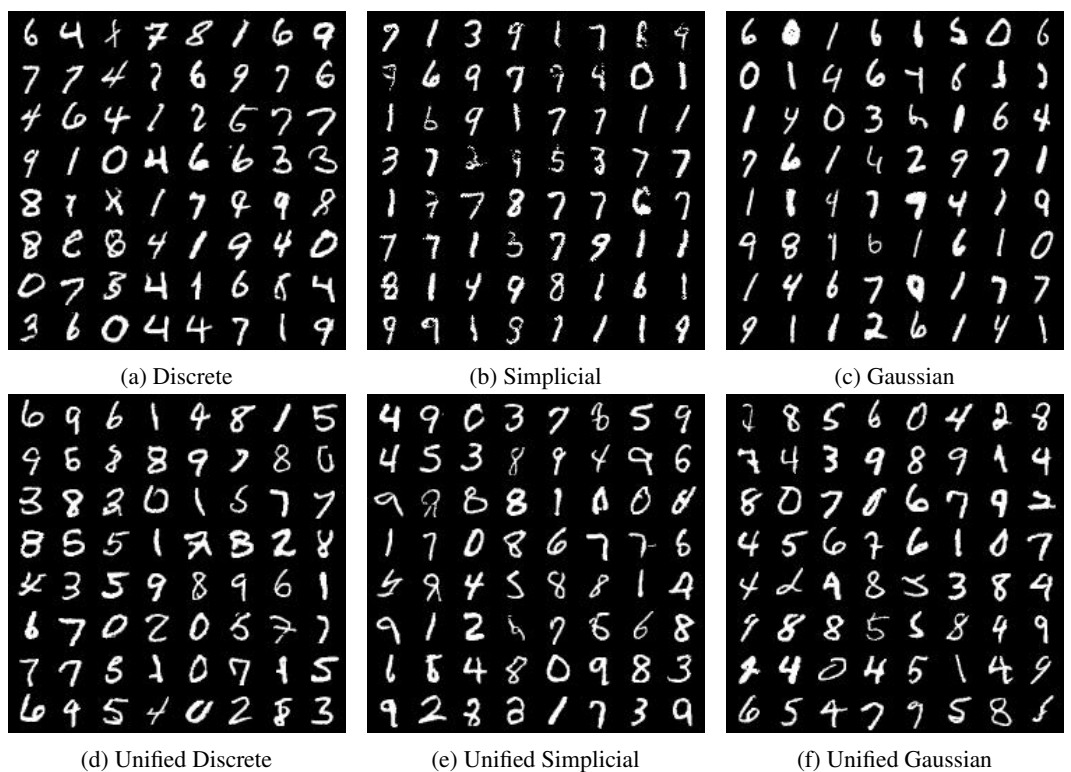

(a) Discrete      (b) Simplicial      (c) Gaussian

(d) Unified Discrete      (e) Unified Simplicial      (f) Unified Gaussian

Figure 12: **The SSP results in no noticeable drop in generation quality for image models.** We plot samples from models trained on MNIST.

