# OpenReview forum: "A Unification of Discrete, Gaussian, and Simplicial Diffusion"
_ICLR.cc/2026/Conference — ICLR 2026 Poster_

### Official Review · Reviewer_eBao · 2025-10-17

**Soundness:** 3
**Presentation:** 3
**Contribution:** 4
**Rating:** 8
**Confidence:** 3

**Summary:**

This paper addresses discrete data generation using diffusion models and proposes a unified framework that integrates direct discrete diffusion, simplex-based diffusion, and embedding-based Gaussian diffusion. Inspired by genetics, this approach enhances performance.

**Strengths:**

The paper reveals an intriguing connection between neutral evolution in genetics and the diffusion process in diffusion models. This bridge between the two fields allows for mutual conceptual enrichment and represents a high level of innovation.

The proposed SSP method is very interesting and inspiring, as it provides an intermediate layer that unifies all perspectives.

According to the experiments presented, the proposed method achieves significantly better performance than traditional approaches.

**Weaknesses:**

- The relationship between genetic drift and the n-simplex method is not clearly explained, and the paper would benefit from providing a more intuitive illustration of their connection.

**Questions:**

- There is a previous work of "Diffusion Evolution" that connects backward diffusion and forward evolution process. The authors proposed the opposite picture: the diffusion process corresponds to reverse evolution, while the denoising process represents forward evolution, naturally incorporating selection and reproductive isolation. How do the authors of the present paper view this distinction? Is there any conceptual connection between the two frameworks?
- Although this paper mainly focuses on discrete diffusion, it also builds a bridge between discrete and continuous spaces. So, I’m wondering if the framework presented in the paper can be applied to continuous diffusion?

---

> ### Author Response · Authors · 2025-11-23
>
> Thank you for your thoughtful review! We address your points and questions below. We have also added results over images to further show the generalizability of our method across many domains.
>
> > The relationship between genetic drift and the n-simplex method is not clearly explained, and the paper would benefit from providing a more intuitive illustration of their connection.
>
> We are happy to clarify this. Wright, Fisher and Kimura suggested versions of this model to study the behaviour of the behaviour of the prevalence of variants in a population under negligible selection. They were in particular interested in the fact that the prevalence of genes can increase or decrease even if it’s not under selection, purely due to stochastic fluctuations. These dynamics are called *genetic drift*. In the genetics literature, these models are used as null hypotheses when attempting to prove that a gene is under selection for example. We’ve added some of these details to the related works section in the newest version of the paper.
>
> > There is a previous work of "Diffusion Evolution" that connects backward diffusion and forward evolution process. The authors proposed the opposite picture: the diffusion process corresponds to reverse evolution, while the denoising process represents forward evolution, naturally incorporating selection and reproductive isolation. How do the authors of the present paper view this distinction? Is there any conceptual connection between the two frameworks?
>
> This is an excellent connection, and we will add this discussion to the related work! That work uses diffusion to suggest an optimization algorithm, “diffusion evolution”, but the authors do not formally connect it with biological evolution. In contrast, our work makes this connection explicit. As well, looking at App D.4, in many cases, diffusion corresponds *exactly* to biological evolution with selective pressure at time $t$ $\nabla p(\vec x_t|t)$. We have added this citation to the newest version of the paper.
>
> > I’m wondering if the framework presented in the paper can be applied to continuous diffusion?
>
> Our framework is primarily designed for discrete diffusion, but it can naturally extend to certain continuous settings. There are many continuous datasets which are directly represented using discrete classes. For example, image datasets like MNIST are often represented using discretized pixel values. Our framework can be directly applied in these settings, allowing a unification of discrete and continuous diffusion. Other continuous domains could also be discretized in a principled way, enabling our discrete diffusion methods to be used as an effective approximation.

---

### Official Review · Reviewer_JD9a · 2025-10-28

**Soundness:** 3
**Presentation:** 3
**Contribution:** 3
**Rating:** 4
**Confidence:** 3

**Summary:**

The paper introduces a one-shoe-fit-all framework that perform diffusion on discrete, gaussian, and simplical data based on Wright-Fisher model. The paper shows a maththematical connection between them, allowing to training multiple types of data at once and yield competitive performance on DNA geneartion.

**Strengths:**

1. Establishing a connection among discrete, Gaussian, and simplex which has a root from Wright-Fisher -- a human population genetics model. This insight is crucial for building next AI model where a model is not limited to any particular type of data.

**Weaknesses:**

1. Limited practical applicability of multi-domain inference: While Figure 7 demonstrates competitive performance when the model handles different data domains at test time, this scenario has limited real-world relevance. In practice, each modality has distinct statistical properties—images typically follow continuous Gaussian distributions, while text is inherently discrete. Using a discrete or simplex representation for images, or Gaussian diffusion for language, is suboptimal for each respective modality. This raises questions about whether a unified framework provides practical advantages over domain-specific approaches.
2. The paper lacks experiments on combined domains like image and text. Since the primary contribution is demonstrating the advantages of a unified framework, the authors should have a baseline to train and evaluate a model on both language and vision domains jointly. Without such experiments, the claimed benefits of the unification approach remain unsubstantiated.
3. The paper lacks vision experiments, despite images being continuous data that naturally align with Gaussian distributions.
4. The paper does not provide a suitable choice of parameterization for a specific data domain.


Minor concenrs:
1. Paper presentation is quite dense. Theorem 4.1 and 5.1 requires multiple readings to understand.

**Questions:**

Q1. The authors claimed the proposed method enable the flexibility of performing diffusion at all three domains at test time. I wonder if this still holds for large model (e.g. Stable Diffusion, Large Diffusion Language Model) where the modality-specific model is guaranteed with the scaling laws.

Q2. The experiment setting is limitted to a single modality, no experiment demonstrates the practical value of the unified framework across genuinely different data modalities. For example: Can a model trained on discrete text data leverage the continuous parameterization for improved performance? Can the same model joinly learn from discrete language tokens and continuous image embeddings?

Q3. What's the effect of the ζ parameter in practical settings? Ablation is needed.

---

> ### Author Response · Authors · 2025-11-23
>
> Thank you for your review. We address your points and questions below.
>
> > The paper lacks experiments on combined domains like image and text. Since the primary contribution is demonstrating the advantages of a unified framework, the authors should have a baseline to train and evaluate a model on both language and vision domains jointly.
>
> We would like to clarify an important misinterpretation. **Our paper does not unify different data domains (like images and text); instead, it unifies three distinct types of diffusion for discrete data: simplicial diffusion, discrete diffusion, and Gaussian diffusions.** Our contribution is the demonstration of how all three of these methods can be parameterized through the Wright Fisher population genetics model, enabling us to compare their noising processes and training objectives under one theoretical framework. Furthermore, this unified framework is domain-agnostic, and we have demonstrated its applicability in diverse domains such as proteins, DNA, language, and now images.
>
> Experiments which train one model across data modalities would not address our core claims because they do not provide insights into the differences between Gaussian, discrete or simplicial diffusion on discrete data.
>
> > The paper lacks vision experiments.
>
> Thanks for your suggestion. To further demonstrate the generalizability of our model across different domains, we have included new experiments on images. Specifically, we have added MNIST results to our paper in App G.2 and Figures 11 and 12 to further demonstrate the generalizability of our approach. We train four separate models: discrete, simplicial, Gaussian, and unified (SSP), and we find again that the unified model is competitive with the single-domain models in likelihood and sample quality. Please see our global response for more details.
>
> > Limited practical applicability of multi-domain inference… Using a discrete or simplex representation for images, or Gaussian diffusion for language, is suboptimal for each respective modality.
>
> The top-performing model may be difficult to reliably predict a priori. While certain domains have intuitive results (for example, we find that discrete diffusion is the most competitive for language), we find that this intuition may break down. For instance, in our experiments on protein sequences, we found that Gaussian diffusion outperformed discrete and simplicial variants, even though proteins are “discrete objects”. And in MNIST, discrete beats Gaussian even though images are "continuous objects".
>
> Therefore, the SSP enables practitioners to have access to all three types of models at inference-time, eliminating the need to guess the optimal model before training.
>
> > What is the effect of the $\zeta$ parameter in practical settings?
>
> We introduced the hyper-parameter $\zeta$ in Figure 1a and Section 4.1 as a theoretical device used to unify the three discrete regimes by modelling repeated letters, allowing us to understand loss and hyperparameter comparisons, and arrive at our practical advancements. In practice, the diffusion models we train in the paper are taken in the $\zeta=1$ (for discrete) or $zeta \rightarrow \infty$ (for simplicial and Gaussian) limits, and thus we do not explicitly use $\zeta$.
>
> > I wonder if this still holds for large models
>
> We designed our experiments to reflect full-scale settings in the language, DNA, and protein modeling literature. Our language and protein experiments use sequence lengths of 1024, comparable to standard benchmarks we cite, while our DNA experiments use sequence length of 500 to maintain consistency with baseline methods such as [1].
>
> While our evaluation covers realistic settings, it is always possible to continue to scale to even larger models; however, these experiments would require extensive computational costs that are beyond the scope of this work.
>
> [1] Stark et al, Dirichlet Flow Matching with Applications to DNA Sequence Design

---

> ### Comment · Reviewer_JD9a · 2025-11-24
>
> Thank you for adding the MNIST results! While the discrete approach performs favorably on this dataset, I have concerns about generalizability. MNIST uses 8-bin quantization on grayscale images (8 possible values), which is relatively easy for models to fit. However, for high-resolution RGB images (e.g., 256×256×3), this would yield only 8³ = 512 possible colors quite limited compared to the original 16.7 million colors. I am concerned that the findings based on MNIST may not extend to high-resolution image settings.
> That said, I appreciate the additional experiments demonstrating the method's flexibility in choosing any models to use at inference time. This is quite a free lunch. Overall, the proposed unified framework is theoretical-grounded and would offer valuable contributions to the community. Given the positive reception from other reviewers, I raise my score accordingly.

---

### Official Review · Reviewer_ia2Y · 2025-10-31

**Soundness:** 3
**Presentation:** 3
**Contribution:** 3
**Rating:** 6
**Confidence:** 3

**Summary:**

This paper unifies various approaches to diffusion under the view of population genetics (Wright-Fisher). They then use this theoretical result to draw from the literature of Wright-Fisher diffusion, practically improving existing simplicial approaches for DNA generation. They finally demonstrate that diffusion can be represented by a sufficient statistic parameterisation that is independent of the timestep and diffusion paradigm.

**Strengths:**

- Paper is well presented with a good mixture of visualisations for better understanding.
- Good theoretical contribution, being able to tie together disparate concepts into a unified framework will be helpful for future researchers aiming to navigate the field. Note that I am a little unconfident here with regards to the specifics of the theory, as I did not check it carefully, and it is out of the domain of my knowledge.
- Demonstrated practical impact from the Wright-Fisher perspective, improving performance on an applied task.
- The implications of SSP are somewhat interesting

**Weaknesses:**

The paper load for ICLR this year has been large, and so I have not been able to spend as much time as I would like on reviewing. I encourage the authors to correct any errors/misunderstandings I may have with regards to the paper.


1. **Experimental weaknesses**
    1. The DNA experiment is quite limited in scale (it is also not clear why the stabilised simplicial diffusion surpasses dirichlet FM).
1. **SSP motivation unclear**
    1. Although the idea of learning multiple diffusion paradigms in a single model is interesting, I think more clarity on the practical utility of such an approach might be of benefit. For example, I don't see a practical benefit in training an image generation diffusion model with an additional discrete diffusion objective on the 256 intensity levels. That is to say, I don't see that much benefit in SSP over comparing different diffusion approaches and picking the individual best.
1. **Better intuition**
    1. For the general reader interested in diffusion, the theoretical results of the paper are a little hard to intuit. However, I also appreciate that the authors have already included a number of helpful visualisations and I found the language and notation to be clear.
1. **Missing relevant references**
    1. [1] is a recent large scale DNA generation model that achieves SoTA by combining autoregressive and discrete diffusion/flow models.
    1. [2] propose to use gaussian diffusion for categorical data, but constrain the *clean data* distribution to the simplex hyperplane and train using cross entropy.


[1] Li et al, Absorb & Escape: Overcoming Single Model Limitations in Generating Genomic Sequences, NeurIPS 2024

[2] Eijkelboom et al, Variational Flow Matching for Graph Generation, NeurIPS 2024

**Questions:**

See above

My weaknesses are all fairly minor and I believe the main contribution (theoretical) is good.
However, this is a little unconfident as I am not knowledgeable about the theoretical domain explored in the paper. As such, I am unlikely to increase my score as I cannot confidently verify the main contribution.

---

> ### Author Response · Authors · 2025-11-23
>
> Thank you for your thoughtful review! We address your points and questions below. We have also added results over images to further show the generalizability of our method across many domains.
>
> > Although the idea of learning multiple diffusion paradigms in a single model is interesting, I think more clarity on the practical utility of such an approach might be of benefit… That is to say, I don't see that much benefit in SSP over comparing different diffusion approaches and picking the individual best.
>
> Indeed, ideally, we would just train three models and pick the one that works the best. Unfortunately, this requires **triple the cost** compared to training one individual model. The benefits of SSP is that all three types of diffusion can be trained in one single model, allowing users to explore multiple options at no additional cost.
>
> This is useful because the top-performing model may be difficult to reliably predict a priori. While certain domains have intuitive results (for example, we find that discrete diffusion is the most competitive for language), these intuitions may not hold. For instance, in our experiments on protein sequences, we found that Gaussian diffusion outperformed discrete and simplicial variants, even though proteins are “discrete objects”.
> And on new MNIST experiments, discrete outperformed Gaussian despite image data intuitively being continuous.
> Therefore, the SSP enables practitioners to have access to all three types of models at inference-time, eliminating the need to guess the optimal model before training.
>
> > The DNA experiment is quite limited in scale (it is also not clear why the stabilised simplicial diffusion surpasses dirichlet FM).
>
> Our DNA experiments use a sequence length of 500 to match the length of DNA of previous methods such as [1] for a fair comparison. Although we looked at 100 targets, we see that the error bars are very small. However, to further validate our results, we update the experiment to include another 900 samples and find the same patterns: our stable simplicial diffusion significantly outperforms baselines. We have added these experiments to the new version of the paper.
>
> There are two possible explanations for why our simplicial diffusion beats flow-matching. First, our method is trained with a maximum likelihood objective rather than a moment matching objective. ML estimators are known to be statistically more efficient, which can lead to improved performance. Second, and possibly more importantly, our sampler reverses an SDE while flow-matching simulates a deterministic ODE. SDE samples generally lead to higher-quality and more diverse samples compared to their ODE counterparts for diffusion [2].
>
> [1] Stark et al, Dirichlet Flow Matching with Applications to DNA Sequence Design
> [2] Song et al, Score-Based Generative Modeling through Stochastic Differential Equations
>
> > For the general reader interested in diffusion, the theoretical results of the paper are a little hard to intuit. However, I also appreciate that the authors have already included a number of helpful visualisations and I found the language and notation to be clear.
>
> We appreciate this very much. We included pictures and as much discussion as we could in the main text, but unfortunately had to move some discussions to the appendix due to space constraints. If you have any questions or suggestions about specific parts of the paper, we would be happy to update the text!
>
> > Missing relevant references
>
> Thank you for the references! We have added these to our related works.
>
> > New empirical results on images
>
> To further demonstrate the generalizability of our model across different domains, we have included new experiments on images.
> Specifically, we have added MNIST results to our paper in App G.2 and Figures 11 and 12 to further demonstrate the generalizability of our approach. We train four separate models: discrete, simplicial, Gaussian, and unified (SSP), and we find again that the unified model is competitive with the single-domain models in likelihood and sample quality.
> Please see our global response for more details.

---

> > ### Comment · Reviewer_ia2Y · 2025-11-26
> >
> > Thanks for taking the time to respond and update the paper. I think it would be good to add the above discussion comparing to Dirichlet flow matching to the main paper.
> >
> > As previously mentioned, I will keep my positive score as is, as I don't feel confident verifying the correctness of the main theoretical contribution.
> >
> > One final point on the SSP, it would be interesting in future work to scale up image experiments. For example, training simultaneously with a masked diffusion loss (e.g. MaskGIT) as well as standard gaussian diffusion (e.g. DiT) and seeing if these two objectives complement each other. (It is known that combining masking in some way or another with gaussian diffusion can improve training convergence, see Masked Diffusion Transformer)

---

### Official Review · Reviewer_YoGL · 2025-11-01

**Soundness:** 3
**Presentation:** 4
**Contribution:** 4
**Rating:** 6
**Confidence:** 3

**Summary:**

This paper proposes a unified formulation of discrete, simplicial, and Gaussian diffusion that, in different limiting cases, converges to each of these three diffusion formulations. The method is based on introducing repeats of tokens in a discrete sequence and diffusing each token independently with a Markov transition kernel. The theoretical results show that, in the limit of an infinite number of repeats, the proposed scheme converges to Gaussian diffusion. The proposed approach is evaluated on the protein modeling task.

**Strengths:**

The paper is well written, and the motivation and importance of the work are clearly explained.

The idea of introducing repeats is interesting and novel, as are the theoretical convergence results.

The source code is provided and well-structured.

**Weaknesses:**

Training a discrete diffusion model for an extended sequence is computationally expensive, so it is unclear why a practitioner would use the proposed approach. This is especially true for Gaussian cases, where working with large sequence lengths becomes infeasible. For example, if one uses a transformer with the number of repeats equal to 10, the full-attention complexity becomes 100 times more expensive. The computational efficiency is not discussed in the main part of the paper. Sequence length used in the evaluation experiments is 200 which is too small for a practical image or language modeling tasks.

Experimental verification is limited to the protein modeling task. There are no evaluations in visual or language domains, for example, on MNIST, where one could generate samples with both Gaussian and discrete diffusion and visually assess sample quality. This raises questions about the generalizability of the proposed approach.

**Questions:**

Line 101: If we consider softmax instead of argmax, the connection between gaussian and discrete diffusion in continuous time is achieved via Ito’s lemma. What will fail if we consider zero temperature in softmax?

Line 111: Could authors explain why simplex diffusion sacrifices the ability to calculate a likelihood? If we have an ODE as in [1] we could compute likelihoods via integration of Jacobians of the vector field.

Line 182: Why is the closeness of ELBOs described as paradoxical? The ELBO is continuous with respect to weak convergence of probability measures. If we take a distribution that is a finite combination of delta measures and convolve it with a Gaussian kernel of small variance, then for sufficiently small $\sigma$ the two distributions are close in Wasserstein distance (and hence in the weak topology). Therefore, if a probabilistic quantity is continuous in the weak topology, we should indeed expect these two distributions to yield close values of that quantity by definition of continuity.

What is the computational overhead of the proposed approach?

The reviewer is willing to raise the score if the weaknesses regarding computational complexity and generalizability beyond the protein modeling task are addressed.

[1] Stark et al. Dirichlet Flow Matching with Applications to DNA Sequence Design

---

> ### Author Response · Authors · 2025-11-23
> **Response [1/2]**
>
> Thank you for your thoughtful review! We address your points and questions below. We have also added results over images to further show the generalizability of our method across many domains.
>
> > Training a discrete diffusion model for an extended sequence is computationally expensive. What is the computational overhead of the proposed approach?
>
> We introduced the hyper-parameter $\zeta$ in Figure 1a and Section 4.1 as a theoretical device used to unify the three discrete regimes by modelling repeated letters, allowing us to understand loss and hyperparameter comparisons, and arrive at our practical advancements. However, **our diffusion models are not trained on these extended sequences**, nor do we instantiate repeated letters in our implementation. In practice, the diffusion models we train in the paper are taken in the $\zeta=1$ (for discrete) or $zeta \rightarrow \infty$ (for simplicial and Gaussian) limits, and thus we do not explicitly represent the repeated letters.
>
> Furthermore, for a hypothetical model which implements intermediate values of $\zeta$, every sequence position is represented as a single point on the simplex, as shown in Alg 3 and Section 4.1. Therefore, even this hypothetical model incurs **no computational overhead** compared to standard discrete training methods.
>
> > Sequence length used in the evaluation experiments is 200 which is too small for a practical image or language modeling tasks.
>
> Our language and protein experiments use sequence lengths of 1024, comparable to many standard tasks. Our DNA experiments use sequence length of 500 to keep the testing environment consistent with baselines such as [1].
>
> We only use length 200 when sampling new proteins *after training our models*; our models can sample sequences up to length 1024.
>
> [1] Stark et al, Dirichlet Flow Matching with Applications to DNA Sequence Design
>
> > Experimental verification is limited to the protein modeling task. There are no evaluations in visual or language domains, for example, on MNIST, where one could generate samples with both Gaussian and discrete diffusion and visually assess sample quality.
>
> We also validated in the DNA and language settings with likelihood and sample quality. In Figure 5 and Section 5.2, we show that our simplicial diffusion model outperforms existing methods on DNA. We also show the performance of our method on language in Figure 7 and Section 6.2
>
> Thanks for your suggestion for images; we have added MNIST results to our paper in App G.2 and Figures 11 and 12 to further demonstrate the generalizability of our approach. We train four separate models: discrete, simplicial, Gaussian, and unified (SSP), and we find again that the unified model is competitive with the single-domain models in likelihood and sample quality. Please see our global response for more details.
>
> > Line 101: If we consider softmax instead of argmax, the connection between gaussian and discrete diffusion in continuous time is achieved via Ito’s lemma. What will fail if we consider zero temperature in softmax?
>
> Indeed if one applies softmax to Gaussian diffusion, you get a diffusion on the simplex. (This is the basis for a few “simplicial” diffusion models as discussed under “Gaussian diffusion which appears as simplicial diffusion” in App B.) Then, as $T\to 0$ you get a diffusion process. However, in App. C we prove that this limiting process, despite being the limit of Markov processes, is not Markov. How can that be?
>
> This result is not surprising when considering the fact that at each $T>0$ we can exactly recover the original Gaussian diffusion using a reverse softmax. Yet at $T=0$, we lose a lot of information, we can only recover the quadrant that the latent Gaussian process lives in.
>
> Another way to describe the contradiction is that when $T$ is extremely small, the process fluctuates by very small amounts near each vertex, and only suddenly jumps between vertices. So it seems that we’re approaching discrete diffusion. What this picture hides is that in the pure mathematical setting, those tiny fluctuations can contain a lot of information about which direction we’re about to transition.

---

> ### Author Response · Authors · 2025-11-23
> **Response [2/2]**
>
> > Line 111: Could authors explain why simplex diffusion sacrifices the ability to calculate a likelihood? If we have an ODE as in [1] we could compute likelihoods via integration of Jacobians of the vector field.
>
> FM writes an objective, but it’s not an ELBO, so it cannot be compared to ELBOs. Nevertheless, as you suggest, one can in principle calculate the likelihood of an ODE model using a change of variables formula, even if this likelihood is not used as an objective.
>
> However we need to account for the difference between the discrete and continuous data. Avdeyev 2023 solved this by building an ELBO (see “Bounding Discrete Data Likelihood with Variational Lowerbound” in Avdeyev 2023). They were able to use the forward process for the variational posterior $q^{\mathrm{Diff}}$ but there is no such closed-form expression for flow matching. Furthermore, even if we could evaluate a similar ELBO for FM, it is not clear if this ELBO is tight beyond a toy setting in Avdeyev 2023. Indeed when we evaluated it on the model from Avdeyev 2023 we found that it gave 8 nats / position, far higher than the trivial 1.39 nats / position.
>
> We understand your position however and we have changed the text to say “these sacrifice the ability to **straightforwardly** calculate a likelihood”. This also reflects that the integral of the Jacobian is expensive to compute.
>
> > Line 182: Why is the closeness of ELBOs described as paradoxical? The ELBO is continuous with respect to weak convergence of probability measures. If we take a distribution that is a finite combination of delta measures and convolve it with a Gaussian kernel of small variance, then for sufficiently small  the two distributions are close in Wasserstein distance (and hence in the weak topology). Therefore, if a probabilistic quantity is continuous in the weak topology, we should indeed expect these two distributions to yield close values of that quantity by definition of continuity.
>
> The ELBO is not continuous with respect to weak convergence of probability measures. Put simply this is because it contains the KL divergence of the forward paths and backwards paths, and convergence in KL divergence is strictly stronger than weak convergence.
>
> One way to see this quickly is noting $N(0, \sigma^2)\to\delta_0$ as $\sigma\to 0$ in Wasserstein but $KL(\delta_0||N(0, \sigma^2))=\infty$ for all $\sigma>0$. In particular, if the ends of backwards paths are near the beginnings of the forwards paths in the weak sense, their supports may still not overlap, so the KL between forward and backwards paths can be infinite. In fact we show that the Gaussian diffusion ELBO is exactly infinite unless one uses the hollow parameterization (details in App F.3).

---

### Author Response · Authors · 2025-11-23
**General Response**

In response to reviewer feedback, we have included additional experiments which further motivate the unification of the different types of diffusion methods. Specifically, we expand our experiments to the image domain, and we also evaluate our methods on a new downstream task for antibody optimization. We have updated the pdf, and all of our edits are marked in purple text.

### Images

We evaluate our unified discrete diffusion framework on the MNIST dataset, consisting of 28x28 grayscale images. We compare the performance of our single unified model (SSP) to the performance of three individually-trained diffusion models: discrete, Gaussian and simplicial. Additional details are provided in Appendix G.2 of the updated pdf. Ultimately we see the similar, if not stronger, results as on protein and language data -- the SSP model is competitive with single-domain models.

We evaluate the model performance using validation likelihood, as shown in Figure 11, and see similar results as our Fig 7.
In fact, we are even able to achieve slightly better performance for discrete and Gaussian diffusion, perhaps because the SSP parameterization is easier to learn from, or because of a benefit from learning on diverse data.
We also generate 64 unconditional samples per model using 1,000 steps of ancestral sampling.
Through our visualizations in Figure 12, we see that the unified model does not lead to a noticeable drop in sample quality compared to individual models.

Interestingly, our results also highlight the utility of a single model trained across domains.
While we might expect Gaussian diffusion to achieve the best data fit due to the continuous nature of the data, we see the opposite: among our individual models, Gaussian surprisingly achieves the worst likelihood.
Therefore, a practitioner would have preferred to train a unified model rather than assume a priori that Gaussian diffusion would have resulted in the best fit.

### Antibody optimization

We also test our protein models on the downstream task of antibody optimization. Specifically, we test our diffusion models’ ability to improve the thermostability of antibody sequences. We sample 50 unique single- and double-point mutants of a parental VHH sequence by repeated noising and denoising, and selecting the top resulting sequence from each repeat experiment. Additional details are provided in Appendix G.1.

In Figure 10, we visualize the predicted improvement in melting temperature. We find that Gaussian diffusion led to the strongest results, despite protein sequences being a discrete domain, further motivating our unified method which enables practitioners to use different types of diffusion at inference-time. We also find that the unified model has competitive performance to the individual models.

---

### Meta-Review · Area_Chair_thDz · 2025-12-31

**Summary:**

The paper introduces a unified framework for diffusion in discrete domains and shows that discrete, Gaussian, and simplicial diffusion can be explained under the Wright-Fisher population genetics model. The authors then show the utility of the unified framework across three domains (after rebuttal): protein data, language, and images. Apart from providing a unified view, the framework shows additional empirical benefits in terms of efficacy.

**Reviewer Concerns:**

The paper generally received positive reviews, with some criticism of technical aspects (e.g., misunderstandings about the computational cost and multi-domain settings) that were resolved during the rebuttal. One recurring concern was with regard to the evaluation, as the authors initially only provided results on protein data and language settings. The authors provided additional results on MNIST during the rebuttal, indicating a similar trend to the remaining domains, though potentially limited, as MNIST may be too simplistic a vision data set. For future research, it would be interesting to extend the evaluation and investigate large-scale settings (e.g., ImageNet) across different domains.

**Reviewer Scores:**

Here is an estimate of how the scores might have changed after the rebuttal.

- YoGL: Likely to have increased the score (7/8)
- ia2Y: Decided to keep score, see rebuttal.
- JD9a: Raised score (5/6), see rebuttal.
- eBao: Likely kept initial score.

---

### Decision · Program_Chairs · 2026-01-26

Accept (Poster)